# Arctic sea ice radar freeboard retrieval from ERS-2 using altimetry: Toward sea ice thickness observation from 1995 to 2021

Marion Bocquet[1,2], Sara Fleury[1], Fanny Piras[2], Eero Rinne[3,4], Heidi Sallila[3], Florent Garnier[1], and Frédérique Rémy[1]

[1]LEGOS, Université de Toulouse, CNES, CNRS, IRD, UPS, Toulouse, France
[2]Collecte Localisation Satellites (CLS), Toulouse, France
[3]Marine Research, Finnish Meteorological Institute, Helsinki, Finland
[4]University Centre in Svalbard (UNIS), PO Box 156, N-9171 Longyearbyen, Norway

**Correspondence:** Marion Bocquet (marion.bocquet@legos.obs-mip.fr)

**Abstract.**

Sea ice volume's significant interannual variability requires long-term series of observations to identify trends in its evolution. Despite improvements in sea ice thickness estimations from altimetry during the past few years thanks to CryoSat-2 and ICESat-2, former ESA radar altimetry missions such as Envisat and especially ERS-1 and ERS-2 have remained underexploited so far. Although solutions have already been proposed to ensure continuity of measurements between CryoSat-2 and Envisat, there is no time series integrating ERS. The purpose of this study is to extend the Arctic radar freeboard time series back to 1995. The difficulty to handle ERS measurements comes from a technical issue known as the pulse-blurring effect, altering the radar echos over sea ice and the resulting surface height estimates. Here we present and apply a correction for this pulse-blurring effect. To ensure consistency of the CryoSat-2/Envisat/ERS-2 time series, a multi-parameters neural network-based method to calibrate Envisat against CryoSat-2 and ERS-2 against Envisat is presented. The calibration is trained on the discrepancies observed between the altimeter measurements during the missions-overlap periods and a set of parameters characterizing the sea ice state. Monthly radar freeboards are provided with uncertainty estimations based on a Monte Carlo approach to propagate the uncertainties all along the processing chain, including the neural network. Comparisons of corrected radar freeboards during overlap periods reveal good agreement between the missions, with a mean bias of 0.30 cm and a standard deviation of 9.7 cm for Envisat/CryoSat-2 and 0.20 cm bias and a standard deviation of 3.8 cm for ERS-2/Envisat. The monthly corrected radar freeboard obtained from Envisat and ERS-2 are then validated by comparison with several independent data sets such as airborne, moorings, direct measurements and other altimeter products. Except for two data sets, comparisons lead to correlation ranging from 0.41 to 0.94 for Envisat, and 0.60 to 0.74 for ERS-2. The study finally provides radar freeboard estimation for winters from 1995 to 2021 (from ERS-2 mission to CryoSat-2).

## 1 Introduction

Several indicators illustrate the evolution of sea ice in response to climate change. Arctic sea ice extent has strongly decreased since the beginning of satellite observation era by radiometry (Stroeve et al., 2012; Meier et al., 2014; Stroeve and Notz, 2018).

The proportion of perennial ice has decreased significantly since 1984, the amount has halved in April between 1984 and 2018 (Stroeve and Notz, 2018). The end of summer 2021 became the second lowest amount of multi-year ice since 1985 (Meier et al., 2021). To improve our knowledge and forecast its evolution, an additional dimension becomes crucial: the thickness. Thick and old ice is disappearing and being replaced by younger, thin ice that has a higher mechanical sensitivity. Thin ice is more prone to deformation (Stroeve and Notz, 2018) that induces area changes, and is more sensitive to climate hazards such as cyclones or strong winds (Rheinländer et al., 2022). Thickness is a key parameter for sea ice study, it varies a lot according to the regions and it modulates the sea ice volume evolution in the Arctic Ocean (Landy et al., 2022). Various campaigns have been carried out in the Arctic since the middle of the 20th century to measure sea ice thickness (Lindsay and Schweiger, 2013; Krishfield et al., 2014). However, these space and time-limited measurements do not allow conclusions to be drawn at basin-scale sea ice volume variations. A quasi-global approach is possible through satellite altimetry, especially with radar altimetry, which is not impacted by the cloud cover and whose missions are continuous since 1991.

Sea ice thickness estimation by spatial altimetry, in its modern form, is introduced by Laxon (1994) and Peacock and Laxon (2004) based on the freeboard methodology. The radar freeboard is obtained by taking the difference between the height measured above the floes and the height over leads interpolated below the floes. The radar freeboard has to be corrected of the radar signal slowdown within the snow layer to retrieve the ice freeboard, which is the thickness of the emerged part of the floe. Given the snow depth, the density of water, ice and snow, it is possible to derive the thickness of the ice assuming hydrostatic equilibrium. Lead and floe heights can be estimated using a heuristic retracker or, more recently developed, physical retrackers (Kurtz et al., 2014; Landy et al., 2019; Laforge et al., 2020). The implementation of this method requires the assumption that the Ku-band radar wave completely penetrates the snow layer, which is still widely discussed and is not the subject of a definitive consensus (Ricker et al., 2014, 2015; Nandan et al., 2017).

The launch of the CryoSat-2 (CS-2) mission and featuring a high-resolution Synthetic Aperture Radar (SAR) mode has enabled important advances in the estimation of sea ice thickness. The benefits are many, especially when compared with altimeters from past missions (ERS-1, ERS-2 and Envisat) operating with older technology, the Low-Resolution Mode (LRM) which has a larger surface footprint size, making thickness estimation more difficult. To reconstruct Envisat sea ice thickness estimation, Guerreiro et al. (2017), Paul et al. (2018) and Tilling et al. (2019) rely on the differences between Envisat and CS-2 during their common flight period to be able to calibrate the Envisat freeboard. Getting back to ERS missions, an additional problem coming from the instrument appears, the 'pulse-blurring' described in Peacock (1998) and in Peacock and Laxon (2004). Laxon et al. (2003) and Giles et al. (2008) published thereafter the first and last ERS thickness estimations for the Arctic and Antarctic sea ice so far (as a map averaging all the estimates of the different winters over the whole flight period of ERS-1 and ERS-2).

This study presents a method to recover a homogenous time series of the Arctic sea ice radar freeboard back to ERS-2 which aims to be used to describe sea ice thickness changes over the last few decades. To minimize intermission bias along with the series, ERS-2 freeboard estimates are adjusted on Envisat radar freeboard estimates which in turn have been previously adjusted on CryoSat-2, taking advantage of respective common flight periods. Consistency between missions is ensured by using the same processing chain regardless of the mission (before calibration), starting with the chosen retracking algorithm:

the empirical threshold first-maximum retracker algorithm (Helm et al., 2014) with a threshold of 50 % (TFMRA50). Note that the term 'radar freeboard' refers to the TFMRA50 radar freeboard in KU band for both LRM and SAR modes (depending on the mission). Since this LRM-TFMRA50 radar freeboard will be corrected to be consistent with CryoSat-2 and not conventionally obtained by making the difference of the height over floes and height over leads, it will be specified as NN FBr, which stands for radar freeboard adjusted using the neural network. This study does not present any new understanding concerning LRM waveforms retracking on sea ice and does not draw any conclusions about links between surface properties and TFMRA50 FBr from LRM, but will focus only on recovering a long and homogeneous time series between different altimeter technologies. We also present the method used to correct the ERS-2 measurements for the effect of pulse-blurring, which is a prerequisite for using ERS measurements over sea ice. The adjustment of LRM measurements on CryoSat-2 is performed using machine learning based on the surface state of the ice in Sect. 3. The associated uncertainties are derived using a Monte Carlo approach. Sect. 4 compares the monthly Envisat and ERS-2 radar freeboard data with various in-situ, space-borne data sets or other altimetry products available during this period. The time series is finally presented as a radar volume time series with trends estimation, providing a first overview of Arctic sea ice changes over the last 27 years.

## 2 Data

### 2.1 Satellite Altimetry Data

#### 2.1.1 Cryosat-2

CryoSat-2 is an ESA altimeter mission launched in 2010. With a nearly polar and geodetic orbit, it enables observations up to 88 °N that makes it particularly adapted for cryosphere observations. Additionally, CryoSat-2 incorporates nadir SAR and SARin technologies (Wingham et al., 2006). The two altimetry approaches exploit the Doppler capabilities of the instrument to reduce the along-track footprint from several kilometers to approximately 300 m compared to LRM (corresponding to a reduction of the footprint area from 5 $km^2$ to 180 $km^2$ (Stammer, 2018)). Increasing the along-track resolution of the aperture radar has led to considerable advances in estimating sea ice thickness. For this study, we use SAR and SARin data at 20 Hz, from the ESA baseline-D L1b product. We derive the radar freeboard for the 7 coldest Arctic months (from October to April) of each year, from November 2010 to present.

#### 2.1.2 Envisat

ESA's Envisat mission, was launched in 2002, reaching latitudes of 81.5° N/S. The satellite carried the radar altimeter RA-2 (operating in Low-Resolution Mode, LRM), with a high Pulse Repetition Frequency (PRF) of 1795 Hz allowing a large number of measurements per second to be performed, resulting in a better accuracy. The return pulses are averaged in batches of 100 to constitute each waveform (Roca et al., 2009). RA-2 L1b version 3 products including waveforms provided by ESA are used in this analysis. Sea ice freeboard is computed for the coldest 7 months of each year from October 2002 to March 2012.

The surface illuminated by the satellite is significantly larger in LRM than in SARM (by a factor of about 30). In addition, surface roughness determines the size of the illuminated footprint in LRM; the greater the roughness, the larger the footprint, whereas it is constant in SARM (Chelton et al., 1989; Raney, 1995). Therefore, surface roughness will have a greater effect on LRM range retrieval than the more nadir-focused SAR techniques (see Sect. 3.4 for more information). There is no waveform model for sea ice to account for the effect of roughness, and conventional retracking methods don't allow relevant radar freeboard estimation using LRM information alone. Our approach is to exploit these processing mode differences to derive an LRM-corrected freeboard. To this end, we compare Envisat and CryoSat-2 datasets during the mission overlap period from November 2010 to March 2012 (see Sect. 3.4).

### 2.1.3 ERS-2

In the 1990s, ESA has launched two European Remote Sensing (ERS) satellites, ERS-1 in July 1991 and ERS-2 in April 1995. ERS-1 was able to perform nominally until June 1996 and ERS-2 until November 2003. To extend the time series while ensuring the continuity with Envisat mission, ERS-2 products from the ESA Reaper project (Brockley et al., 2017) were used until July 2003. The ERS RA altimeter operated at a lower PRF than Envisat RA-2 altimeter (1020 Hz against 1795 Hz respectively). Thus, the 20 Hz waveforms are made up of 50 elementary echoes instead of 100 for RA-2. That leads to a higher speckle noise for ERS missions than for Envisat (see Sect. 3.5 for further details).

Another significant difference between RA and RA-2 comes from the tracker-board control loop that aims at centering the expected echo in the altimeter acquisition window. The delay between the transmission and the reception of the radar waveform depends on the vertical distance between the altimeter and the Earth's surface. This distance varies along with the satellite orbit and the ground topography. The time between the transmission of the radar wave and the opening of the acquisition window must therefore be constantly adapted (called window-delay (s) or tracker-range (m) if it is whether a time or a distance). The distance, or range, between the altimeter and the measured surface, is then equal to the sum of the tracker range and the epoch, i.e. the position of the waveform in the window. Since a waveform is an average of 50 individual pulses, it is important that each pulse is correctly centered in the window (to be aligned with each others). Otherwise, the resulting averaged 20 Hz waveforms will be blurred. This is unfortunately what happened to ERS altimeters over sea ice-covered surfaces (Peacock and Laxon, 2004).

### 2.2 Ancillary Data

Whether it is for the calculation of the radar freeboard itself, the LRM calibration or the comparison of our results to in situ data, we use various additional data sets. We present here additional data sets that have been used for that purpose.

The sea ice concentration field is needed to restrain the freeboard computation over sea ice covered area. The product used is the NSIDC 0051 product based on Nimbus-7 SMMR and DMSP SSM/I-SSMIS Passive Microwave Data (Cavalieri et al., 1996). This product is also used to compute radar freeboard volume in Sect. 4.3 as well as for the LRM/SARM calibration. The study also requires a sea ice type product, this information is derived from the NSIDC 0611 sea ice age product (Tschudi et al., 2019) that is aggregated into two classes (MYI and FYI) according to the oldest age of the ice within the grid cell (FYI

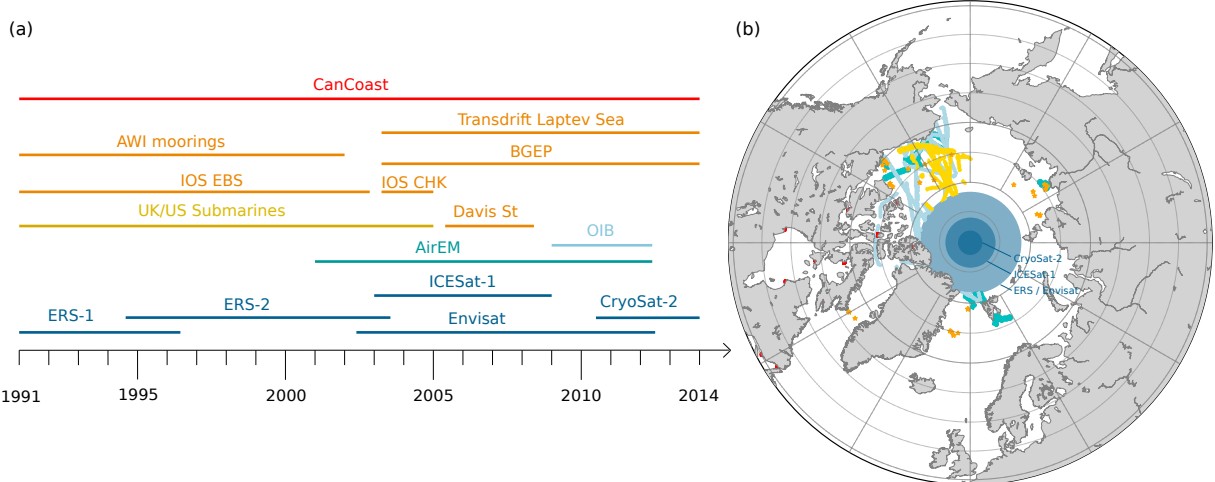

**Figure 1.** Summary of various available dataset for Envisat and ERS validation. Colors distinguish the different types of data. Dark blue for satellite products, light blue for airborne data, yellow for submarines, orange for anchored moorings, green for buoys and red for direct measurements. (a) Temporal availability (b) Spatial availability and extent of missions data gaps. Blue rounds represent altimeters coverage limitation due to their orbit inclination (81.5°N for Envisat, 86°N for ICESat-1 and 88°N for CryoSat-2)

: ice age between 0 and 1 year, MYI : ice age of at least one-year). Data are respectively available as daily and weekly map with a 12,5 km grid resolution. The proportion of multi-year ice of a given grid cell refers, in this study, to the mean ice type observed by all the tracks (for each month of each mission) that pass within a 25 km radius of this grid cell. This value is computed during the gridding step. The proportion would consequently be overestimated compared to what can be estimated

with ice age tracking algorithms.

SnowModel-LG (Liston et al., 2020a; Stroeve et al., 2020) is a snow depth product from a snow evolution model forced by different reanalysis, we use the version forced by ReAnalysis-5th Generation (ERA5) (Liston et al., 2020b). The data set is available from August 1, 1980, and July 30, 2018, in a 25 km resolution. Although the most commonly used product is still W99 climatology (Warren et al., 1999), it is no longer consistent for the recent period, as well as an altimetry-based

product such as ASD climatology (Garnier et al., 2021) would not be a relevant choice before 2010s, we then justify the use of SnowModel-LG to ensure the relevance of comparisons presented in Sect. 4.

## 2.3 Validation data

The results obtained in this study are compared with different independent data sets presented in this section. Comparisons are detailed in Sect. 4. Most of the following data sets are included in Lindsay and Schweiger (2013) data set and further described

in the corresponding publication Lindsay and Schweiger (2015). Data availability is summarized in Fig. 1.

### 2.3.1 Airborne

Operation Ice Bridge (OIB) was a mission led by NASA. It consisted of airborne measurement campaigns using scanning lidar altimeter and snow radar to measure both snow depth and ice thickness (Kurtz et al., 2013). The data we use are from the Unified Sea Ice Thickness Climate Data Record of Lindsay and Schweiger (2013), Operation Ice Bridge Version 2 processed by Kurtz et al. (2013). These measurements were carried out between 2009 and 2013, during each early spring or early autumn near the coasts of the Canadian Archipelago and Alaska.

Airborne Electromagnetic Induction (AirEM) can measure total thickness (snow plus sea ice), the methodology is described in Haas et al. (2009). Air-EM data that are used in this study is provided by (Lindsay and Schweiger, 2013) and are available from 2001 until 2013 from to 22 campaigns in the Arctic Ocean and Fram Strait.

### 2.3.2 Moorings & Submarines

The following data are all measured with upward looking instruments that are installed either on anchored moorings or on board of submarines. These instruments measure the sea ice draft i.e. the height of the immersed part, from which can be derived the sea ice thickness to be compared with altimetry data.

The Beaufort Gyre Expedition Project (BGEP) is composed of a network of 4 moorings located in the Beaufort Sea (Krishfield et al., 2014). The moorings, equipped with ULS record drafts every 2s with a precision evaluated to $\pm 0.3$ cm. Data are currently available from August 2003 to September 2018. The data were collected and made available by the Beaufort Gyre Exploration Project based at the Woods Hole Oceanographic Institution.

Belter et al. (2020) perform and diffuse a daily sea ice draft data set based on Upward-Looking Acoustic Doppler Current Profilers (ADCPs). These data are located in the Laptev Sea and are available from August 2003 to September 2016 located in the Laptev Sea.

The Institute of Ocean Sciences (IOS) provides two ULS draft measurement data sets named IOS-Eastern Beaufort Sea (IOS-EBS) and IOS-Chukchi Sea (IOS-CHK) (Melling, 2008). For IOS-EBS, data are available from April 1990 to September 2003 on a network of 9 sites in the Beaufort Sea. IOS-CHK is composed of data from a single site located in the Chukchi Sea between August 2003 and August 2005. The sea ice draft product for IOS-EBS comes from Melling (2008), and IOS-CHK from Lindsay and Schweiger (2013). Draft can be measured with precision about 0.05 m for young ice and can be overestimated up to 0.3 m for older and rougher ice.

Davis Strait sea ice draft (Davis St) product from anchored moorings has been detailed in Drucker et al. (2003). Data used in the study comes from the Unified Sea Ice Thickness Climate Data Record (Lindsay and Schweiger, 2013) and are available from 2005 to 2008.

Alfred Wegener Institute for Polar and Marine Research (AWI) have processed and distributed a sea ice draft data set consisting of data from 11 moorings in the Greenland Sea and Fram Strait (Witte, 2005). The data span from 1991 to 2002, and the draft is recorded with a 5 minutes frequency with an accuracy of $\pm 0.2$m.

The last sea ice draft data set presented in this section is derived from data collected by both U.S. Navy and Royal Navy submarines in the Arctic Ocean from 1975 to 2005 (National Snow and Ice Data Center, 2006; Wadhams and Horne, 1980; Wadhams, 1984; Wensnahan, 2005). It gathers data from 39 cruises. According to Rothrock and Wensnahan (2007), sea ice drafts are estimated to have an overall bias of 29 cm and a standard deviation of 25 cm from the actual draft.

### 2.3.3 Coastal stations

Environment and Climate Change Canada compiled weekly measurements from 27 monitoring stations along the coasts of the Canadian Archipelago in one product named CanCoast. Measurement methods can vary from one station to another (boreholes, hot-wire thickness gauges, etc.) but all stations provide at least sea ice thickness and snow depth estimation with an accuracy of less than a centimeter.

### 2.3.4 Satellite Altimetry products

Three satellite altimeter sea ice freeboard products have also been used for comparisons.

Guerreiro et al. (2017) presents the first Envisat radar freeboard dataset consistent with CryoSat-2 mission. Radar freeboards are available as monthly maps from November to April between 2002 and 2012 between 65°N and 81.5°N. This dataset will be referred as Envisat PP because LRM TFMRA50 FBr was corrected using the Pulse Peakiness (PP).

The product generated by the ESA Climate Change Initiative Program (CCI) for sea ice thickness estimation, (Hendricks et al., 2018), includes the whole Arctic sea ice-covered region for all winters (October-April) of Envisat mission (2002-2012). It provides monthly grids of sea ice thickness, radar and ice freeboard combined with the related uncertainties. The corresponding methodology is described in Paul et al. (2018). In this study, this product will be named Envisat CCI.

Finally, ICESat-1 was a mission operated by NASA, was launched in 2003 and ceased operating in 2009. It was composed of a laser altimeter that allows retrieving the total sea ice freeboard (snow depth plus sea ice freeboard). ICESat-1 product provides estimations for 15 periods of about 30 days between February 2002 and November 2008 in a 25 km grid resolution, the version used is the NASA-Goddard one processed by Zwally et al. (2008); Yi and Zwally (2009). The ICESat-1 total sea ice freeboard measurement accuracy is estimated to be about 0.05 m.

## 3 Methods

### 3.1 ERS pulse-blurring correction

The values of the on-board tracker heights ($h_{trk}$) while ERS-2 flights over sea ice reveal instabilities of several meters that can not be explained by the sea surface topography. This phenomenon is better observed by computing the Surface Height Anomaly ($h_{rtrk}$) over all type of surfaces. This height anomaly is calculated according to Eq.(1), where $alt$ is the altitude of the satellite, $range$ is the on-board tracker range, $epoch$ is obtained after the waveform retracking using TFMRA50, $MSS$ is the DTU15 Mean Sea Surface, (Andersen et al., 2016) and $geophysical\_corrs$ is the sum of all geophysical corrections.

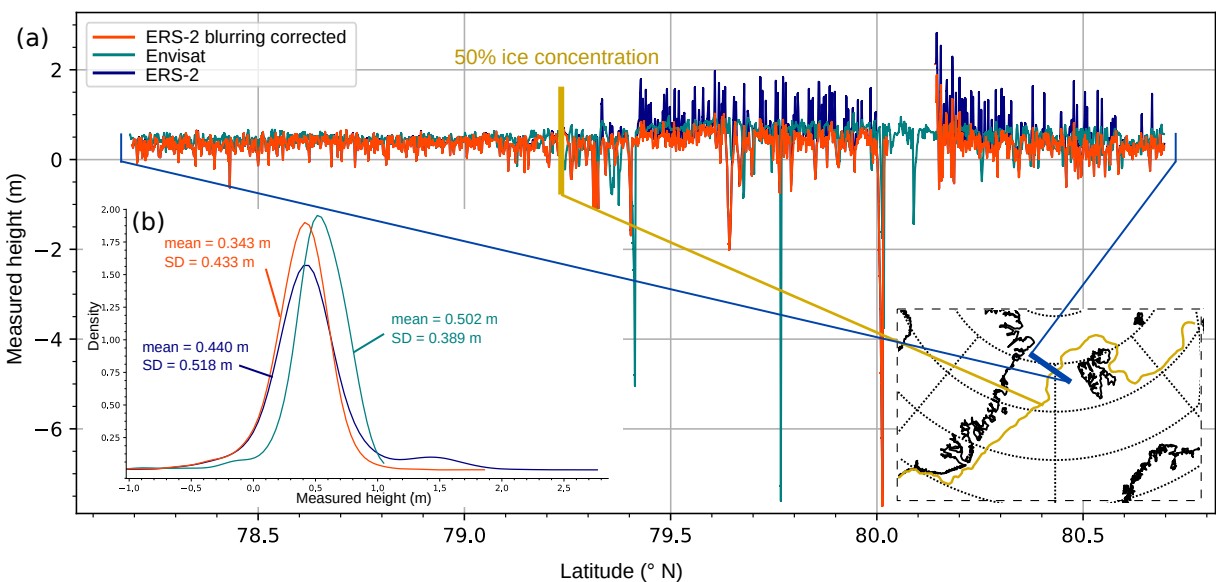

**Figure 2.** Profiles of surface height anomaly over sea ice and ocean for pass 25 between 78° N and 81° N for Envisat in blue-green (cycle 12), ERS-2 in blue and ERS-2 blurring corrected in orange (cycle 80). The red line represents the limit of 50 % concentration of sea ice, so as the limit between open ocean and an ice-covered area. The dark blue line shows the location of the pass between Svalbard Island and Greenland. (a) The surface height along the latitude and (b) the probability density function of surface height for the three passes with the associated statistics, the average and the standard deviation (SD). The color legend is identical for both sub-figures.

Figure 2, shows the TFMRA50 retracked height estimation for Envisat and ERS-2 along the collocated pass 25 respectively for cycles 12 and 80 (beginning of January 2003). Regarding the behavior of Envisat measurements, ERS-2 heights anomalies
show instabilities of about 1 m that make the measurements unusable.

$$h_{rtrk} = alt - range - epoch - MSS - geophysical\_corrs \tag{1}$$

The instabilities of the height anomalies (mainly over sea ice) are known as "pulse blurring" and are a consequence of the on-board tracker settings. This phenomenon occurs for both ERS-1 and ERS-2 missions. A simplified version of the ocean mode tracking system is represented in Peacock (1998, p.71). The tracking system is composed of three tracking loops to
maintain echoes within the radar acquisition window: the Height Tracking Loop (HTL), the Slope Tracking Loop (STL), and the Automatic Gain Control (AGC). The role of the HTL is to maintain the successive waveforms in the middle of the acquisition window. For this purpose, the tracking system is able to estimate the position of the tracker for each individual echo that composed the average sequence. The tracker position is therefore adapted at 1020Hz with a low-pass $\alpha\beta$ filter described by Eq.(2) and Eq.(3) according to the Height Tracking Loop error $\varepsilon$. $T$ is the interval for the low-pass filter to be updated,
$T = \dfrac{1}{PRF}$s for the HTL and $h_n$ the tracker height for the $n^{th}$ echo, with $n$ between 0 and 49. In ocean mode, the algorithm

used to estimate $\varepsilon$ is a Sub-optimal Maximum Likelihood Estimator (SMLE). Nevertheless, the SMLE has been developed for Brown-like waveforms, that can be found over the open ocean, but is not suitable for specular waveforms found over sea ice.

$$h_n = h_{n-1} + \alpha\varepsilon + T\dot{h}_n \tag{2}$$

$$\dot{h}_n = \dot{h}_{n-1} + \frac{\beta\varepsilon}{T} \tag{3}$$

Both Eq. (2) and Eq. (3) can be combined to give Eq. (4) which shows that the range window correction increases with a $n^2$ factor.

$$h_n = h_0 + nT\dot{h}_0 + n\alpha\varepsilon + \frac{n}{2}(n+1)\beta\varepsilon \tag{4}$$

A high $\varepsilon$ (due to an inappropriate sea ice height error estimation algorithm (Roca et al., 2009)) coupled with this low-pass filter could drive to the large variation of range window inside the same averaging sequence, which will "blur" the final averaged

waveform echo due "the bad overlay between the individual echoes" (Peacock and Laxon, 2004). Because the height error is estimated at the end of each averaging sequence, a sudden change in the range window is then possible, especially since the error estimate is also affected by the pulse-blurring and can explain tracker height oscillations. The problem mainly comes from the choice of the SMLE to estimate the HTL error. Indeed, it has been elaborated for ocean-like waveforms with a long trailing edge contrary to peaky waveforms whose power decreases suddenly after the max power peak.

A methodology has been developed by Peacock and Laxon (2004) to deal with this issue. This method consists of finding a relation between the height error parameter $\varepsilon$ and the difference between the measured surface (over an area covered by ice) and the "same area if it was not covered by ice" (Peacock, 1998). We interpreted it as the difference between the raw surface measurements and the interpolated ocean level measurements $\Delta h$.

Figure 3 illustrates that our interpretation of Peacock (1998) theory (on the right) fits with his results (on the left). Our results

reproduce the linear relation between $\Delta h$ and $\varepsilon$ found by Peacock (1998) and Peacock and Laxon (2004) with a slope equal to $\frac{-1}{5}$ when $\varepsilon$ is negative (Eq.(5)).

$$\begin{cases} h_{corr} = & h - \dfrac{\varepsilon}{5}, \quad \varepsilon \leq 0 \\ h_{corr} = & h, \qquad \varepsilon > 0 \end{cases} \tag{5}$$

This correction is applied and presented in Fig.2. The correction of the pulse-blurring effectively reduces the instabilities of measurements. The correction is similarly asymmetric so that the variations towards the positive height anomaly are more

corrected than the others. The corrected surface height anomaly of ERS-2 now appear more similar to Envisat in terms of noise and amplitude of variation. For this particular pass, the standard deviation has been reduced by 16 % and get closer to Envisat's one. Figure A1 shows more results on the impact of blurring correction on ASA (All Surface Anomaly, e.g. floes and leads) noise reduction comparing to Envisat during a whole cycle (80 for ERS and 12 for Envisat).

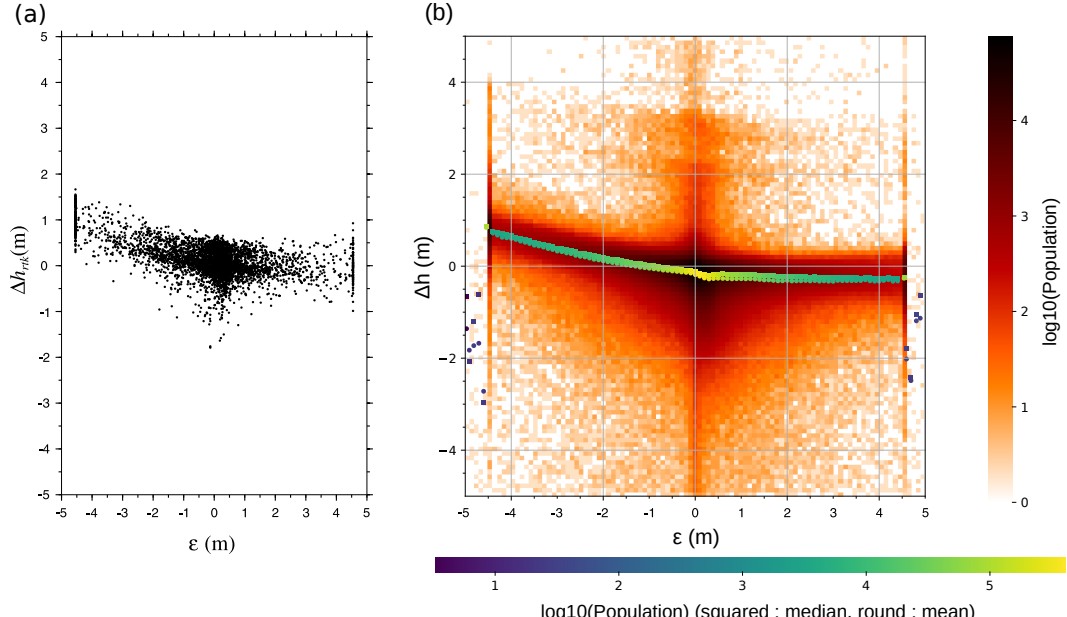

**Figure 3.** Difference between the raw surface measurements and the interpolated ocean level measurements as a function of the height error parameter $\varepsilon$. (a) is taking from Peacock and Laxon (2004) and (b) a reproduction for the Cycle 83 of ERS-2. On (b) squares and circles are respectively the median and the mean of $\Delta h$ for each value of $\varepsilon$ (on the x-axis).

## 3.2 Along track radar freeboard retrievals

This section aims to describe the radar freeboard (FBr) processing chain. This procedure is common to all missions to preserve homogeneity and continuity.

The FBr is the difference between the sea ice surface height measured over floes and the sea surface height measured over leads. The use of satellite altimetry for these estimates was introduced by Laxon (1994); Laxon et al. (2003). Sea ice freeboard (FBi) can be derived from FBr after a correction of the lower wave-propagation speed into the snow layer (Kwok, 2014). In this 245 study, we focus on the radar freeboard (without the speed propagation correction) to avoid the introduction of errors related to snow depth estimation.

It is therefore necessary to discriminate leads and sea ice floes. This essential step is based on the peakiness of the radar waveform that quantifies the specularity of the surface (Laxon et al., 2003; Peacock and Laxon, 2004), we use the definition of Guerreiro et al. (2017) (Eq.(6)). Pulse peakiness thresholds can depend on the radar altimeter. To ensure the continuity 250 between Envisat and ERS-2, these thresholds have been adapted to keep the same lead/floe proportion during their common flight period, see Tab.A1.

$$PP = \frac{max(WF)}{\sum_{i=0}^{Nb_{WF_{bins}}} WF_i} \tag{6}$$

Range are estimated using the TFMRA50 retracker for all surfaces and all missions mentioned here to maintain inter-missions continuity. Height anomaly measurements are then expressed relatively to the DTU15 Mean Sea Surface (Andersen
et al., 2016) and corrected for the common geophysical corrections (oceanic, polar, solid earth and load tides as well as tropospheric and ionospheric corrections) according to Eq. (1). The tide model used is the FES14 from Carrere et al. (2015). ERS-2 the pulse-blurring correction (detailed in Sect. 3.1) must be applied at this stage of the freeboard computation.

Surface height anomaly is then split into two variables using the Pulse-Peakiness classification explained above, the "ice level anomaly" (ILA), over floes and the "sea level anomaly" (SLA), over leads. ILA and SLA outliers are removed by filtering
data that are outside the interval : rolling mean $\pm$ 3 rolling Standard Deviation, with a 60 km large sliding window. After filtering, ILA and SLA are smoothed using a rolling mean at 12.5 km, then SLA and ILA are linearly interpolated (including bellow floes for SLA and above leads for ILA) and are again smoothed using a rolling mean at 12.5 km. No limit of distance is used to discard radar freeboard, but the interpolation as well as smoothing and filtering is not done between values separated by land. Indeed, the processing is done within ocean segments, separated by land, in order to isolate statistics between segments.
In this study, we will only use the FBr measurements that are made over floes, indeed, the LRM data correction, explained in section 3.4, is based on floes characteristics.

## 3.3 Data gridding

We performed the LRM FBr correction using monthly maps in EASE2 (Brodzik et al., 2012) with a 12.5 km resolution. The FBr gridding is done by averaging the values within a 25 km radius from each pixel weighted by the inverse of the uncertainty,
for other variables not weighting is applied. Only sea ice concentrations above 50% are considered. To limit the outliers of LRM FBr at the sea ice-ocean boundary, data with ice concentration lower than 85% is removed when the waveforms have a LEW higher than 2.5 gates.

## 3.4 Correction of LRM radar freeboards against SARM freeboards using neural networks

The radar freeboard maps obtained from the process presented in sect. 3.2 for Envisat and CryoSat-2 are shown in Fig. 4.
Important differences between Envisat and CryoSat-2 can be noticed, both in terms of patterns and in mean values. Negative radar freeboards are mainly due to the retracker choice. Indeed, a TFMRA50 is used to retrack heights on both leads and floes, this introduces a bias on the height over leads. The TFMRA threshold to retrack heights over leads should be closer to 80% and the use of a 50% threshold corresponds to the position of the retrack point for ocean surfaces, not specular ones Poisson et al. (2018), the surface over leads is measured to be higher than it is and even higher than the surface over floes. The SLA bias
(over leads) is evaluated constant for SARM altimeter in the study of (Laforge et al., 2020), this conclusion is also relevant for LRM altimeters as waveforms over leads are peaky and similar from a lead to an other. This positive constant bias over leads results to a negative bias on the radar freeboard. To avoid this bias, the retracker threshold could be adapted for leads or the SLA could be corrected on CryoSat-2 one. Nevertheless, a threshold of 50% ensure the stability of the range (Poisson et al., 2018)[fig.9]) contrary to higher thresholds (80%-95%) that could lead up to 47 cm of random error on the SLA. A TFMRA
at 50 % for both leads and floes is preferred in this study as a constant bias is easier to correct than an undetermined random

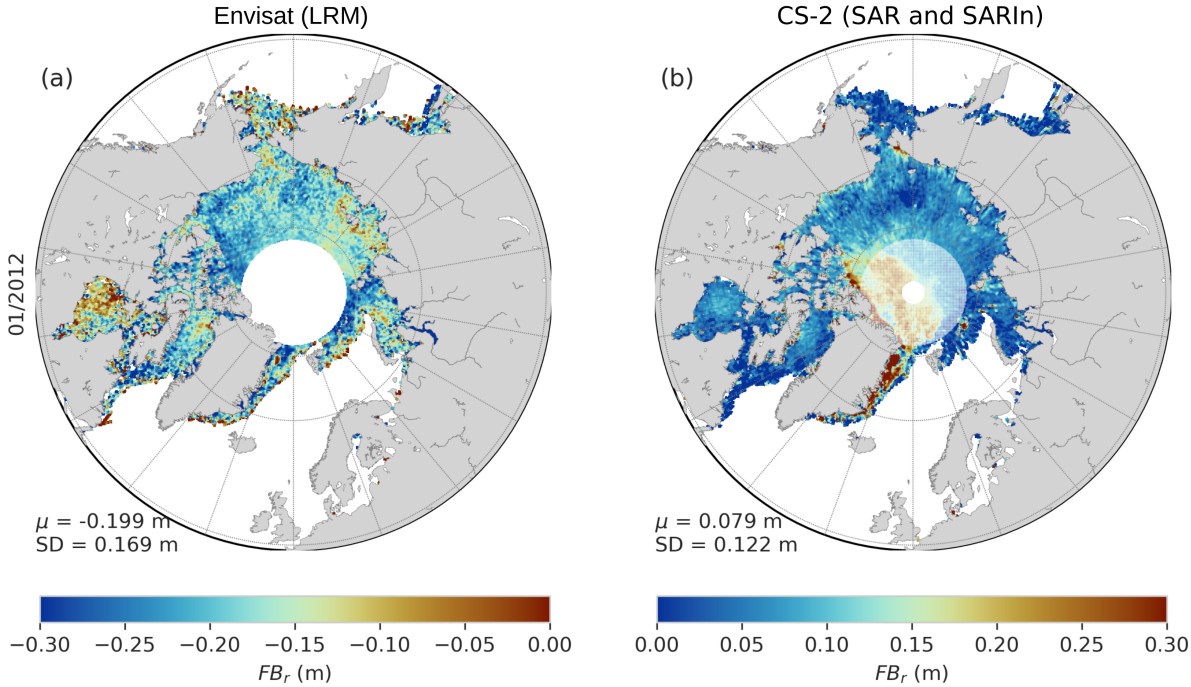

**Figure 4.** Pan-Artic radar freeboard maps for January 2012 for (a) Envisat uncorrected and (b) CryoSat-2

error. Nevertheless, beyond this bias, a lack of representative sea ice patterns can be observed, for instance, thick ice regions do not appear for the Envisat mission (cf Fig.4). This phenomenon also appears in the ERS-2 FBr since it is also in LRM with the TFMRA50 retracker. This inconsistency comes from the LRM itself (plus the retracker choice), which has a larger footprint than the resulting one in SARM. The larger footprint leads to a large impact of the surface roughness on the reflected echo. It is therefore impossible to distinguish the contribution of heights and roughness without a physical model of the effect of sea ice roughness on the reflected echo.

Paul et al. (2018) and Guerreiro et al. (2017) show that there is a significant correlation between the patterns of the parameters characterizing the surface roughness and the differences between the FBr of CS-2 and Envisat. These observations led to the development of correction methods of Envisat relatively to CS-2 taking advantage of the missions-overlap period. In the same way, we propose to correct ERS-2 against Envisat, once corrected, using the common flight period between Envisat and ERS-2.

So far, several empirical methods of Envisat freeboard correction using Cryosat-2 have been developed (Guerreiro et al., 2017; Paul et al., 2018; Tilling et al., 2019). In Guerreiro et al. (2017), the correction consists of finding a link between Envisat and CS-2 freeboard differences and the PP of Envisat's waveforms. The correction proposed by Paul et al. (2018), adapts the TFMRA threshold of Envisat waveform retracking according to the Leading Edge Width (LEW) of the waveforms and the surface backscatter. Tilling et al. (2019) suggests correcting the Envisat sea ice thickness (SIT) utilizing the distance between leads and floes. All methods are based on the comparison of monthly gridded data. The first two studies share the

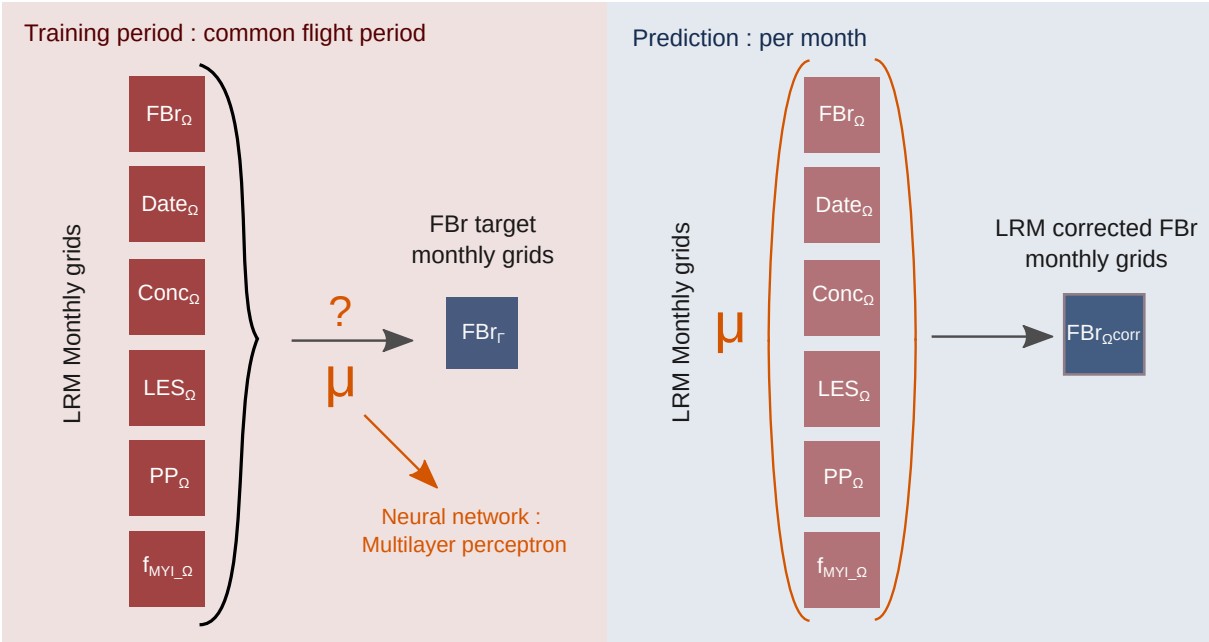

**Figure 5.** Diagram illustrating the principle of freeboard correction by neural network with the two main steps: on the left panel, the neural network training phase, and on the right panel, the prediction (correction) phase. $\Omega$ corresponds to the inputs and $\Gamma$ the output of the Neural Network. FB: radar freeboard, Conc : Sea ice concentration, LES: Leading Edge Slope, PP, Pulse Peakiness, $f_{MYI}$, MYI proportion.

common approach of trying to correct the Envisat FBr using the surface roughness (characterized by one or more parameters as proxy). They both propose a 3rd degree polynomial function to link Envisat and CS-2 FBr differences with surface properties. Paul et al. (2018) was the first to propose to use two distinct parameters that characterize two roughness scales to correct

LRM measurements that can impact differently the waveform shape. Our method follows the same approach (Pulse Peakiness, Leading Edge Slope) with other additional parameters that define sea ice state such as ice concentration, ice type, season, etc. The Leading Edge Slope refers to the LE-height divided by the LE-width computed between 30% and 70% of the max power of the first waveform peak. The LRM correction model procedure is based on a neural network in order to manage strong non-linearities. The procedure is illustrated on Fig. 5.

For each mission to be corrected, the neural network (NN) is trained on the common flight period between the mission considered as a reference and the one to be corrected. It takes as inputs monthly grids of the following parameters: LRM FBr (to be corrected), pulse peakiness, leading-edge slope, ice concentration, MYI proportion, and the period and as a target, the FBr of reference (SARM FBr or LRM-corrected FBr). Note that the period of the year is taken to capture better the seasonal variability, as snow on sea ice as well as sea ice physical properties change along the seasons. Inputs and targets are standardized

before the training step.

The neural network used is a multilayer perceptron (MLP). LRM FBr correction have been performed with Scikit learn (Pedregosa et al., 2011). The MLP is composed of 4 hidden layers, each composed of 100 neurons. The choice of hyperparameters

: number of neurons, the learning rate, the regularization term, batch size, activation functions, solver for the weights optimization, have been done using a grid search method. The evaluation criterion, called the score, is chosen as the determination

coefficient. Models are trained on 90% of the data set and tested on the remaining 10%, the splitting in random. During the tuning step, models are cross validated, it means that they are each trained 5 times with the same combination of hyperparameters but without the same train/test data set, the 5 scores are then analyzed to determine the best combination. Cross validation gives a better idea of the model performance as the dependence to the training data set is limited. The activation function for the hidden layers neurons is a sigmoid, related by possible negative radar freeboard values, and the optimizer ADAM (Kingma

and Ba, 2014). Moreover, in order to avoid over-fitting, an early stopping criterion is used to stop the model training as soon as the score is not improved during 10 consecutive iterations, with a defined tolerance (the validation fraction is set to 10% for the early stopping).

Finally, once the hyperparameters combination is set (see Tab.A3 for hyperparameters selection), the MLP is trained on the whole data set to provide the correction function. The trained model is then applied to the LRM monthly grids to obtain a

330 monthly LRM-corrected radar freeboard.

### 3.5 Radar freeboard uncertainties quantification

This section aims to estimate the uncertainties for Envisat and ERS-2 FBr estimations. The uncertainty budget is split into two steps corresponding to the two main parts of the freeboard processing chain. The first step covers the along-track processing up to the gridding which is common to all missions, and the second step concerns the correction of the LRM freeboard which

consists of predicting the corrected FBr with the neural network (NN FBr).

The uncertainty budget methodology concerning the first part is taken from Landy et al. (2020) and Ricker et al. (2014). We assume that for this step there are 3 sources of uncertainty. Two of them are random uncertainties: the speckle noise, largely discussed in Wingham et al. (2006) and the accuracy of the Sea Level Anomaly (SLA) measurement. The last one is linked to both retracker choices, surface roughness and snow radar signal penetration (Ricker et al., 2014).

According to Wingham et al. (2006), the speckle noise generates an "error" ($\sigma_{l1}$) from 7 to 14 cm (depending on the acquisition mode) on the range measurement for CryoSat-2 altimeter. Estimations of speckle noise "error" on range for other missions can be relied on the individual echoes number used to compute the averaged waveforms (for LRM sensors). Indeed, the speckle noise generates an "error" on the range that function of $\sqrt{n}$, where $n$ is the ratio between the number of individual pulses used for each averaging sequence for CS-2 and for the mission we want to estimate the "error" (Calafat et al., 2017) (see

Tab. A2 for LRM RA characteristics). All final $\sigma_{l1}$ values are summarized in Tab. 1. Note that Wingham et al. (2006) uses the terminology "error" but it refers to uncertainty.

SLA uncertainty ($\sigma_{SLA}$) estimation is taken from (Ricker et al., 2014). The uncertainty on the SLA is estimated to be the standard deviation of the SLA within a sliding window of 25 km if there are some leads within this window. If not, the SLA uncertainty is taken as the difference between the interpolated and smoothed SLA and the mean SLA computed as the mean

of raw SLA measurement at leads within a segment of ocean (if the pass is over land, statistics are made segment of ocean by

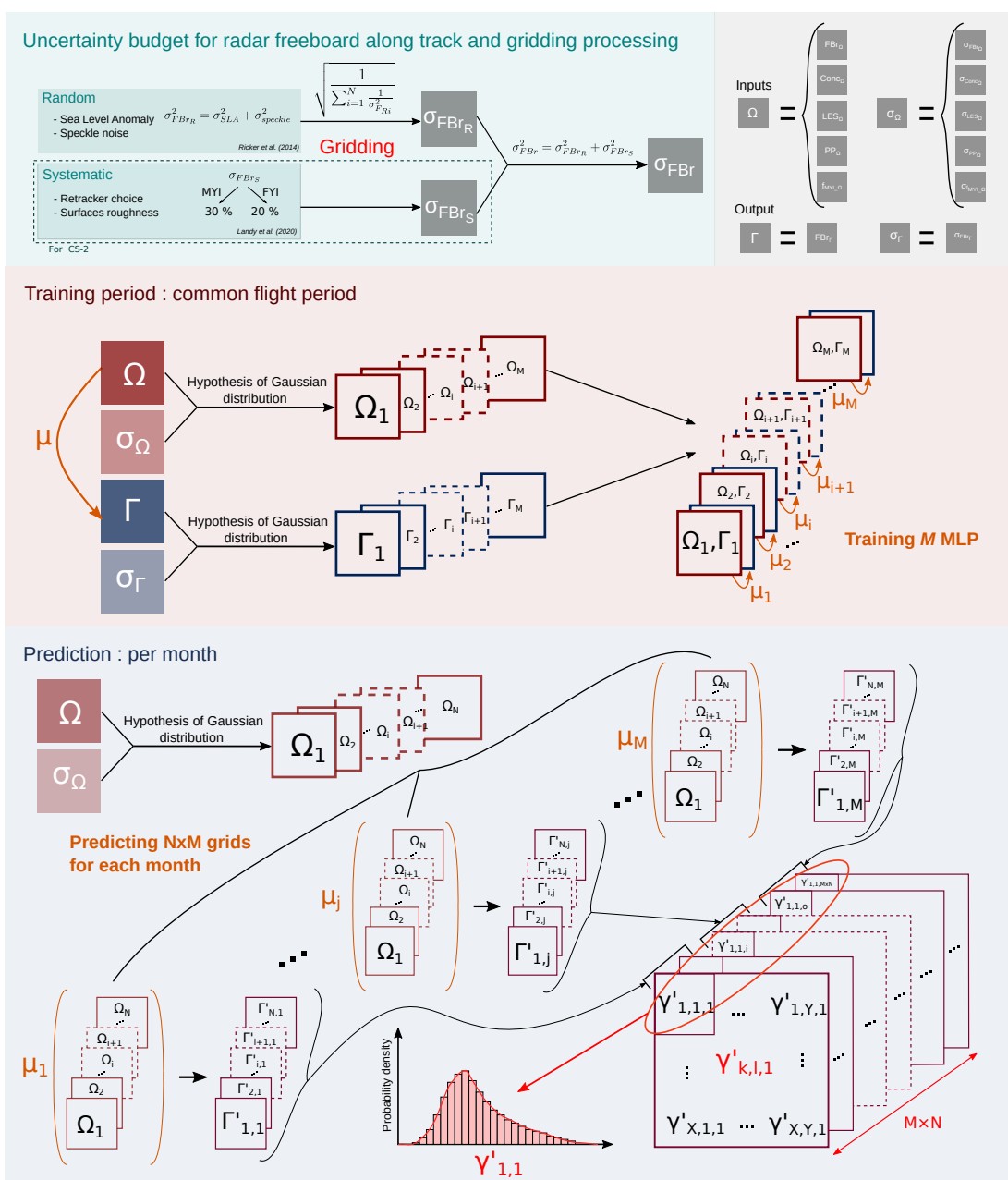

**Figure 6.** Summary diagram of uncertainty budget during along track, gridding and training, correction steps. Top left panel corresponds to the along track to grid uncertainty budget. Top right panel defines the notations, for the Monte Carlo procedure : $\Omega$ for the Neural Network input parameters, $\Gamma$ for the Neural Network output parameter (radar freeboard) with $\sigma_\Omega$ and $\sigma_\Gamma$, the corresponding uncertainties. The middle panel corresponds to the training of M models with noisy inputs and outputs. Bottom panel show the predictions of the N noisy input with the M trained neural network. $\gamma$ is the predicted radar freeboard estimation for one pixel of the MxN predictions. M=100, N=200.

**Table 1.** Summary of the "range error" for various missions governed by the speckle noise (Wingham et al., 2006)

| Mission (RA mode) | $\sigma_{l1}$ (m) |
|---|---|
| Cryosat-2 (SAR) | 0.10* |
| Cryosat-2 (SARIn) | 0.14* |
| Cryosat-2 (LRM) | 0.07* |
| Envisat | 0.068 |
| ERS-2 | 0.096 |

* Wingham et al. (2006)

segment of ocean). Finally, we consider that $\sigma_{SLA}$ and $\sigma_{l1}$ are not correlated and can be combined to give the random part of radar freeboard uncertainties ($\sigma_R$) following Eq.(7).

$$\sigma_R^2 = \sigma_{SLA}^2 + \sigma_{l1}^2 \tag{7}$$

The radar freeboard (including uncertainties) gridding methodology is taken from Ricker et al. (2014, Sect. 2.4) in order to take into account the random uncertainties in the radar freeboard gridding process.

In Landy et al. (2020), the FBr systematic uncertainty budget is decomposed in two parts, on the one hand, the uncertainties due to the penetration of the signal in the snow (depending on its salinity or if it is composed of metamorphic snow, according to the type of ice) and in the other hand, the surface roughness. We assume, as in Ricker et al. (2014), that the comparison of the freeboard from different retrackers does not enable to separate the contribution of the roughness from the signal partial penetration. We therefore assume to consider both sources as one mixed contribution, estimated to be respectively about 20 % and 30% of the sea ice thickness for FYI and MYI (Landy et al., 2020). The systematic uncertainties can be underestimated, as the penetration of the radar waves in the snow uncertainty may be poorly handled. Note that this systematic uncertainty budget only concerns CS-2 mission which is afterward propagated to Envisat and ERS-2, indeed other missions will be corrected based on CS-2 estimates.

Therefore, we combined random and systematic uncertainties with a quadratic sum to have the total radar freeboard uncertainty in a grid as well as for the related radar freeboard estimation. The uncertainty of the other inputs (LES, PP, sea ice concentration, MYI proportion), is considered to be, for each grid cell, two times the standard deviation of the measurements used to calculate the average value (grid cell value) divided by the number of tracks passing through the corresponding grid cell.

As explained in sect. 3.4, the LRM radar freeboard correction is predicted by a neural network. The uncertainty propagation through the neural network is not straightforward, since the inputs and outputs of the model are not linked by a mathematical relation. We have chosen to estimate the propagation of uncertainties through a Monte Carlo approach. This method allows to propagate uncertainties through non-analytically represented systems. Nevertheless, the Monte-Carlo method requires a representation of the distribution of the input and target parameters. As discussed above, there is no thorough knowledge of

the distribution of the input data for the NN. The method consists in training a number $M$ of NN with noisy inputs. The noise has been added to all inputs (for each grid cell and each month) according to a Gaussian distribution centered on the estimated value and the corresponding uncertainty as standard deviation. Trainings and predictions are done for all the noisy inputs/output, then the distribution of MxN radar freeboard predictions (from the M noisy NN models applied on N noisy inputs) has been analyzed of each grid cell and each month. The whole uncertainty budget process is summarized in Fig. 6.

Unfortunately, we have not been able to identify known distributions such as normal, log-Normal, gamma, etc., for predicted FBr distributions. However, we can still derive various statistics, such as the quantiles at 2.5% and 97.5%, which represent 95% of the values or the standard deviation to describe the FBr distribution for each pixel of all monthly grids.

## 4   Results

This section first presents the correction performance when applied to Envisat with respect to CS-2 and thereafter to ERS-2
with respect to Envisat. After this we present the Envisat and ERS-2 freeboard comparisons against independent validation data sets.

### 4.1   Correction performances

LRM correction methodology is presented in Sect. 3.4, it was successively applied to Envisat (Env) with respect to CS-2, and then to ERS-2 with respect to Envisat NN FBr. Figure 7 compares Envisat NN FBr and CS-2 FBr during December 2010 and
390 April 2011. Figure 8 presents the same feature for Envisat and ERS-2 NN FBr during December 2002 and April 2003.

    As can be noticed on these maps, Envisat correction allows recovering typical patterns of the Arctic sea ice with thick ice near the coasts of the Canadian Archipelago and Greenland, and thinner ice on the eastern part of the basin.

    Nevertheless, compared to CS-2, the radar freeboard of thick ice is slightly underestimated, and thin ice is slightly overestimated. However, the mean difference between the two products is close to zero (1 or 2 mm) for both months. We also note
that the correction results in an asymmetric distribution, with a tendency towards a log-normal distribution at the basin scale, whereas the CS-2 distribution for the same mask is centered and appears Gaussian-like.

    Radar freeboards of ERS-2 and Envisat (see Fig.8) are even more similar. Again, the bias is negligible, and the standard deviation (SD) of their difference is two to three times lower than for Env/CS-2, with a SD of 0.03 m and 0.05 m.

    The better performance of ERS-2 correction can be explained because ERS-2 and Envisat carried a similar LRM altimeter,
and they flight on a same orbit with 28 minutes delay between the two missions which allow them to observe nearly the same surface. On the other hand, CryoSat-2 operates in SARM and flies on a quasi-polar orbit with cycles and sub-cycles very different from Envisat ones. These comparisons show that the neural network correction gives satisfying results, at least during the common flight periods. The correction function is then applied to the 10 years of Envisat mission and 8 years of ERS-2 one.

Table A4 and Table A5 respectively represent statistics for Envisat/CryoSat-2 and ERS-2/Envisat radar freeboard for each month of missions-overlap periods. For both correction, averaged radar freeboard are close. The highest mean difference

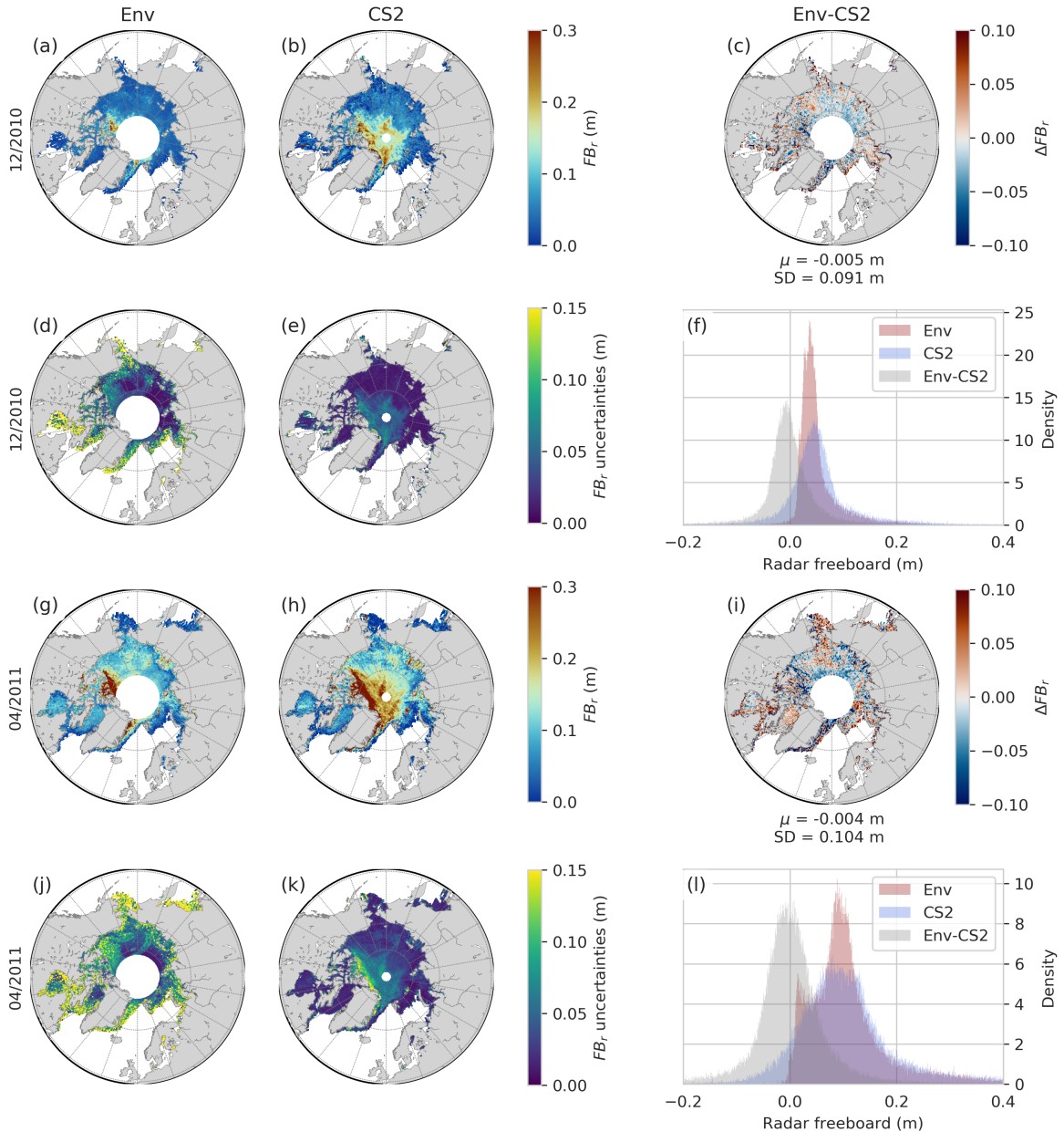

**Figure 7.** Comparison of Envisat NN FBr against CryoSat-2 FBr for December 2010 in the upper half and April 2011 in the lower half. Maps (a) and (g) refer to Envisat aside with the corresponding CryoSat-2 radar freeboard (b) and (h). Maps bellow (d), (e), (j) and (k) are the related uncertainties. The right column presents freeboard difference maps (Env-CS-2) ((c) and (i)). (f) and (l) are the distribution of Envisat $FBr$ in red, CryoSat-2 $FBr$ in blue and $\Delta FBr$ in grey. Histograms only include common data between Envisat and CryoSat-2, data north or 81.5 °N are excluded. $\mu$ refers to the mean difference and $SD$ to the Standard Deviation of the difference.

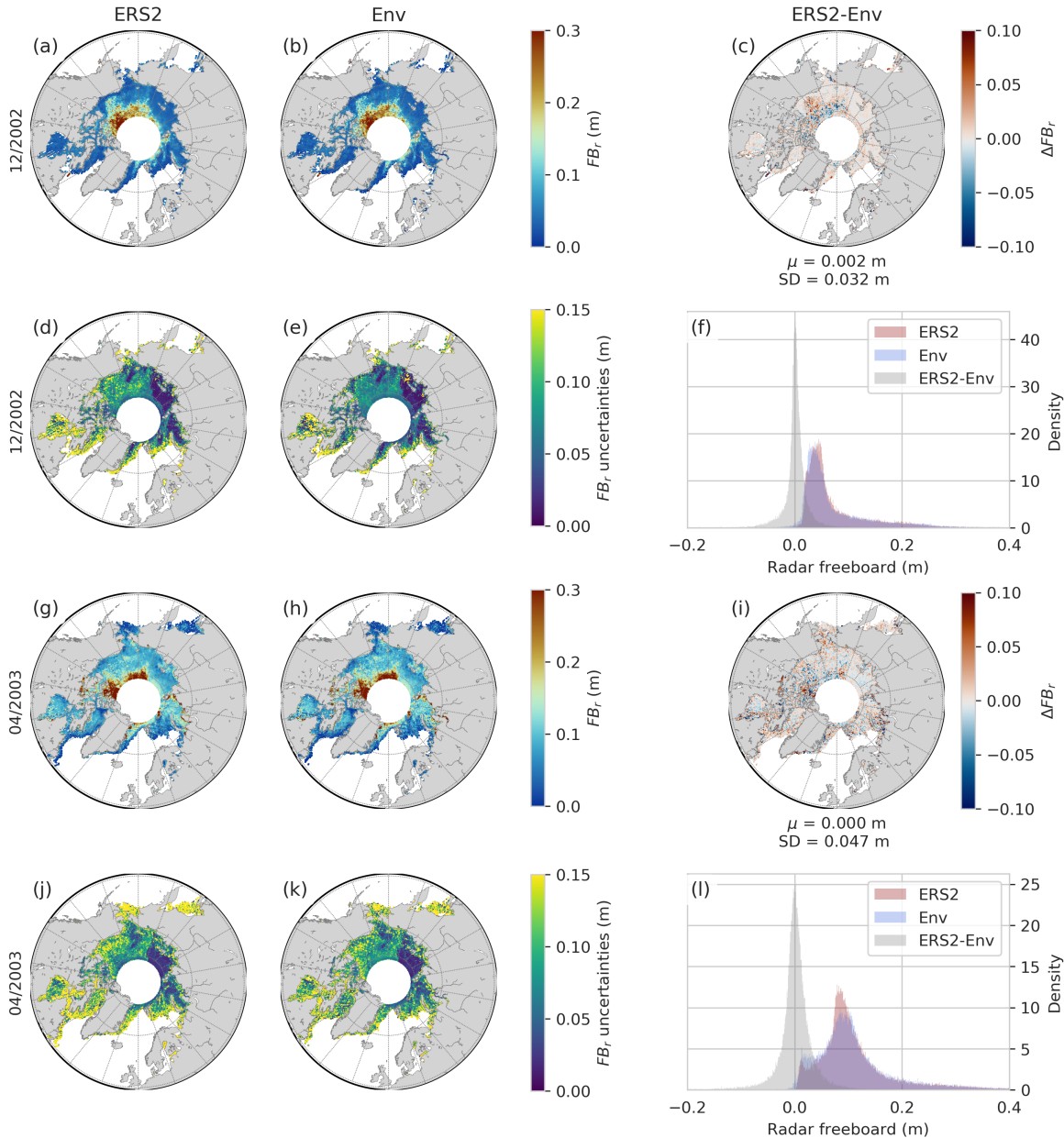

**Figure 8.** Same as Figure 7 but for ERS-2 and Envisat during December 2002 and April 2003. Histograms only include data for the coinciding region between ERS-2 and Envisat. $\mu$ refers to the mean difference and $SD$ to the Standard Deviation of the difference.

reaches 7 mm in February 2011 for Envisat, i.e. 9.5% of the mean Envisat NN FBr. Concerning ERS-2, the mean difference between ERS-2 and Envisat NN FBr does not exceed 3 mm, 3.3% of ERS-2 mean NN FBr. Concerning all the overlap period,

the mean FBr difference is 3 mm for Env/CS-2, 4.1% of Envisat mean NN FBr and -2 mm for ERS-2/Env one, about 2.2% of ERS-2 mean NN FBr.

In both Fig.7 and Fig.8, the uncertainties presented for Envisat and ERS2 are in fact two times the standard deviation of the respective NN FBr distributions as output of Monte Carlo simulations for each pixel grid. Detailed statistics for uncertainties are also provided in Table A4 and Table A5.

The median uncertainty on the radar freeboard for the period 2010/2012 is 6.3 m for Envisat and 2 cm for CS-2, regardless of the month, the mean and median uncertainties of Envisat are always larger than those of CS-2. Concerning the period 2002/2003, the median uncertainty is 8 cm for ERS-2 radar freeboard and 7.3cm for Envisat, similarly, statistics on uncertainties are globally higher for ERS-2 estimates (see Tab.A4 and Tab.A5 for detailed statistics).

## 4.2 Validation

In this section, Envisat and ERS-2 NN FBr are evaluated against a large set of independent data. These data are presented in Sect. 2.3 and include in situ, airborne and space-based measurements providing sea ice freeboard, draft or thickness. Data have been converted to sea ice thickness to make it comparable, except for SI-CCI and LEGOS-PP Envisat products which also provide FBr. The conversions are based on the hydrostatic balance assumption of sea ice covered by snow in sea water that is described in Eq.(8).

$$SIT = \frac{\rho_w}{\rho_w - \rho_i} \cdot FB_i + \frac{\rho_s}{\rho_w - \rho_i} \cdot h_s \tag{8}$$

The snow depth ($h_s$) is taken from the data set itself if given (e.g. OIB and CanCoast) with a constant snow density, $rho_s = 300\text{kg} \cdot \text{m}^{-3}$. Otherwise, the SnowModel-LG, ERA5 version is used for the snow load. The water density value used is $\rho_w = 1024\text{kg} \cdot \text{m}^{-3}$, $\rho_i^{FYI} = 917\text{kg} \cdot \text{m}^{-3}$ and $\rho_i^{MYI} = 882\text{kg} \cdot \text{m}^{-3}$ (Alexandrov et al., 2010). The sea ice density depends on the MYI-proportion within the grid cell, such as described by Eq.(9).

$$\rho_i = (1 - f_{MYI})\rho_i^{FYI} + f_{MYI}\rho_i^{MYI} \tag{9}$$

For Ku-band measurements, the speed reduction of the wave in the snow layer is taken into account to obtain the ice freeboard $FB_i = FB_r + h_s(\frac{c}{c_s} - 1)$ (Mallett et al., 2020) with $c$ the speed of light in vacuum and $c_s$ the speed of light in the snow estimated to be $c_s = c(1 + 0.00051\rho_s)^{-1.5}$ determined by Ulaby et al. (1986).

The different data sets are then gridded into monthly EASE2 grids of a 12.5 km resolution, as to facilitate the comparison to Envisat and ERS-2 monthly grids. Concerning moorings or coastal measurement stations, data are averaged to get one value per month.

#### 4.2.1 Envisat

Although numerous data sets are available during Envisat flight period, the spatio-temporal coverage of the Arctic basin remains very patchy. The following comparisons are presented with several types of sensors to reinforce the relevance of the validation. The consistency and discrepancies are discussed in the following.

Figure 9. gathers comparisons between Envisat and different data sets coming from airborne, space borne, submarines, drifting buoys and fixed coastal stations, Fig.10 presents comparisons with fixed moorings.

Comparisons presented in Fig. 9 provide satisfying statistics, with correlation values between 0.41 (CanCoast) and 0.71 (Envisat SI-CCI). Correlation between Envisat and OIB is in good agreement with Kurtz et al. (2014) that showed that the auto-correlation of OIB varies from 0.46 to 0.60. Nevertheless, Air EM data are poorly correlated with Envisat and statistics

reveal high bias and RMSE (up to 1 m). Disregarding CanCoast and space-borne estimations, comparisons reveal negative bias with Envisat from -3.9 cm to -2.9 cm, which could suggest an underestimation of Envisat sea ice thickness. The relevant statistics with CanCoast, with a bias of 8.9 cm and a RMSE of 6.4 cm suggest that this underestimation could not only be attributed to Envisat but maybe also to snow depth or other parameters. However, bias remains within the estimated range of uncertainty. The bias between OIB and Envisat estimation could also be attributed to the OIB snow depth whose estimation is

sensitive to the algorithm used (Kwok and Haas, 2015; Kwok et al., 2017).

The comparison with ICESat-1 data reveals a strong dispersion with a low bias of 20 cm and a correlation of 0.50. Envisat radar freeboard product established in the framework of the CCI and the version presented in this study are coherent with a bias close to zero, a standard deviation of the difference of 5.2 cm and a fairly high correlation of 0.71. Envisat CCI and Envisat NN data sets are consistent with a low mean bias, whereas Envisat-PP radar freeboard from Guerreiro et al. (2017) present a

thinner mean FBr than Envisat NN estimations.

The Envisat NN solution is also compared to several moorings (BGEP, IOS, Davis strait and Transdrift Laptev sea) data sets (Fig. 10). The four campaigns yield fairly high correlations with Envisat NN estimates, greater than or equal to 0.62 and reasonable standard deviations of 50-60 cm, down to 32 cm for the Laptev Sea estimates from Transdrift. While Envisat has low negative bias of -9.2 cm with respect to BGEP, other campaigns biases are largely negative, with values from -53 cm to

460    -76 cm.

Figure 10 (b) also provides a comparison between Envisat SI-CCI version and BGEP similarly to Fig. 10 (a). With respect to BGEP estimations, the statistics are slightly better for the product presented in this study than for SI-CCI's. Nevertheless, the two products seem to be relatively consistent with each other.

#### 4.2.2 ERS-2

The ERS-2 freeboard are also compared with measurements from AirEM, US and UK submarines and CanCoast in Fig.11.

The results of ERS-2 validation are close to those of Envisat. Figure 11 shows similar strong discrepancies with Air EM, but even better results for CanCoast and submarines comparison in terms of correlations (respectively 0.60 and 0.74 instead of 0.41 and 0.55 for Envisat) and standard deviations of the differences are equivalent to those found for Envisat (62 and 57

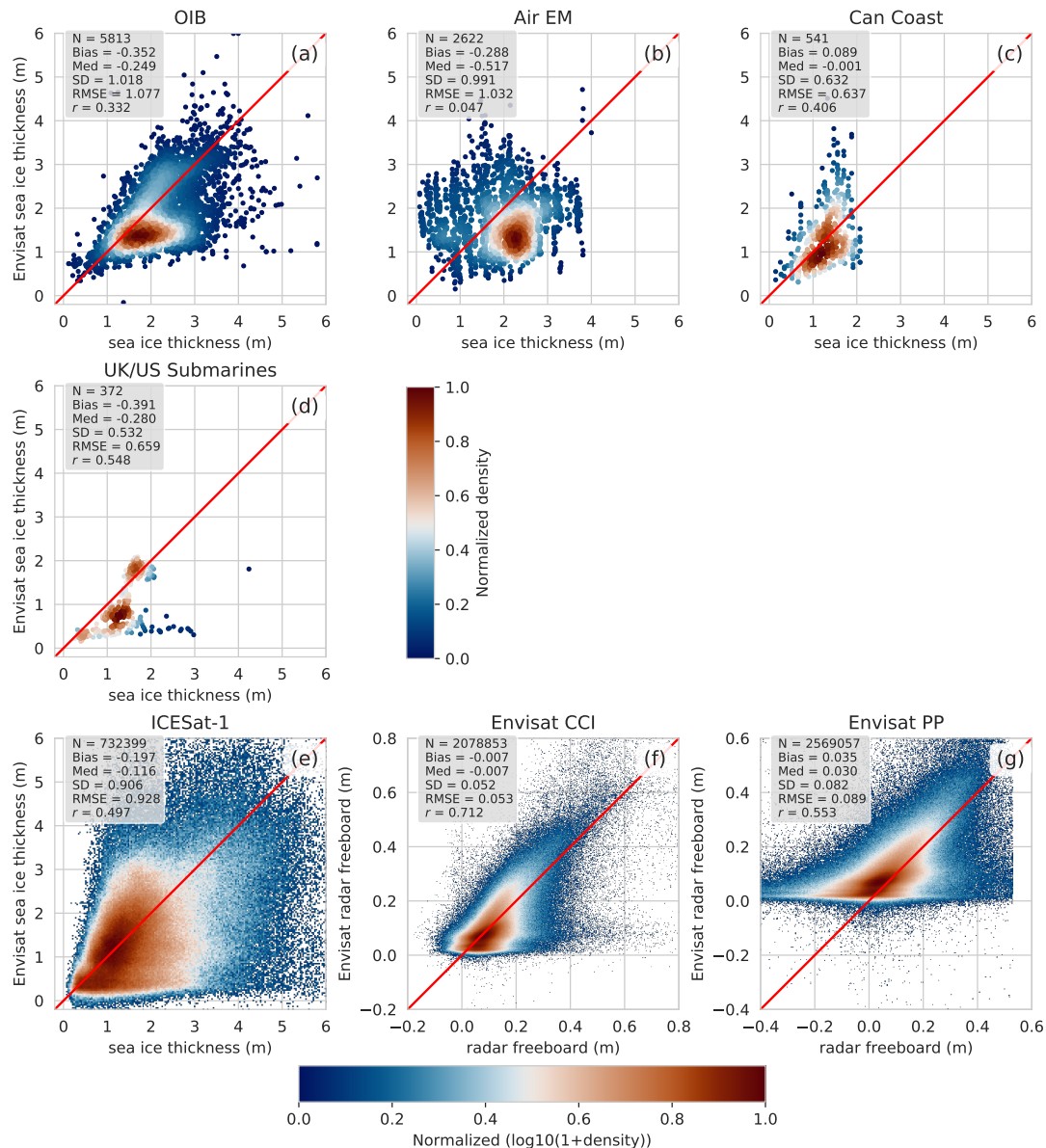

**Figure 9.** Comparative scatter-plots between Envisat NN sea ice thickness or radar freeboard estimations and other data sets. The x-axis indicates the sea ice thickness from (a) OIB total ice freeboard, (b) Air EM snow plus ice thickness, (c) Can Coast ice thickness, (d) UK/US submarines draft and (e) ICESat-1 total freeboard. (f) compares our Envisat radar freeboard with SI-CCI Envisat solution and (e) with Envisat-PP solution from Guerreiro et al. (2017). Colorbars represent the normalized density. A $log_{10}$ has been applied before the normalization for (e) and (f) due to the large number of data. N is the number of the couple of values that are compared, Med refers to the Median, SD the Standard deviation, RMSE the Root Mean Square Error and $r$ the correlation coefficient.

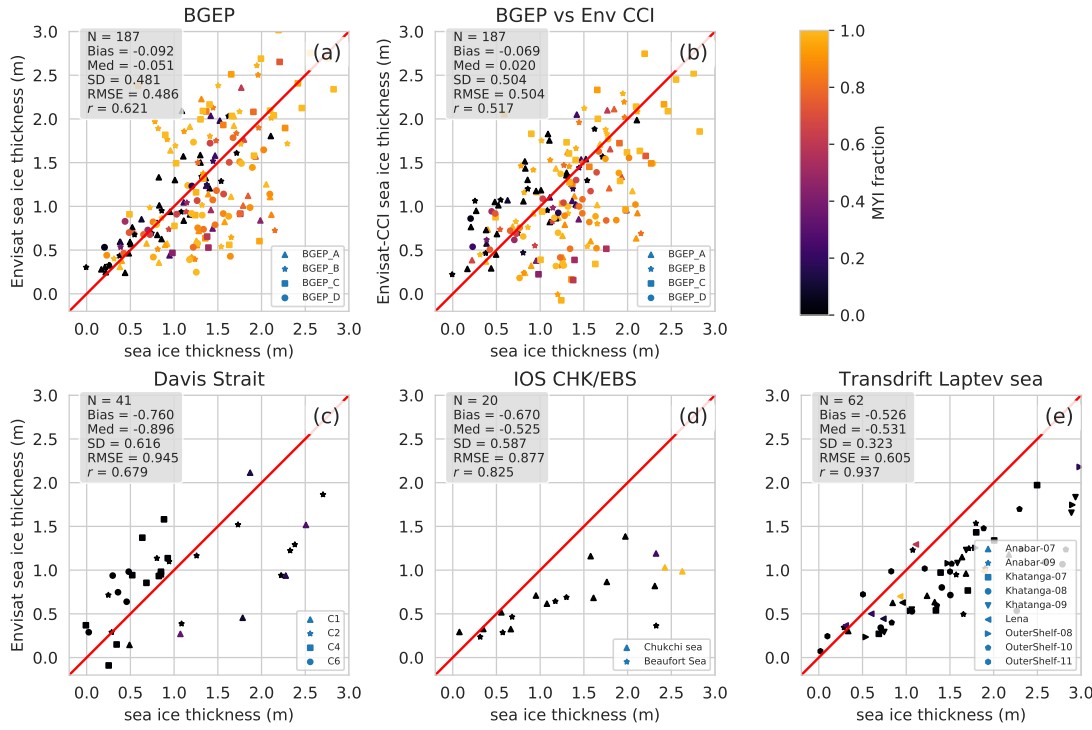

**Figure 10.** Comparative scatter-plots between Envisat NN sea ice thickness estimations and anchored moorings data sets. Each dot corresponds to a monthly averaged value. The x-axis indicates the sea ice thickness from (a) BGEP, (b) BGEP vs Env CCI, (c) Davis Strait, (d) IOS CHK/EBS and (e) Transdrift Laptev Sea ice draft. The colorbar shows the MYI proportion. N is the number of the couple of values that are compared, Med refers to the Median, SD the Standard deviation, RMSE the Root Mean Square Error and $r$ the correlation coefficient.

cm compared to 63 and 53 cm). The comparisons between Envisat and the mooring illustrated in Fig.12 are also relevant,
with correlations of 0.67 and 0.66. Similarly to Envisat comparisons with ULS data reveal non-negligible negative biases. As a comparison, the bias between CryoSat-2 and OIB between 2010 and 2019 is about 16 cm and the RMSE about 77 cm. Concerning CS-2/BGEP comparisons, the bias is 21 cm with the same overestimation of FYI thickness for CS-2 and with Transdrift Laptev Sea comparisons shows a negative bias of -38 cm.

Although draft measurements by moorings are among the most accurate measurements (they measure 90% of the total
475 thickness), they will tend to overestimate the true thickness when the ice bottom surface is rough, which is inherent to the method. Indeed, the chaotic aspect of the lower surface of sea ice can impact the ULS returned echoes. This strong deformation concerns mainly the thick and rough ice, which can explain the tendency of ULS measurements to overestimate thick ice relatively to Envisat and ERS-2. Eventually, the methods used for above comparisons can be questioned insofar as monthly averages are compared with punctual measurements (spatially and temporally), which may indeed induce biases (e.g. OIB,
AirEM, etc.). Conversions from radar freeboard to sea ice thickness can also be suspected to bias comparisons, as it depends

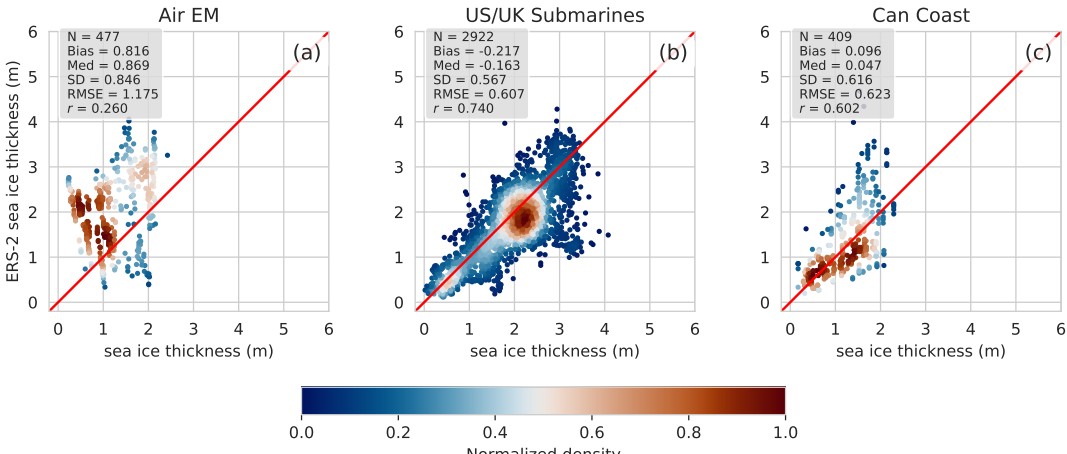

**Figure 11.** Comparative scatter-plots between ERS-2 NN sea ice thickness estimations and 3 in-situ data sets. The x-axis indicates the sea ice thickness from (a) AirEM total thickness, (b) UK/US Submarines draft and (c) Can Coast sea ice thickness. Colorbar indicates the normalized density. N is the number of the couple of values that are compared, Med refers to the Median, SD the Standard deviation, RMSE the Root Mean Square Error and $r$ the correlation coefficient.

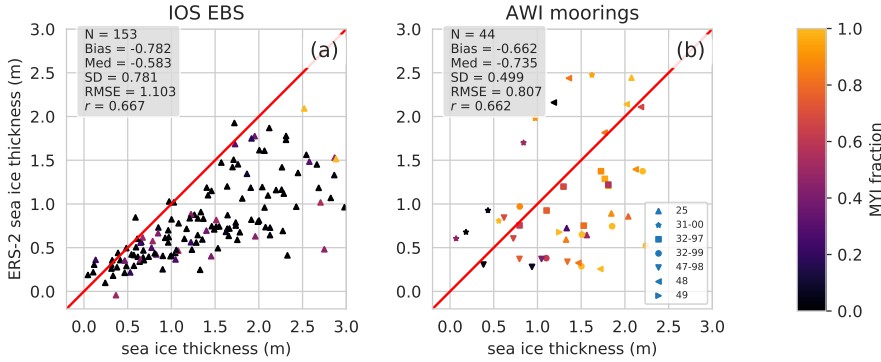

**Figure 12.** Comparative scatter-plots between ERS-2 NN sea ice thickness estimations and 2 anchored moorings data sets. The x-axis shows sea ice thickness estimations from (a) IOS Beaufort Sea and (b) AWI moorings sea ice draft. The color bar indicates the respective MYI proportion. N is the number of the couple of values that are compared, Med refers to the Median, SD the Standard deviation, RMSE the Root Mean Square Error and $r$ the correlation coefficient.

on the snow depth product. SnowModel-LG has been chosen because it is the only continuous data set at the Arctic-bassin scale product that covers more than the 27-year required in this study.

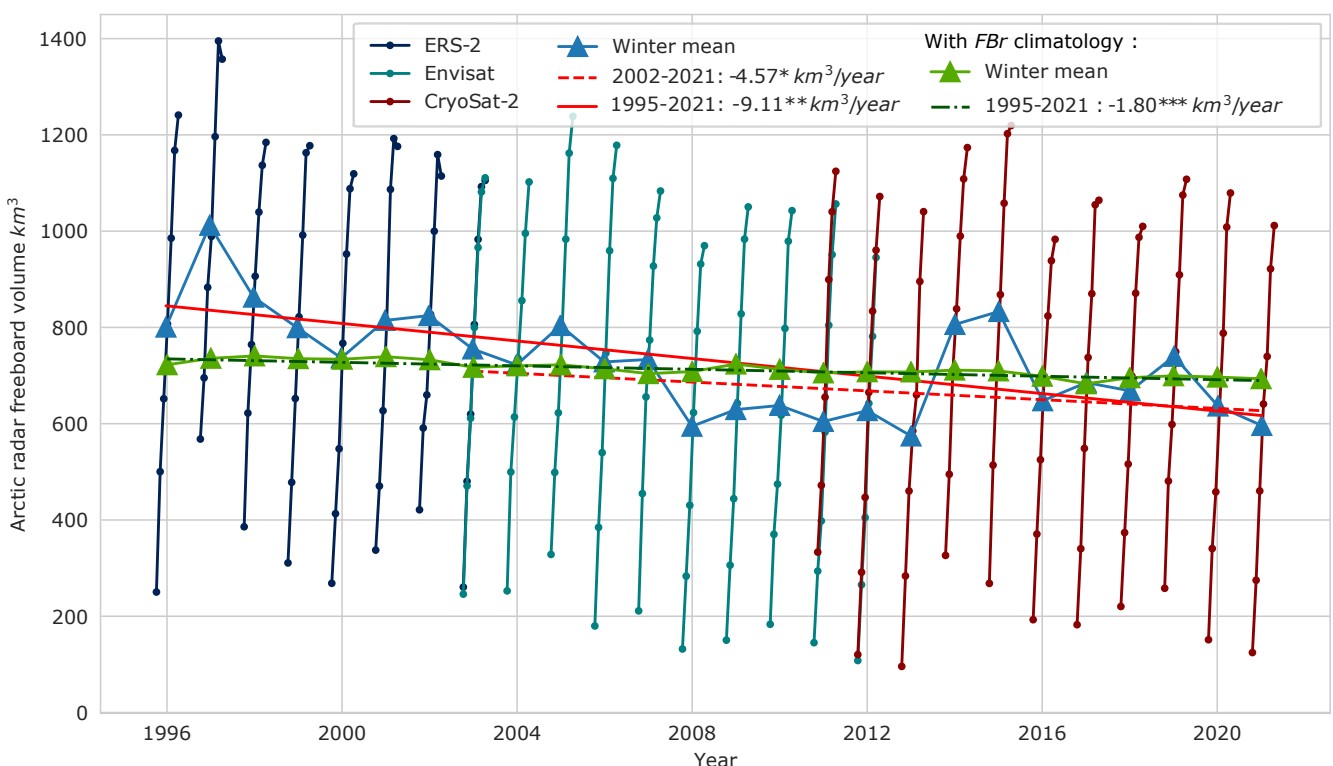

**Figure 13.** Time series representing radar freeboard volume up to 81.5°N for each winter month for ERS-2 in dark blue, Envisat in teal and CS-2 in dark red. Blue triangles are winter mean radar freeboard volumes. Red lines are linear regressions of winter mean volumes from 2002/2003 for dashed line and 1995/1996 for solid line, estimated trends are respectively $-4.57 \pm 8.73 \mathrm{km}^3/\mathrm{year}$ and $-9.11 \pm 5.16 \mathrm{km}^3/\mathrm{year}$. Green triangles represent winter mean radar freeboard volumes computed with a climatology of radar freeboard between 1995 and 2021, dash-dot green line is the regression for FBr volume with FBr climatology, the estimated trend is $-1.80 \pm 0.42 \mathrm{km}^3/\mathrm{year}$. *(1-p)<0.5, **(1-p)>0.99, ***(1-p)>0.999999, the probability value of the Mann-Kendall test.

## 4.3 Radar freeboard volume time series from 1995 to 2021

The time series represented in Fig. 13 is derived from monthly maps processed as developed in Sect. 3 and validated in Sect. 4.2.
Nevertheless, even after gridding process, missing values can occur especially where tracks density becomes low (e.g. closed to ice-ocean boundary). To ensure comparable volumes from a month to another, missing data have been replaced by interpolated values from Gauss-Seidel relaxation method implemented in pangeo-Pyinterp (https://github.com/CNES/pangeo-pyinterp), a python library developed by the CNES (French National Space Agency). It's important to note that the volumes presented in Fig. 13 are only considering value up to 81.5 °N (ERS-2 and Envisat orbit limitation).
Monthly radar freeboard volume has been finally estimated with Eq. (10), by summing radar freeboard volumes within the $P$ cells, with $S_i$, the grid cell area ($12500^2 \mathrm{m}^2$), $FB_{ri}$ and $SIC_i$, the individual radar freeboard and sea ice concentration from

NSIDC (introduced in Sect. 2.2) for each grid cell. We have decided not to convert estimations to sea ice volume to limit bias coming from snow depth estimates.

The evolution of the snow load is not taken into account in Fig. 13, which means that the evolution of the volume is not fully represented, in the same way as if the total volume were derived with a snow depth climatology. Indeed, a decrease in FBr volume may merely indicate that the snow depth is greater and the ice thickness unchanged.

$$V_{FBr} = \sum_{i=1}^{P} S_i \cdot FB_{ri} \cdot SIC_i \tag{10}$$

Three trend lines fitted on winter mean volumes are represented in Fig. 13, one is computed considering Envisat and CS-2 estimates only (dashed line), a second one from all missions estimations (solid line) and a third one with a $FBr$ climatology from 1995 to 2021 (dashed-dot line). Trends are performed using the Theil-Sen estimator (Theil, 1950; Sen, 1968) using SciPy (Virtanen et al., 2020) provided with uncertainties on the regression given with a 95% confidence interval, along with a statistical significance Mann-Kendall test (Mann, 1945; Kendall, 1948). All trends are decreasing, respectively $-4.57 \pm 8.73 \mathrm{km}^3/\mathrm{year}$, $-9.11 \pm 5.16 \mathrm{km}^3/\mathrm{year}$ and $-1.80 \pm 0.42 \mathrm{km}^3/\mathrm{year}$. The negative trend is strengthened with the integration of ERS-2 estimates and turns statistically significant. The uncertainties from regression remain high but are induced by the large inter annual variability. The trend in mean winter FBr volume obtained from a FBr climatology between 1995 and 2021 is lower but more reliable, suggesting that FBr variation over the last 30 years dominates FBr volume variations in contrast to sea ice concentration. (In comparison, the Ob river mean discharge is about $400 \mathrm{km}^3/\mathrm{year}$, if the radar freeboard volume is roughly converted to total volume with a factor 10, the Arctic sea ice decline rate (up to 81.5° N) is about a quarter of the Ob mean annual discharge).

## 5 Conclusions

This study presents a methodology to recover the radar freeboard by first correcting ERS-2 heights from the pulse-blurring effect and then an adjustment of radar freeboard over the Envisat and CryoSat-2 missions. The pulse-blurring effect correction is based on Peacock (1998) and Peacock and Laxon (2004) approach. The adjustment function developed in this study relies on a multi-layer perceptron, which is trained during the common flight period between ERS-2 and Envisat missions, using the Envisat radar freeboard as a reference. To ensure consistency along the three altimeters, Envisat radar freeboard has been preliminarily corrected against CryoSat-2 using the same neural network. The choice of a fixed-threshold retracker TFMRA to process the waveforms was motivated for continuity purpose, as it can be used for all radar altimetry missions. The methodology to estimate ERS-2 and Envisat radar freeboard is based on CryoSat-2 TFRMA50 radar freeboard, which is assumed to be the reference in the study, this hypothesis can be balanced regarding the progresses on physical retrackers. The final NN FBr does not conventionally result from a difference of two retracked heights, but corresponds to a TFMRA50 SAR-like radar freeboard corrected by a neural network.

The uncertainty estimation is initially tackled, referring to previous studies of Ricker et al. (2014) and Landy et al. (2020). These uncertainties are then propagated through the neural network thanks to a Monte Carlo approach. Uncertainties of LRM-calibrated radar freeboard run from a few millimeters up to about 15 cm depending on the ice type and the density of the along-track measurements in the grid cell. One of the limitations of the uncertainty budget is the potential underestimation of the impact of radar penetration within the snow layer, which could lead to an underestimation of the radar freeboard uncertainty.

Envisat corrected radar freeboards show good consistency with CryoSat-2 estimation, with a mean bias of 3 mm for both common winters and a SD of 9.8 cm. The Envisat radar freeboard was then compared to a large sample of validation data. 9 of the 10 data sets give consistent results, especially the strong correlation with the moorings (0.63 to 0.94) and CanCoast stations. Apart with CanCoast, these results are nearly systematically negatively biased, suggesting an underestimation of the radar freeboard. A part of the latter bias is probably due to the draft measurement's method, suggested by the increase of the bias depending on the thickness of the ice. In any case, these biases remain within the range of estimated uncertainties. The result is also consistent with the solution proposed by SI-CCI. Even if the main purpose of this study is to extend the radar freeboard time series to ERS-2, it is nevertheless fundamental to ensure the reliability of Envisat as a reference.

ERS-2 corrected radar freeboard is close to Envisat corrected ones with a correlation of 0.88, a bias of 2 mm and 3.8 cm of standard deviation for the difference. Comparisons with the few sets of in-situ data reveal the same positive bias with Can-Coast as for Envisat, and high negative biases for submerged draft measurements. Except for Air EM, the comparisons provide consistent correlation values between 0.55 and 0.74. Indeed, these statistics are those expected for comparisons between measurements from different technologies (airborne laser, ULS moorings, etc.), recording different physical quantities (draft, radar freeboard) with different spatial and temporal availability (point, monthly averages). Unless the comparison methodology is reconsidered, it seems difficult to obtain better correlations.

This work finally allows reconstructing 27 years of Arctic radar freeboard up to 81.5 °N and suggests a decline in sea ice radar freeboard volume of $9.11 \pm 5.16 \mathrm{km}^3/\mathrm{year}$ (about an order of magnitude more for total volume). This decline is significantly greater than considering only Envisat/CryoSat-2 period. Radar freeboard variations have a predominant influence on volume variations but also on the trend, in contrast to sea ice concentration which seems to have a moderate impact. In the near future, the methodology will be extended to ERS-1 mission as well as for Austral sea ice to recover 30 years of sea ice volume variation for both hemispheres. This extended data will also be freely available to the community at large. This radar freeboard time series product based on CryoSat-2 estimations intends to provide a record of monthly sea ice changes over the last 3 decades and for climate studies.

*Data availability.* CryoSat-2, Envisat and ERS-2 radar freeboard data sets produced in this study (Bocquet and Fleury, 2023) are avaiblable at : https://doi.org/10.5281/zenodo.7712503. ERS-2 RA GDR L1b product from ESA Reaper project (Brockley et al., 2017) is available at https://earth.esa.int/eogateway/catalog/ers-1-2-radar-altimeter-reaper-sensor-geophysical-data-record-sgdr-ers_alt_2s-. Envisat RA-2 L1b v3 from ESA is available at https:80//doi.org/10.5270/EN1-ajb696a. CryoSat-2 baseline-D L1b product is available at https://doi.org/10.5270/CR2-2cnblvi for SAR mode and https://doi.org/10.5270/CR2-u3805kw for SARin mode. Sea ice concentration from NSIDC

0051 https://doi.org/10.5067/8GQ8LZQVL0VL (Cavalieri et al., 1996), Sea ice age from NSIDC 0611 available at https://doi.org/10.5067/UTAV7490FEPB (Tschudi et al., 2019). SnowModel-LG snow depth and snow density https://nsidc.org/data/nsidc-0051/versions/2 (Liston et al., 2020a). Data taken from Unified Sea Ice Thickness Climate Data Record (OIB, AirEM, Davis Strait sea ice draft, IOS-CHK.) are available at https://doi.org/10.7265/N5D50JXV (Lindsay and Schweiger, 2013). BGEP data set available at http://www.whoi.edu/beaufortgyre (Krishfield et al., 2014). Transdrift Laptev sea data set available at (https://doi.pangaea.de/10.1594/PANGAEA.912927

(Belter et al., 2020). IOS-EBS available at https://doi.org/10.7265/N58913S (Melling, 2008). AWI moorings drafts are available at https://doi.org/10.7265/N5G15XSR (Witte, 2005). Submarine Upward Looking Sonar Ice Draft Profile Data and Statistics are available at https://doi.org/10.7265/N54Q7RWK. CanCoast data set is available on Environment and Climate Change Canada website https://www.canada.ca/en/environment-climate-change/services/ice-forecasts-observations/latest-conditions/archive-overview/thickness-data.html. Envisat product from LEGOS presented in Guerreiro et al. (2017) is available at 10.6096/CTOH_SIT_NH_ENV_2017_01. Envisat radar freeboard produced in the

framework of CCI is available at http://dx.doi.org/10.5285/f4c34f4f0f1d4d0da06d771f6972f180 (Hendricks et al., 2018). ICESat-1 NASA-Goddard data set is available at https://doi.org/10.5067/SXJVJ3A2XIZT (Yi and Zwally, 2009). Last visited December 2021 for all data sets

## Appendix A: Appendices

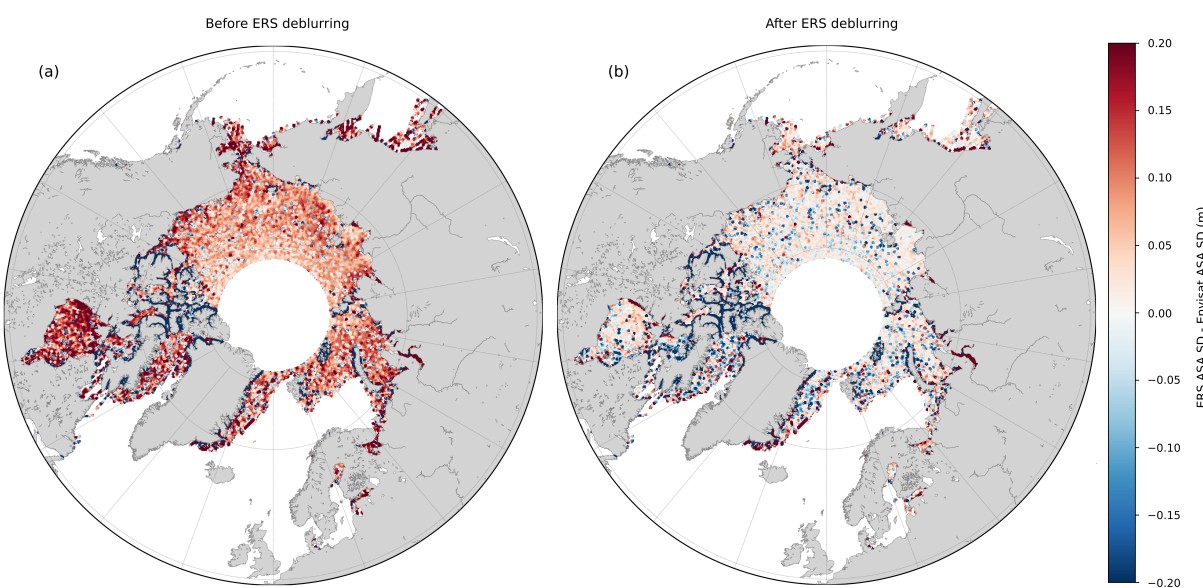

**Figure A1.** Comparison of the standard deviation of ASA between Envisat (cycle 12) and ERS (cycle 80) before (a) and after blurring correction (b) within each grid cell of a 12.5km resolution grid. Median of the standard deviation difference for (a) is 7.5cm and 0.80 cm for (b).

**Table A1.** Pulse peakiness thresholds for lead/floe classification

| Mission (RA mode) | PP lead threshold | PP floe threshold |
|---|---|---|
| CryoSat-2 (SAR) | 0.3* | 0.1* |
| Envisat (LRM) | 0.3* | 0.1* |
| ERS-2 (LRM) | 0.2839 | 0.1328 |

* Guerreiro et al. (2017)

**Table A2.** Radar Altimeter characteristics with 'Nb of echoes' the approximate number of individual echoes that have been summed up to deliver the 20hz or the 18hz waveforms, $\sigma_{1b}$ is the estimated error on range from speckle noise

| Mission (RA mode) | PRF | Data frequency | Nb of echoes | $\sigma_{1b}$ |
|---|---|---|---|---|
| CryoSat-2 (LRM) | 1.97 kHz | 20 Hz | 94 | **7** cm* |
| Envisat (LRM) | 1.80 kHz | 18 Hz | 100 | 6.8 cm |
| ERS-2 (LRM) | 1.02 kHz | 20 Hz | 50 | 9.6 cm |

* Wingham et al. (2006)

**Table A3.** Neural Network selected hyperparameters, other values was set by default in MLP regression function from Scikit-learn (Pedregosa et al., 2011)

| Hyperparameters | Value |
|---|---|
| Solver | ADAM |
| Activation | Sigmoid |
| L2 regularization alpha | $1 \cdot 10^{-6}$ |
| batch size | 1024 |
| Learning rate | $1 \cdot 10^{-3}$ |
| Number of iteration | 223 |

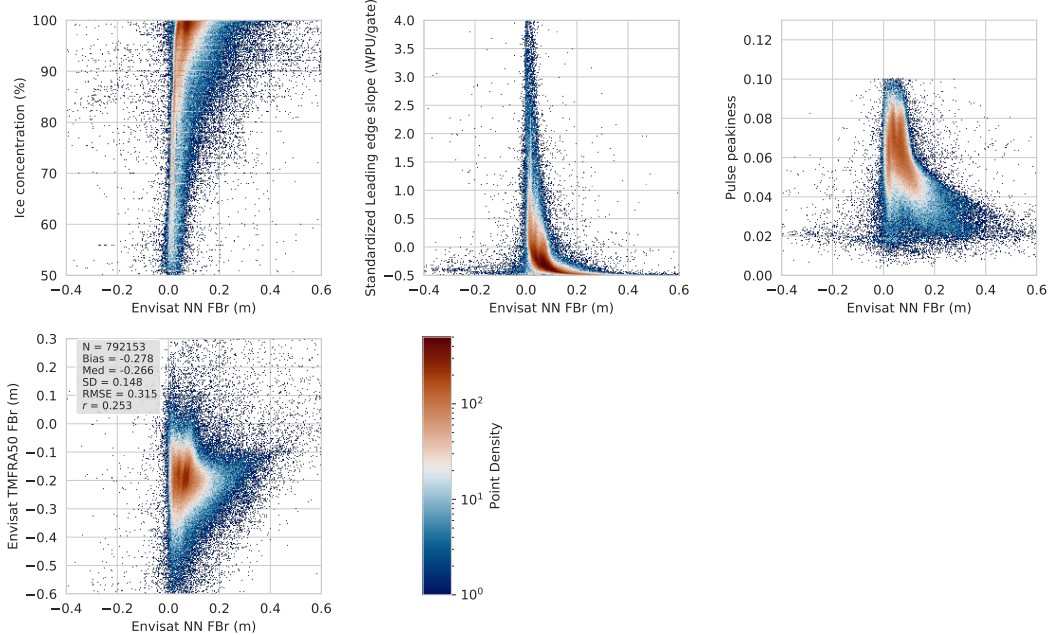

**Figure A2.** An outline of the link established by the NN between some of the inputs (Standardized LES, PP, sea ice concentration and TFMRA50 FBr) and Envisat NN FBr. WPU means Waveform power unit, for Envisat correction.

**Table A4.** Monthly Statistics (average and standard deviation) on radar freeboard for each winter month from the overlap period between Envisat and CryoSat-2 and averaged for both winters (2010-2012). SD corresponds to Standard Deviation.

| | Month | FBr mean (m) | | FBr median (m) | | FBr SD (m) | | Correlation | FBr unc mean (m) | | FBr unc median (m) | |
|---|---|---|---|---|---|---|---|---|---|---|---|---|
| | | Env | CS2 | Env | CS2 | Env | CS2 | Env/CS2 | Env | CS2 | Env | CS2 |
| — Winter 2010/2011 — | Nov | 0.050 | 0.052 | 0.036 | 0.037 | 0.050 | 0.096 | 0.521 | 0.125 | 0.025 | 0.057 | 0.017 |
| | Dec | 0.052 | 0.057 | 0.041 | 0.048 | 0.050 | 0.102 | 0.447 | 0.101 | 0.025 | 0.051 | 0.016 |
| | Jan | 0.065 | 0.068 | 0.059 | 0.063 | 0.047 | 0.100 | 0.451 | 0.127 | 0.027 | 0.057 | 0.019 |
| | Feb | 0.084 | 0.091 | 0.073 | 0.079 | 0.062 | 0.121 | 0.522 | 0.150 | 0.031 | 0.068 | 0.022 |
| | Mar | 0.098 | 0.103 | 0.086 | 0.089 | 0.076 | 0.129 | 0.581 | 0.160 | 0.074 | 0.033 | 0.024 |
| | Apr | 0.112 | 0.116 | 0.096 | 0.097 | 0.088 | 0.135 | 0.643 | 0.158 | 0.035 | 0.080 | 0.025 |
| — Winter 2011/2012 — | Oct | 0.043 | 0.043 | 0.036 | 0.033 | 0.042 | 0.115 | 0.365 | 0.185 | 0.028 | 0.069 | 0.019 |
| | Nov | 0.045 | 0.048 | 0.038 | 0.036 | 0.040 | 0.096 | 0.400 | 0.142 | 0.024 | 0.057 | 0.018 |
| | Dec | 0.051 | 0.052 | 0.042 | 0.044 | 0.054 | 0.111 | 0.510 | 0.114 | 0.025 | 0.050 | 0.017 |
| | Jan | 0.066 | 0.067 | 0.061 | 0.058 | 0.068 | 0.114 | 0.592 | 0.156 | 0.028 | 0.054 | 0.019 |
| | Feb | 0.080 | 0.081 | 0.071 | 0.071 | 0.076 | 0.125 | 0.596 | 0.155 | 0.030 | 0.068 | 0.021 |
| | Mar | 0.091 | 0.087 | 0.077 | 0.073 | 0.090 | 0.135 | 0.657 | 0.160 | 0.031 | 0.071 | 0.021 |
| | 2010-2012 | 0.074 | 0.077 | 0.063 | 0.062 | 0.071 | 0.120 | 0.583 | 0.144 | 0.029 | 0.063 | 0.020 |

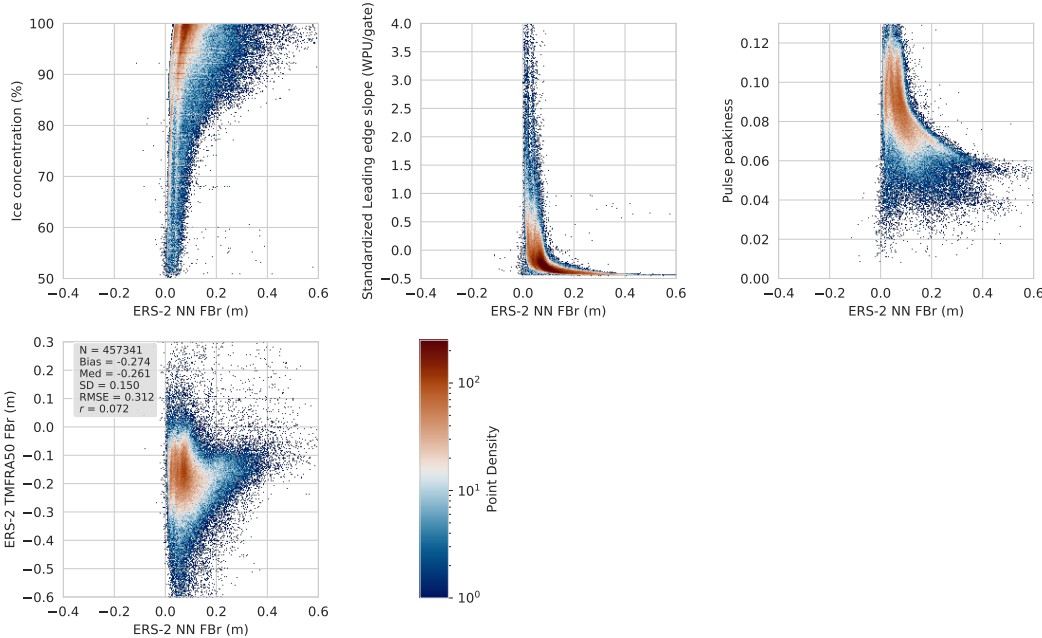

**Figure A3.** An outline of the link established by the NN between some of the inputs (Standardized LES, PP, sea ice concentration and TFMRA50 FBr) and ERS-2 NN FBr. WPU means Waveform power unit, for ERS-2 correction

**Table A5.** Statistics (average and standard deviation) on radar freeboard for each winter month from the overlap period between ERS-2 and Envisat and averaged for the whole winter (2002-2003). SD corresponds to Standard Deviation.

|  | Month | FBr mean (m) | | FBr median (m) | | FBr SD (m) | | Correlation | FBr unc mean (m) | | FBr unc median (m) | |
|---|---|---|---|---|---|---|---|---|---|---|---|---|
|  |  | ERS-2 | Env | ERS-2 | Env | ERS-2 | Env | ERS-2/Env | ERS-2 | Env | ERS-2 | Env |
| Winter 2002/2003 | Oct | 0.061 | 0.058 | 0.049 | 0.043 | 0.044 | 0.048 | 0.881 | 0.176 | 0.152 | 0.087 | 0.066 |
|  | Nov | 0.074 | 0.073 | 0.048 | 0.044 | 0.064 | 0.070 | 0.937 | 0.145 | 0.134 | 0.075 | 0.062 |
|  | Dec | 0.076 | 0.074 | 0.050 | 0.048 | 0.067 | 0.074 | 0.900 | 0.146 | 0.115 | 0.075 | 0.064 |
|  | Jan | 0.083 | 0.082 | 0.065 | 0.065 | 0.056 | 0.068 | 0.862 | 0.164 | 0.148 | 0.078 | 0.071 |
|  | Feb | 0.095 | 0.093 | 0.075 | 0.074 | 0.068 | 0.074 | 0.895 | 0.172 | 0.156 | 0.082 | 0.076 |
|  | Mar | 0.105 | 0.103 | 0.085 | 0.084 | 0.074 | 0.092 | 0.834 | 0.167 | 0.168 | 0.087 | 0.081 |
|  | Apr | 0.112 | 0.112 | 0.094 | 0.096 | 0.076 | 0.088 | 0.845 | 0.168 | 0.157 | 0.093 | 0.087 |
|  | 2002-2003 | 0.090 | 0.088 | 0.073 | 0.071 | 0.068 | 0.079 | 0.876 | 0.164 | 0.148 | 0.082 | 0.073 |

**Table A6.** Statistics of sea ice thickness difference between Envisat and each validation data set.

| | Campaign | Bias (m) | Median (m) | Standard Deviation (m) | RMSE (m) | Correlation |
|---|---|---|---|---|---|---|
| FBr | Envisat CCI | -0.007 | -0.007 | 0.052 | 0.053 | 0.712 |
| | Envisat PP | 0.035 | 0.030 | 0.082 | 0.089 | 0.553 |
| Sea Ice Thickness | ICESat-1 | -0.197 | -0.116 | 0.906 | 0.928 | 0.497 |
| | OIB | -0.352 | -0.249 | 1.018 | 1.077 | 0.332 |
| | Air EM | -0.288 | -0.517 | 0.991 | 1.032 | 0.047 |
| | Can Coast | 0.089 | -0.001 | 0.632 | 0.637 | 0.406 |
| | UK and US submarines | -0.391 | -0.280 | 0.532 | 0.659 | 0.548 |
| | BGEP | -0.092 | -0.051 | 0.481 | 0.486 | 0.621 |
| | Davis Strait | -0.760 | -0.896 | 0.616 | 0.945 | 0.679 |
| | IOS Chukchi and Beaufort sea | -0.670 | -0.525 | 0.587 | 0.877 | 0.825 |
| | Transdrift Laptev sea | -0.526 | -0.531 | 0.323 | 0.605 | 0.937 |

**Table A7.** Statistics of sea ice thickness difference between ERS-2 and each validation data set.

| | Campaign | Bias (m) | Median (m) | Standard Deviation (m) | RMSE (m) | Correlation |
|---|---|---|---|---|---|---|
| Sea Ice Thickness | Air EM | 0.816 | 0.869 | 0.846 | 1.175 | 0.260 |
| | UK and US submarines | -0.217 | -0.163 | 0.567 | 0.607 | 0.740 |
| | Can Coast | 0.096 | 0.047 | 0.616 | 0.623 | 0.602 |
| | IOS Beaufort sea | -0.782 | -0.583 | 0.781 | 1.103 | 0.677 |
| | AWI moorings | -0.662 | -0.735 | 0.499 | 0.807 | 0.662 |

## Appendix B: List of Acronyms

**Table B1.** List of Acronyms

| ACRONYMS | Description |
|---|---|
| ASA | All Surface Anomaly |
| AGC | Automatic Gain Control |
| ASD | Altimetric Snow Depth |
| CHK | Chukchi |
| CS-2 | CryoSat-2 |
| EBS | Eastern Beaufort Sea |
| Env | Envisat |
| ERS | European Remote-Sensing Satellite |
| FB | Sea ice Freeboard |
| FBr | Radar Freeboard |
| FBt | Total Freeboard |
| FYI | First-year Ice |
| HTL | Height Tracking Loop |
| ILA | Ice Level Anomaly |
| LRM | Low Resolution Mode |
| MSS | Mean Sea Surface |
| MYI | Multiyear Ice |
| MLP | Multi-layer Perceptron |
| NN | Neural Network |
| PRF | Pulse Repetition Frequency |
| RA | Radar Altimeter |
| RMSE | Root Mean Squared Error |
| SAR | Synthetic Aperture Radar |
| SARM | Synthetic Aperture Radar Mode |
| SARIn | Synthetic Aperture Radar Interferometric |
| SD | Standard Deviation |
| SIT | Sea Ice Thickness |
| SLA | Sea Level Anomaly |
| STL | Slope Tracking Loop |
| TFMRA | Threshold First-Maximum Retracker Algorithm |
| ULS | Upward Looking Sonar |

*Author contributions.* The methodology of this paper was performed by M.B. and supervised by S.F. . M.B. processed the data and make the validation. M.B. and S.F. wrote the paper. All authors have participated in the present article and brought contributions to the elaboration of its final version.

*Competing interests.* The authors declare that they have no conflict of interest.

*Acknowledgements.* This research was supported by the CNES and CLS thanks to a doctoral allowance granted to Marion Bocquet. This study is supported by ESA in the framework of the FDR4ALT project. It has also benefited from the support of the CNES TOSCA CASSIS project. The Scientific colour map "roma" and "vik" (Crameri, 2021) are used in this study to prevent visual distortion of the data and exclusion of readers with colourvision deficiencies (Crameri et al., 2020).

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
