# Peer review of "Arctic sea ice radar freeboard retrieval from ERS-2 using altimetry: Toward sea ice thickness observation from 1995 to 2021"

_EGUsphere, 2022_

## Referee Comment (RC2)

I left a community comment on this manuscript (https://doi.org/10.5194/egusphere-2022-214-CC1) before being nominated as a referee. I have therefore read and considered the manuscript again.

As part of this, I investigated the data that was made available to me as a nominated reviewer. I wanted to see the size of the correction/calibration applied by the neural network presented in this paper. This has led me to question the nature of the 'correction' being applied, and whether it is reasonable to present this data product as a series of 'corrected' radar freeboard values at all. I would like to review this manuscript again once the queries raised here have been addressed.

Firstly, it's possible that I have misunderstood how to read the data here, or made a mistake in my analysis. I have used the netcdf 'radar_freeboard_corr' variable as the 'corrected' Rfbs, although I did also run the same code on the 'corr_median' and 'corr_mean' and results were very similar. Please do let me know if I'm mistaken in any way. I've uploaded my code here: https://github.com/robbiemallett/ERS2_rev

In this paper the authors present a method of `correcting' the radar freeboard values generated by the ERS2 satellite. To do this, the authors retrack raw ERS2 waveforms using the TFMRA50 algorithm to generate a radar freeboard value. They then apply a neural network to the retracked heights, which also assimilates sea ice age and concentration data alongside other waveform statistics like pulse peakiness. The neural network takes in all this data and returns a 'corrected radar freeboard', which is theoretically consistent with other missions.

I began by mapping the raw and corrected ERS2 radar freeboards for each month, as presented in the data. It immediately struck me that the 56 pairs of maps did not look visually similar. That is to say, it was not clear that areas with larger TFMRA50 radar freeboards ended up as areas of higher corrected radar freeboards.

I then took a quantitative approach; For each year/month of data, I converted the raw radar freeboard values to anomalies from the month's mean, and did the same for the corrected values. I then performed a linear correlation between the two data sets for each month. This produced a table with 56 rows (one for each year/month combination from 1995 – 2003), here's the first few rows:

| date | r_val_rfb | r_val_age |
|---|---|---|
| 1995-10-16 | 0.056351 | 0.330371 |
| 1995-11-16 | -0.043177 | 0.461461 |
| 1995-12-16 | -0.070674 | 0.428449 |
| 1996-01-16 | -0.007595 | 0.377684 |
| 1996-02-15 | 0.078294 | 0.375440 |
| 1996-03-16 | 0.153395 | 0.382188 |
| 1996-04-16 | 0.209462 | 0.404116 |
| 1996-10-26 | 0.197490 | 0.392116 |
| 1996-11-16 | 0.219052 | 0.384350 |

*Illustration 1: First few rows of correlations between corrected RFb anomaly and TFMRA50 RFb (col 2) and sea ice age (col 1)*

The correlations between raw and corrected freeboard anomalies are very low, the mean value is r=0.07. I then performed the same analysis, but instead correlated the `corrected' radar freeboard anomalies with the sea ice age data (reconstructed per the manuscript). I set the values of MYI to 1, and FYI to 0. The mean r-values for this are 0.33. This means that the sea ice age data is a much stronger determinant of the corrected radar freeboard value than the raw radar freeboard value itself.

I then calculated the correlation between the size of the correction and the corrected anomaly itself. The mean r-value for this is 0.49. This means that the correction itself is a strong determinant of the end product, by comparison to the raw, retracked radar freeboard value itself.

This is all a bit concerning. If my analysis is correct, it seems like the presented neural network is really modelling the radar freeboard based on the sea ice age and other assimilated variables (e.g. PP), and more or less ignoring the raw retracked elevations. I therefore don't think it's reasonable to describe the neural network as applying a `correction' or 'calibration' to the raw values, since the resulting values are so unrelated.

To illustrate how the sea ice age is a stronger determinant of the corrected Rfb than the raw Rfb, I've plotted the r-values for each month as a histogram below. As you can see, the r-values between corrected Rfb and age are much larger than the corresponding corrected vs raw values.

[Figure]

*Illustration 2: Correlations between age and corrected radar freeboard (blue) and correlations between raw radar freeboard and corrected radar freeboard (orange).*

To further investigate, I took all the years of a given month and plotted the raw anomalies against the corrected anomalies. Again, there's no strong relationship. My major concern is therefore this: to what extent can you claim that the resulting product is a corrected or calibrated retrieval when it doesn't reflect the variability in the raw, retracked values?

[Figure]

*Illustration 3: Relationships between the monthly raw radar freeboard anomaly and the corrected radar freeboard anomaly. It seems that ice with a higher-than-average raw radar freeboard in a given month does not end up with a higher-than-average corrected radar freeboard value.*

In my above analysis I have considered anomalies – for each month in the netcdf files I calculated anomalies of the raw and corrected radar freeboards, such that I could investigate whether higher radar freeboards in a given month translated into higher corrected values.

I now just consider the absolute values of all 3,054,312 valid measurements presented in the data. To do so I took the raw and corrected RFb value of every grid cell in every month and differenced them to get the size of the correction. Doing this shows there is still essentially no relationship between the raw (input) Rfb and the corrected (output) Rfb, the r-value is 0.094 (not shown). This again is concerning – it implies to me that larger measured radar freeboards are simply not translating to larger corrected freeboards. If this isn't happening then what is the point of measuring the radar freeboard? The end RFb value seems to be entirely dictated by variables other than the raw Rfb. Once again, the r-value between age and corrected Rfb is much larger (0.33).

The result of all this is that there's a highly linear, negative relationship between the raw radar freeboard and the correction size (r=-0.82). I fear the neural network has learned to shift the input distribution and then scale down high input values and scale up low values such that destroys any information encoded in the raw Rfbs.

Along these lines, it is telling that the raw radar freeboards are weakly correlated with sea ice age, in fact the r-value is negative (r = -0.087). This suggests to me that the raw radar freeboards do not currently represent a geophysical signal that can be 'corrected'. Instead, the signal:noise ratio from TFMRA processed ERS2 waveforms may just be too low to be useful.

[Figure]

*Illustration 4: Left: input, correction, and output. Right: the highly linear nature of the correction that's applied. Seems to squash all signal out of the TFMRA50 RFbs.*

**Summary**

Based on the above analysis, it appears that almost nothing of original radar freeboard measurement remains in the 'corrected value'. If this is the case, I think the method in this paper needs to be fundamentally revisited, as the retracked elevations from ERS2 **must** have an impact on the final product if it is to be honestly presented as a "radar freeboard product". If the authors feel that a product based primarily on different data (e.g. sea ice age or PP) is more appropriate, then this should be clearly presented and the product renamed to unambiguously indicate this.

I have a couple more comments, but we should focus on the above first. But just as as heads up:

- L280: I think you should by convention use the coefficient of determination rather than Pearson-r as a test score. Otherwise you'll end up with highly correlated relationships that have the wrong slope?

- You need to explain quite a lot more about what's going on in Figure 6. The manuscript should not feature undefined letters and symbols, and there are many in this figure.

- Similar to above, you should explain much more about what's going on between lines 277 & 285. Papers in The Cryosphere should be accessible to scientists without extensive experience in machine learning. Don't be afraid to use the supplement for this, as I appreciate it's wordy. For instance, why did you choose 5 hidden layers and 100 neurons, and what are the implications of your choice? Why a sigmoid? There are noticeably no references to support your choices, and there's no element of later discussion about the impacts.

- I also have the view that `radar freeboard' is not a geophysical quantity to be measured with an uncertainty. Instead it is precisely the retracked elevation of a waveform returning from sea ice, and is specific to a given radar's geometry and the chosen retracking algorithm. See the original definition in the supplement of Armitage & Ridout 2015, and Tilling et al. 2019 for how different radars will generate different Rfbs even if they could 'look at' the same ice. Similarly, different retrackers will generate different Rfbs when `looking' at the same waveform, all of them valid and precise.

  So I think you should change the phrase 'radar freeboard correction' to 'radar freeboard calibration', as you're not correcting some uncertain value. Instead you're calibrating the Rfb from one instrument so that it's consistent with another instrumental geometry. The same with `radar freeboard estimation' - you're not estimating it: it's a precise value resulting from the radar geometry and choice of retracker. I have a lot more to add on this issue, but it's quite philosophical/ subjective and I think we need to first focus on the issue concerning the representation of the TFMRA50 Rfbs in the `corrected' product.

- One additional concern is that Garnier et al. (2022) uses the same neural network approach (citing this discussion paper). If I am correct in my analysis above, I think the authors should be prepared to re-analyse the data and show that their Envisat TFMRA50 RFb values have a practical impact on their calibrated RFb values. I think this situation may be better, as the calibration from Envisat → CS2 is more direct than for ERS2 → EnviSat → CS2.

---

## Community Comment (CC1)

I was really pleased to see this paper come up in TCD. I think that generating radar freeboard data from ERS1/2 is one of the most pressing tasks for the sea ice community, and so I agree with Jack Landy's review. Overall I think the paper is well written and addresses what is a very significant gap in our knowledge of the Arctic Ocean. In particular I think the figures are well designed.

I do have a couple of concerns, questions and suggestions over wordings, citations etc. I hope the authors will take these in the spirit of discussion, rather than as negative criticism. I really do think that this research is high-quality and useful.

L25) I would question whether "thin ice is more sensitive to climatic hazards". Bitz and Roe (2004) argued the opposite: that thick ice is thinning faster, because areas of thin ice grow more quickly in winter. Age products also show that thicker, older ice is dissappearing from the Arctic and being replaced by thinner, seasonal ice (e.g. Nghiem et al., 2007). So I'm not sure it makes sense to say that thin ice is more sensitive to climatic hazards, when thin ice is coming to dominate the Arctic and is more robust to temperature perturbations.

L32) I don't agree that it's "commonly accepted" that Ku-band radar waves penetrate the snow layer when it is sufficiently cold. I think that assumption is still up for discussion, and I would argue the opposite. I'm not aware of any in-situ or airborne CryoSat evaluation ever done over sea ice that has produced evidence that Ku-band radar waves consistently return from the snow-ice interface.

For instance, neither of the airborne CryoVex 2006 and 2008 campaigns (Willatt et al. 2011) indicated that this was consistently the case over FYI. Results from a different radar system in Antarctica (Willatt et al. 2010) also showed that radar waves do not always return from the ice surface. Results from a third radar system deployed on MOSAiC (on SYI) indicate that more Ku-band power comes back from the snow surface than from the ice surface (Stroeve et al., 2020 Fig. 7; Nandan et al., 2022 Fig. 8). Garnier et al. (2022; Figure 9) shows results from CryoVex 2017 where the difference between Ka and Ku band ranging is at times negative, further casting doubt on the assumption.

Moving to satellite-based evidence, Armitage and Ridout (2015) calculated CryoSat-2's penetration factor as 82%. Ricker et al. (2015) used buoys to show that snow accumulation caused increases in Rfb, not decreases (implying that the radar waves are not penetrating fully). This agrees with the work of Gregory et al., (2022; Figure 9) that shows that snowfall is correlated (not anti-correlated) with Rfb over both ice types.

I would also argue that the often-cited work of Beaven et al. (1995) was not realistic – it featured snow that was shovelled, sifted through a screen, and then artificially smoothed at the surface by the weight of a metal plate before measurement. It is also striking that what the authors identify as the snow-ice interface appears at 20 cm range when it was 21 cm away in free space. Since it was 21 cm away in free space it should have appeared further away, at something like 25 cm in range due to the wave-propagation delay.

There's no need to mention all this in your paper, but I wanted to briefly state my evidence before making the point that *full Ku-band penetration* is not a settled consensus, even for cold, dry snow. I think it would be fair to say that full penetration is "commonly assumed in satellite-based sea ice thickness products". But just because we're forced to assume it in our products doesn't mean the we should actually believe or accept the assumption.

L381: Year of this citation is 1986.

L65: I think we're not really measuring sea ice thickness, but instead estimating it based on freeboard measurements (or radar-altimetry measurements). This might seem like a semantic point, but I think users of sea ice thickness products do benefit from this distinction. "estimates" rather than "measurements" is more commonly used by convention (e.g. Tilling et al., 2018, Kurtz et al., 2014; Landy et al., 2017).

L75: I think readers like me who aren't expert in roughness would benefit from a citation here. Is LRM definitely more impacted by a given roughness than SARM? I can believe it, but would like to read some evidence.

L103: I think you mean NSIDC 0611? This product gives the maximum of the ice age distribution in a grid cell at each timestep (see quote below). So I'm not sure how you've used these max values to generate an MYI fraction product? I think it could be done if you had access to the Lagrangian data, which is out there. But if you've used this I think you should state that.

(Tschudi et al. (2020) states "This approach does not consider new ice that may form within a grid cell because it retains only the oldest ice in its accounting. Thus, the product is effectively an estimate of the oldest ice in a given grid cell.")

L115: I think at some point you should direct the reader to Kwok and Haas (2015), which discusses some key issues in the product that you've chosen.

L310: "Surface roughness is identified as the largest source of uncertainty" - I didn't really understand how you made it to this conclusion.

I think this is specifically a reference to Fig. 8 of Landy et al. (2020). The error in the sea ice roughness over FYI is 4cm, and the error from the snow basal salinity (just part of the "penetration bias") is 7 cm, and the uncertainty due to snow depth is 6 cm. So over FYI the roughness uncertainty is smaller than either the snow depth or the snow salinity. As such I don't think roughness can be reasonably characterised as "the largest source of uncertainty" over FYI based on Landy et al. 2020 Fig. 8. Over MYI the sea ice roughness uncertainty is equal to the snow depth uncertainty, and admittedly larger than "partial snow penetration" uncertainty. So the statement is narrowly true if you only consider MYI and don't factor in the (highly related) uncertainty in snow depth in the comparison.

But I think that only considering the largest source of uncertainty and ignoring the other uncertainties is a pretty risky strategy, given the other sources are comparable and perhaps actually larger in magnitude? If you are wedded to this approach, I think you should state that this will induce a pretty serious underestimate in your uncertainty values (which is important info for product end-users).

Fig. 6: I see in the top panel that you've "summed the squares", which has the implicit assumption that uncertainties that you have considered are un-correlated. It may be that you have good evidence to support this that I'm ignorant of, but it seems, for instance, that speckle noise may well be (anti?)correlated with surface roughness? Just as an example. I think that the omitted snow uncertainties involving penetration & depth are more likely than not to be correlated in some way. I think you should state that you've assumed the uncertainties are uncorrelated in your analysis, and give the reader some information as to what the results of that assumption may be.

Figs. 7 & 8: These are really well designed and presented

L381: 4) Why take snow density as constant? SnowModel-LG outputs depth and density, and includes some physics of densification/settling over time. So I think it's odd to use one of its variables and not the other, since they're so linked in the model. Snow impacts thickness retrievals by weighing the floe down and slowing radar waves: both of these effects are proportional to the mass of overlying snow – not the depth (see Mallett et al., 2021). So I think it makes a lot more sense to use both the depth and density (the SWE) in your thickness retrievals rather than just the depth. Here's a plot of the seasonal densification of SnowModel-LG snow north of 88N for the period 1995-2018. You'll see that as well as being more dense than your assumption, it also evolves over the season.

[Figure]

Fig 13: I'm a little unclear what the radar freeboard timeseries is supposed to represent. I imagine it mostly reflects the trend and variability in sea ice extent, and I think you should point this out to the reader. A simple correlation with SIE would quantify this relationship and reveal if the quantity is useful. For the part of it that doesn't represent SIE, would decreasing volume reflect a thinning of sea ice? Thinning snow (Webster et al., 2014) will mask the effect of thinning ice on the Rfb. In areas where the snow is really thinning quickly, the Rfb could potentially even increase even if the ice is thinning. I guess I would like to see a little interpretation of this quantity & figure 13 rather than being left to do it as the reader.

L450: I think you should state the limitations in your uncertainties here. In particular (and I think this is key), do the "observed" thicknesses fall within your uncertainty bounds? If not, then either your uncertainty bounds are wrong or the validation data is wrong. I think uncertainty bounds on retrievals are not useful unless you can show that observed data fall within them.

Data availability: I was disappointed to see that the data were not made available given the community-focus of The Cryosphere. Also, because the code is not available, the work is not reproducable or checkable by the nominated reviewers. The data policy of TC is clear that:

"If the data are not publicly accessible, a detailed explanation of why this is the case is required".

The context of this is also given by the data policy:

"The output of research is not only journal articles but also data sets, model code, samples, etc. Only the entire network of interconnected information can guarantee integrity, transparency, reuse, and reproducibility of scientific findings. Moreover, all of these resources provide great additional value in their own right. Hence, it is particularly important that data and other information

underpinning the research findings are "findable, accessible, interoperable, and reusable" (FAIR) not only for humans but also for machines."

I really would encourage the authors to follow the data policy, especially as it is likely to significantly increase the impact of their work.

**References**

Landy, Jack C., et al. "Sea ice roughness overlooked as a key source of uncertainty in CryoSat 2 ice freeboard retrievals." Journal of Geophysical Research: Oceans 125.5 (2020): e2019JC015820.

Tilling, Rachel L., Andy Ridout, and Andrew Shepherd. "Estimating Arctic sea ice thickness and volume using CryoSat-2 radar altimeter data." Advances in Space Research 62.6 (2018): 1203-1225.

Kwok, Ron, et al. "Intercomparison of snow depth retrievals over Arctic sea ice from radar data acquired by Operation IceBridge." The Cryosphere 11.6 (2017): 2571-2593.

Kwok, R., and C. Haas. "Effects of radar side-lobes on snow depth retrievals from Operation IceBridge." Journal of Glaciology 61.227 (2015): 576-584.

Landy, Jack C., et al. "Sea ice thickness in the Eastern Canadian Arctic: Hudson Bay Complex & Baffin Bay." Remote Sensing of Environment 200 (2017): 281-294.

Webster, Melinda A., et al. "Interdecadal changes in snow depth on Arctic sea ice." Journal of Geophysical Research: Oceans 119.8 (2014): 5395-5406.

Mallett, Robbie DC, et al. "Faster decline and higher variability in the sea ice thickness of the marginal Arctic seas when accounting for dynamic snow cover." The Cryosphere 15.5 (2021): 2429-2450.

Tschudi, Mark A., Walter N. Meier, and J. Scott Stewart. "An enhancement to sea ice motion and age products at the National Snow and Ice Data Center (NSIDC)." The Cryosphere 14.5 (2020): 1519-1536.

Armitage, Thomas WK, and Andy L. Ridout. "Arctic sea ice freeboard from AltiKa and comparison with CryoSat 2 and Operation IceBridge." Geophysical Research Letters 42.16 (2015): 6724-6731.

Willatt, Rosemary C., et al. "Field investigations of Ku-band radar penetration into snow cover on Antarctic sea ice." IEEE Transactions on Geoscience and remote sensing 48.1 (2009): 365-372.

Willatt, Rosemary, et al. "Ku-band radar penetration into snow cover on Arctic sea ice using airborne data." Annals of Glaciology 52.57 (2011): 197-205.

Ricker, Robert, et al. "Impact of snow accumulation on CryoSat 2 range retrievals over Arctic sea ice: An observational approach with buoy data." Geophysical Research Letters 42.11 (2015): 4447-4455.

Gregory, William, Isobel R. Lawrence, and Michel Tsamados. "A Bayesian approach towards daily pan-Arctic sea ice freeboard estimates from combined CryoSat-2 and Sentinel-3 satellite observations." The Cryosphere 15.6 (2021): 2857-2871.

Beaven, S. G., et al. "Laboratory measurements of radar backscatter from bare and snow-covered saline ice sheets." Remote Sensing 16.5 (1995): 851-876.

Garnier, Florent, et al. "Advances in altimetric snow depth estimates using bi-frequency SARAL and CryoSat-2 Ka–Ku measurements." The Cryosphere 15.12 (2021): 5483-5512.

Bitz, C. M., and G. H. Roe. "A mechanism for the high rate of sea ice thinning in the Arctic Ocean." Journal of Climate 17.18 (2004): 3623-3632.

Nghiem, S. V., et al. "Rapid reduction of Arctic perennial sea ice." Geophysical Research Letters 34.19 (2007).

Stroeve, Julienne, et al. "Surface-based Ku-and Ka-band polarimetric radar for sea ice studies." The Cryosphere 14.12 (2020): 4405-4426.

Nandan, Vishnu, et al. "Wind Transport of Snow Impacts Ka-and Ku-band Radar Signatures on Arctic Sea Ice." The Cryosphere Discussions (2022): 1-38.

Kurtz, Nathan T., N. Galin, and M. Studinger. "An improved CryoSat-2 sea ice freeboard retrieval algorithm through the use of waveform fitting." The Cryosphere 8.4 (2014): 1217-1237.

---

## Author Comment (AC1)

**Title: Arctic sea ice radar freeboard retrieval from ERS-2 using altimetry : Toward sea ice thickness observation from 1995 to 2021**

*Marion Bocquet, Sara Fleury, Fanny Piras, Eero Rinne, Heidi Sallila, Florent Garnier, and Frédérique Rémy*

**Jack Landy (referee n°1) - global comment**

The authors construct a 25-year record of Arctic sea ice radar freeboard by reconciling the measurements from three radar altimetry missions, one ongoing and two historic. Their primary motivation is to generate the first step towards a long-term sea ice thickness record for the Arctic Ocean. This would be the first observational sea ice thickness record spanning such a long period and would offer valuable comparison to existing proxy sea ice thickness (SIT) records based on ice age and models. In my view, a robust 25+ year time series of Arctic sea ice thickness would represent a major scientific breakthrough with implications for understanding global climate changes in the modern era and validating and improving sea ice models, among other potential applications.

Generally, I find the approach and methods to be scientifically sound. I have some minor comments but nothing that questions the rigour of the generated time series. The validation against existing SIT data from satellites, airborne and in situ sensors is comprehensive and convincing.

Excellent work on a really valuable study – it was a pleasure to read! Feel free to get in touch if you have any questions, Jack Landy

**Answer to Jack Landy (referee n°1) - global comment**

We would like to thank the reviewer for his careful reading of the manuscript, for this positive feedback and for the relevant and constructive remarks that have helped to improve the quality of the manuscript. In order to fit with your comments, we have made a revision of the manuscript that should have corrected the textual issues and well improved the readability of the document. We hope that these modifications will meet your requirements. Please find below the details on how your specific comments have been taken into account. *In this document, the referee's comments are in bold type, the answers are in italic type, and the corrections to the revised manuscript are in normal type.*

**Answers to Jack Landy (referee n°1) : specific comments**

**Line 2. Sea ice volume's..?**

*This correction has been done.*

**L14-15. I would suggest including other statistics of the variability on the bias within the abstract. Given the ML algorithm aims to remove the bias I would argue the stats on variability are more interesting for the reader.**

*These statistics have been added to the abstract. The following modification has been done :*

L 14-15: Comparisons of corrected radar freeboards during overlap periods reveal good consistencies between missions, with a mean bias of 3 mm for Envisat/CryoSat-2 and 2mm for ERS-2/Envisat.
**replaced by**
Comparisons of corrected radar freeboards during overlap periods reveal good consistencies between missions, with a mean bias of 3 mm and a standard deviation of 9.7 cm for Envisat/CryoSat-2, and respectively 2 mm and a 3.8 cm for ERS-2/Envisat.
* * *
**L28. Technically past radar altimeters have not allowed basin scale, so altimetry doesn't offer a 'global approach' over the long term. But this is nit-picky.**

*That's true, we have modified a little the sentence.*

L28. A global approach is possible through satellite altimetry, especially with radar altimetry, which is not impacted by the cloud cover and whose missions are continuous since 1991.
**replaced by**
A quasi-global approach is possible through satellite altimetry, especially with radar altimetry, which is not impacted by the cloud cover and whose missions are continuous since 1991.
* * *
**L51. Explain 'heuristic retracker TFMRA50'.**

*Indeed, the sentence was not clear. It has been corrected. The purpose of this sentence was to explain that before any calibration (LRM/SAR) all the waveforms for both missions were processed with the same algorithm and the retracker that has been used is a TFMRA retracker (which is categorized as an empirical retracker based on a heuristic approach).*

L51 : The consistency between missions is preserved by using the same processing chain regardless of the mission, with the heuristic retracker TFMRA50 (Helm et al., 2014)
**replaced by**
The consistency between missions is preserved by using the same processing chain regardless of the mission (before calibration), starting by the retracking algorithm : the empirical threshold first-maximum retracker algorithm (Helm et al., 2014) with a threshold of 50 % (TFMRA50).
* * *
**L65 : Check Appendix Table 1. Does this tally?**

*The appendix indicates the RA characteristics. The table is supposed to help the reader to understand the estimation of uncertainties from the speckle noise for Envisat and ERS-2. Wingham et al. (2006) estimates the uncertainty due to speckle noise for CryoSat-2 in SAR, SARIn Mode as well as for LRM. CryoSat-2 LRM mode data are not used (neither pLRM) in this paper, but the CS-2 LRM speckle noise error on range is used to compute ERS-2 and Envisat uncertainties that come from the speckle*

*noise (L297-302). The legend of table A1 (now A2) has been developed, and a reference has been added in the corresponding section 3.5.*

70 ───────────────────────────

**L103-104: How is it aggregated? Bit vague.**

*The following modification has been done to try to improve the clarity of the manuscript:*

L103-104 : This information comes from the NSIDC 0061 sea ice age product (Tschudi et al., 2019) that is aggregated into two classes (MYI and FYI)

75 **replaced by :**

The study also requires a sea ice type product, this information is derived from the NSIDC 0061 sea ice age product (Tschudi et al., 2019) that is aggregated into two classes (MYI and FYI) according to the age of the ice (FYI : ice age between 0 and 1 year, MYI : ice age of at least one-year) at a weekly frequency. Data are respectively available as daily and weekly map with a 12,5 km grid resolution. The fraction of MYI is derived from the ice type information during the gridding processing step.

80 ───────────────────────────

**L134-135. Requires citations.**

*The citations were a few lines after, but for clarity we have added one in the first line.*

───────────────────────────

**Figure 1. Could you add here a map of the satellite coverage and the locations of different validation datasets? This**
85 **would be useful for the reader to understand limits of the record and interpret differences to specific validation data.**

*Figure 1. has been completed with a sub figure to represent the locations of different validation data sets with satellite coverage limitations.*

───────────────────────────

90 **L166-167. Which version of the IS-1 data was used?**

*ICESat-1 version used is the one from NASA Goddard not from NASA JPL. This precision has been added in the description paragraph.*

───────────────────────────

**Figure 2. I would suggest to add histograms to one side for each of the three elevation profiles, so it is easier to visualize**
95 **any differences/biases**

*As suggested a sub figure with probability density function, especially density probability function has been added in the following figure 2:*

───────────────────────────

[Figure]

**Figure 1.** Summary of various available dataset for Envisat and ERS validation. Colors distinguish the different types of data. Dark blue for satellite products, light blue for airborne data, yellow for submarines, orange for anchored moorings, green for buoys and red for direct measurements. (a) Temporal availability (b) Spatial availability and extent of missions data gaps. Blue rounds represent altimeters coverage limitation due to their orbit inclination (81.5°N for Envisat, 86°N for ICESat-1 and 88°N for CryoSat-2)

[Figure]

**Figure 2.** Profiles of surface height anomaly over sea ice and ocean for pass 25 between 78° N and 81° N for Envisat in blue-green (cycle 12), ERS-2 in blue and ERS-2 deblurred in orange (cycle 80). The red line represents the limit of 50 % concentration of sea ice, so as the limit between open ocean and an ice-covered area. The dark blue line shows the location of the pass between Svalbard Island and Greenland. (a) The surface height along the latitude and (b) the probability density function of surface height for the three passes with the associated statistics, the average and the standard deviation (SD). The color legend is identical for both sub-figures.

**L215. Sure, but how much are they improved quantitatively if we are using Envisat as the reference ?**

100     *Pulse Blurring has to be seen as an asymmetric noise. As the FB or SLA processing are mainly form by statistical operations, succession of smoothing or interpolations, this noise will be reduced, some outliers due to the blurring will be removed. The resulting impact of blurring will be a positive non-constant bias on the SLA or on the FB, nevertheless the comparison with Envisat is not as easy because both missions (even if the Radar altimeter instrument is identical) are biased, so ERS ASA averaged over the basin with or without blurring compared to Envisat won't be so much relevant. However, we can see the*

105     *impact of the deblurring on the noise of the data. We thus suggest to compare the standard deviation of ASA within each grid cell of a 12.5km resolution grid between Envisat (cycle 12) and ERS (cycle 80) before and after deblurring as an appendice (A1), see Fig.3.*

The deblurred surface anomalies of ERS-2 now appear similar to Envisat.

**replaced by :**

110 The deblurred surface height anomalies of ERS-2 now appear more similar to Envisat in terms of noise and amplitude of variation. For this particular track, the standard deviation has been reduced by 16 % and get closer to Envisat's one. Figure A.1, shows more results on the impact of deblurring on ASA noise reduction comparing to Envisat during a whole cycle.

[Figure]

**Figure 3.** Comparison of the standard deviation of ASA between Envisat (cycle 12) and ERS (cycle 80) before (a) and after deblurring (b) within each grid cell of a 12.5km resolution grid. Median of the standard deviation difference for (a) is 0.075m and 0.008 m
* * *
**L226-227. Can you add a table of the thresholds after they have been calculated to keep lead/floe proportions the same**
115 **during overlap periods ? This would aid the repeatability of the study.**

*We recompute the threshold only for ERS-2, considering Envisat as a reference. Thank you for this suggestion, the clarification has been made in section.The following table : Tab.1 has been added in appendices (A1) in the manuscript.*

**Table 1.** Pulse peakiness thresholds for lead/floe classification

| Mission (RA mode) | PP lead threshold | PP floe threshold |
|---|---|---|
| CryoSat-2 (SAR) | 0.3* | 0.1* |
| Envisat (LRM) | 0.3* | 0.1* |
| ERS-2 (LRM) | 0.2839 | 0.1328 |

\* Guerreiro et al 2017
* * *
**L236-237. More information is required on the interpolation method and procedure.**

*As suggested, we have added some precision to the interpolation procedure.*

L236-237 : Outliers are filtered out with a three standard deviation threshold along 25 km sliding windows. ILA and SLA are interpolated respectively over leads and floes and smoothed with a 25 km rolling mean. The difference between the measured height over floes and over leads is finally made to retrieve the radar freeboard. For the remains of the study, we will only use the FBr measurements made above the floes because the characteristics of the waveforms are used.

**replaced by:**

ILA and SLA outliers are removed by filtering data that are outside the interval : rolling mean $\pm$ 3 rolling Standard Deviation, with a 60 km large sliding window. After filtering, ILA and SLA are smoothed using a rolling mean at 12.5 km, then SLA and ILA are linearly interpolated (including bellows the floes for SLA and above the leads for ILA) and are again smoothed using a rolling mean at 12.5 km. No limit of distance is used to discard radar freeboard, but the interpolation as well as smoothing and filtering is not done with values separated by land. Indeed, the processing is done within ocean segments, separated by land, in order to isolate statistics between segments. In this study, we will only use the FBr measurements that are made over the floes, indeed, the LRM data calibration, explained in section 3.4, is based on floes characteristics.
* * *
**L237-238. Do you discard rFBs above a max distance to the nearest lead? If so what limit do you use?**

*No, we don't discard freeboard above a max distance to the nearest lead. This information has been added commonly within the previous comment. To do so, we have applied the usual geophysical corrections in altimetry, taking into consideration the choice of these data so that they are particularly appropriate for polar oceans.*
* * *
**L250. Can you explain a little more about this constant SLA bias in LRM? Why does it appear and what could be done, in theory, to remove it?**

*The SLA bias comes from the choice to use an empirical retracker with the same fixed threshold for both leads and floes. Poisson et al 2018, explains that over rough surfaces such as ocean, the usual retracked point is close to the position of the half power of the waveforms. Specular waveform that can be found over leads should have retracked point higher, nevertheless, measurements are more stable with lower threshold (Poisson et al 2018 fig.9). This explains why we have a bias for the SLA.*

*Laforge et al 2020 shows that over leads comparing to physical retracker, the SLA bias is constant for altimeters in SARM, nevertheless this conclusion is also relevant for LRM as peaky waveform are all the same over leads. To remove this bias, another threshold should be use e.g. 80 or 90 or a calibration over another mission such as CS-2. The choice over a higher threshold can introduce noise due to the power sampling of the waveform. It could be noticed that this bias also occurs for mission in SAR mode, but it is much smaller due to the fact that SAR waveforms on leads are more peaky than for LRM.*

L248-251. In LRM, most of this error comes from a constant bias on the Sea Level Anomaly

**replaced by**

Negative radar freeboards are mainly due to the retracker choice. Indeed, a TFMRA50 is used to retrack height on both leads and floes, this introduces a bias on the height over the leads. The TFMRA threshold to retrack heights over leads should be closer to 80% and because we use a threshold of 50% that corresponds to the position of the retrack point for ocean surfaces, not specular ones (Poisson et al 2018), the leads are measured higher than they are and even higher than the floes. The SLA bias (in leads) is evaluated constant for SARM altimeter in the study of Laforge et al 2020, this conclusion is also relevant for LRM altimeters as waveforms over the leads are peaky and similar from a lead to an other. This positive constant bias over the leads results to a negative bias on the radar freeboard. To avoid this bias, the retracker threshold could be adapted for leads or the SLA could be calibrated on CryoSat-2 one. Nevertheless, a threshold of 50% ensure the stability of the range (Poisson et al 2018, Fig.9) contrary to higher threshold (80%-95%) that could lead up to 47 cm of random error on the SLA. A TFMRA at 50 % for both leads and floes is preferred in this study as a constant bias is easier to correct than an undetermined random error.
* * *
**Figure 4. Add the sensing mode to the plot. The CS2 data here is SAR mode right, not calculated from pLRM?**

*Yes, CS-2 data is in SAR+SARIn mode here. This precision has been added to the plot (cf 4)*
* * *
**L278. Explain these terms.**

The activation function is a sigmoid.
**replaced by:**
The activation function for the hidden layers neurons is a sigmoid, motivated by possible negative radar freeboard values and the optimizer is and ADAM [Kingma et Ba, 2014].
* * *
**L283-284. What does this mean? Retrained again or just some sort of tuning? Might it be very different from the training with 90-10 split?**

*Once the hyper-parameter combination is set, the MLP is trained with the whole dataset" means that the chosen model (with the optimal combination) is trained 'again' with 100% of the data (again, but weight are reinitialized). The training on 90% comparing to the one on 100% are not slightly different at all, but it's supposed to be better because with have trained it with a larger dataset (larger the dataset is, better the model is supposed to be). For instance, as we have only one winter for Envisat-ERS-2 mission overlap period, 10% is not negligible as it represents a bit less than a month. We have clarified this part, hope it would help the reader to understand better this part of the methodology.*

[Figure]

**Figure 4.** Pan-Artic radar freeboard maps for January 2012 for (a) Envisat uncorrected and (b) CryoSat-2

L283-284 : Hyper-parameters have been tuned by dichotomy by choosing at each step the hyper-parameter combination with the highest mean score (average score made on 5 models) on the test sample. The score used for this regression is the Pearson correlation coefficient. To determine the most suitable hyper-parameter combination, the dataset is randomly split into a training and a testing dataset, corresponding respectively to 90% and 10% of the initial dataset. The activation function used is a sigmoid. Once the hyper-parameter combination is set, the MLP is trained with the whole dataset.

**replaced by:**

The neural network used is a multilayer perceptron (MLP). Both calibrations have been processed with Scikit learn [Pedregosa et al, 2011]. The MLP is composed of 5 hidden layers, each composed of 100 neurons. The choice of hyperparameters : number of neurons, the learning rate, the regularization term, batch size, activation functions, solver for the weights optimization, have been done using gridding methodology, e.g. testing combinations and take the one that give best score. The evaluation criterion, called the score, is chosen as the determination coefficient. Models are trained on 90% of the dataset and tested on the remaining 10%, the splitting in random. During the tuning step, models are cross validated, it means that they are each trained 5 times with the same combination of hyperparameters but without the same train/test dataset, the 5 scores are then analyzed to determine the best combination. Cross validation give a better idea of the model performance as the dependence to the training dataset is limited. The activation function for the hidden layers neurons is a sigmoid, motivated by possible negative radar freeboard values and the optimizer is and ADAM [Kigma et Ba, 2014]. Moreover, in order to avoid over-fitting, an early stopping criterion is used to stop the model training as soon as the score is not improved during 10 consecutive iterations, with a defined tolerance.

Finally, once the hyperparameters combination is set, the MLP is trained on the whole dataset to provide the calibration function. The trained model is then applied to the LRM monthly grids to obtain a monthly LRM-corrected radar freeboard.

**L301-303. Needs more info. Why do you calculate uncertainty differently between leads and floes? The uncertainty at floes is governed by the variability in height measurements at proximal leads.**

**What distance is used to calculate an along-track mean elevation? Is the variability in individual floe height obs around this not just a measure of the topography? It will be higher over MYI but does this realistically mean the uncertainty is higher?**

*We compute differently the SLA uncertainty for the floes (where we interpolate the SLA). Indeed, the rolling standard deviation of the interpolated SLA (where we do not have leads) does not really make sense to estimate the SLA uncertainty on floes. That's the reason why the uncertainty of the SLA where there are no leads is different and is assumed to be the difference between the estimated (interpolated and smoothed SLA) and the mean SLA, this method is taken from Ricker et al 2014. Concerning the along track mean elevation, no limit of distance was used, nevertheless, such as for the interpolation or the smoothing (section 3.2), they are computed per segments of water (so there is one mean SLA per segment of water). As we consider values on floes, we assume that the main part of the random uncertainty of the ILA (ice level anomayl, above the floes) comes from the speckle noise, even if ILA is smoothed, this could be a limitation of the random uncertainty budget. If we understood well the last part of your comment, this will not be the topography of the ice, but the difference between the SLA we have and the mean SLA, which is supposed to represent the SLA we should have. We suggest the following modification in the manuscript, with more details.*

SLA uncertainty ($\sigma_{SLA}$) estimation depends on the surface type (leads or floes). For leads, we take the standard deviation of the measured height within a sliding 25 km window. Concerning floes, the uncertainty is estimated as the difference between the height measurement and the mean elevation along the track (Ricker et al., 2014).

**replaced by :**

SLA uncertainty ($\sigma_{SLA}$) is estimated to be the standard deviation of the SLA within a sliding window of 25 km if there are some leads within this window. If not, the SLA uncertainty is taken as the difference between the interpolated and smoothed SLA and the mean SLA computed as the mean values of measured SLA at leads within a segment of ocean (if the pass is over land, statistics are made segment of ocean by segment of ocean).
* * *
**L307. How are the uncertainties reduced during gridding? Speckle noise should drop as a function of N observations, but SLA uncertainty should only drop as a function of N tracks (because SSH error is highly correlated along track).**

*The way the random uncertainties are computing during the gridding are slightly more complicated than that. We have done the same choice as in Ricker et al, 2014 and compute the grid-cell resulting radar freeboard with a weighted average, with the radar freeboard uncertainty as weights. For each grid-cell :* $FB_r = \sqrt{\dfrac{\sum_1^N \frac{1}{\sigma_{Ri}^2} \cdot FB_{ri}}{\sum_1^N \frac{1}{\sigma_{Ri}^2}}}$ *, so the resulting uncertainty computation should take it into account that weighted average as* $\sqrt{\dfrac{1}{\sum_1^N \frac{1}{\sigma_{Ri}^2}}}$ *. Taken this method, it's difficult to draw a way to make uncertainties for the speckle noise and the SLA dropping differently, even we agree that theoretically the $\sigma_{SLA}$ is correlated along the tracks, thus, the random uncertainty can be a little bit underestimated.*
* * *
**L312. Systematic uncertainty due to roughness is 20-30% of the freeboard as well as of the thickness.**

*Thank you for this confirmation. We adapted the sentence to the radar freeboard.*

Roughness is estimated to be respectively about 20% and 30% of the sea ice thickness for FYI and MYI (Landy et al., 2020).
**replaced by :**
Roughness is estimated to be respectively about 20 % and 30 % of the sea ice thickness for FYI and MYI according to Landy et al., 2020, this results is also applicable for the freeboard
* * *
**L316-318. Based on the schematic in figure 6 everything you've done seems fine, but it is still confusing to follow all the steps. What are these 'other inputs'? And which variables do you divide by the sqrt of the number of observations vs the sqrt of the number of tracks when gridding?**

*We apologize for the confusion raised by this figure. The other inputs are the one in the grey box in the top right of Fig 6, and detailed in Fig.5, (except the date) : the concentration, the Leading edge slope, the Pulse Peakiness, and the Multi-year ice fraction. The uncertainty of these 4 variables is defined as : two times the standard deviation of the values of these variables within the grid cell divided by the sqrt of number of tracks within this grid cell. The uncertainty of the radar freeboard, as said in Sec 3.5, is neither multiplied by two nor divided by the number of tracks. In order to make this steps clearer, we have developed the legend of Fig.5 and Fig.6. and modified the following paragraph :*
L316-317 : The uncertainty of the other inputs is considered to be, for each grid cell, the standard deviation of the measurements used to calculate the average value (grid cell value) divided by the number of tracks passing through the corresponding grid cell.

**replaced by:**
The uncertainty of the other inputs (LES, PP, sea ice concentration, MYI fraction), is considered to be, for each grid cell, two times the standard deviation of the measurements used to calculate the average value (grid cell value) divided by the number of tracks passing through the corresponding grid cell.

As well as :

L325-327 : The method consists of training a number M of NN with noisy inputs (noise has been added to all inputs according to a Gaussian distribution), and then to analyze the distribution of radar freeboard predictions from the M noisy NN applied on noisy N inputs for each considered grids. The whole uncertainty budget process is summarized in Fig. 6.

**replaced by :**

The method consists in training a number M of NN with noisy inputs. The noise has been added to all inputs (for each grid cell and each month) according to a Gaussian distribution centered on the estimated value and the corresponding uncertainty as standard deviation. The calibration processing (training and prediction) is done for all the noisy inputs/output, then the distribution of MxN radar freeboard predictions (from the M noisy NN models applied on N noisy inputs) has been analyzed of each grid cell and each month. The whole uncertainty budget process is summarized in Fig. 6.
* * *
**L325-329. How do you estimate the gaussian noise distribution statistics? is this the $sigma = 2 * sigma_{omega}$ in Figure 6? The output from a monte carlo error budget depends closely on the assumptions taken for the error distributions so this is important.**

*We hope we have clarified this information with the modifications made for the previous comment.*

295 ___________________________

**L340. For which months in Fig 8?**

*This missing information has been added to the manuscript.*

Figure 8 presents the same feature for Envisat and ERS-2 radar freeboard during December 2002 and April 2003.

300 ___________________________

**L357. What are these numbers as a % of the mean rFB?**

*We have added this information in the manuscript.*

The higher mean difference is 7 mm and concerns February 2011, the Envisat calibration. For the ERS-2 calibration, the mean
305 freeboard difference with Envisat does not exceed 3 mm. Concerning all the overlap times, the mean difference is 3 mm for Env/CS-2 calibration, and -2 mm for ERS-2/Env one.

  **has been replaced by :**
The highest mean difference reaches 7 mm in February 2011 for Envisat calibration, i.e. 9.5% of the mean Envisat radar
310 freeboard. For ERS-2 calibration, the mean freeboard difference between ERS-2 and Envisat does not exceed 3 mm, 3.3% of ERS-2 mean radar freeboard. Concerning all the overlap times, the mean difference is 3 mm for Env/CS-2 calibration, 4.1% of Envisat mean radar freeboard and -2 mm for ERS-2/Env one, about 2.2% of ERS-2 mean radar freeboard.
* * *
315 **L359. Again what are these as a % of the mean rFB?**

*This information has been added to the previous paragraph (see previous comment).*
* * *
**L363. Can you do the same for the ERS2-Envisat comparison?**

*Yes, this information has been added as following :*

320 Similarly, for the period 2002/2003, the mean and median uncertainties of ERS-2 are always larger than those of Envisat by about 6 cm over the median radar freeboard (see Tab.A5 and Tab.A3 for more statistics)

  **has been replaced by :**
Concerning the period 2002/2003, the median uncertainty is 8 cm for ERS-2 radar freeboard and 7.3cm for Envisat, similarly,
325 statistics on uncertainties are globally higher for ERS-2 estimates (see Tab.A3 and Tab.A4 for detailed statistics).

**Figure 7b. It looks like you may have some spurious tracks in Hudson Bay, Baffin Bay and Bering Strait that could contaminate the comparisons?**

*Unfortunately, in spite of the different filtering, spurious tracks remains, especially close to the coast. Nevertheless, few val-*
330 *idation data are available in these areas, so comparisons with independent data sets would not be contaminated with these spurious tracks.*
* * *
**Figure 7 caption. Emphasize the distributions include CS2 data only for the coinciding region south of 81.5N.**

*Figure 7 caption has been modified as followed :*

335    Comparison of Envisat calibrated radar freeboard against CryoSat-2 reference for December 2010 in the upper half and April 2011 in the lower half. The maps (a) and (g) refers to Envisat aside with corresponding CryoSat-2 radar freeboard (b) and (h). Maps bellow (d), (e), (j) and (k) are the related uncertainties. The right column presents differences freeboard maps (Env-CS-2) ((c) and (i)). (f) and (l) are the distribution of Envisat F Br in red, CryoSat-2 F Br in blue and in grey.

340    **has been replaced by**

    Comparison of Envisat calibrated radar freeboard against CryoSat-2 reference for December 2010 in the upper half and April 2011 in the lower half. The maps (a) and (g) refers to Envisat aside with the corresponding CryoSat-2 radar freeboard (b) and (h). Maps bellow (d), (e), (j) and (k) are the related uncertainties. The right column presents freeboard difference maps
345 (Env-CS-2) ((c) and (i)). (f) and (l) are the distribution of Envisat $FBr$ in red, CryoSat-2 $FBr$ in blue and $\Delta FBr$ in grey. Histograms only include common data between Envisat and CryoSat-2, data north or 81.5 °N are excluded. $\mu$ refers to the average and $SD$ to the Standard Deviation.
* * *
**L384. 'static data'?**

350 *'Static data' refers to moorings, data set have fixed longitude and latitude for a given time. We have clarified this point as following:*

Static data are monthly averaged to get one value per month.

    **replaced by :**
355

    Concerning moorings or coastal measurement stations, data are averaged to get one value per month.
* * *
**L392. I think it is reasonable to discount IMBs because they represent only the single floe they are deployed on (usually a thicker floe) and not their surrounding 12x12 km grid cell area. The authors could remove these comparisons so they**
360 **don't draw reader's attention and they come to the wrong conclusions about the satellite data validity; but that is up to the authors.**

*Thank you for this suggestion. We agree with this point of view. This comparison was a part of the validation for completeness, but also to get feedback from the community as we were struggling while using this data. To remove the confusion, we decided not to compares Envisat data with IMB as advised.*

**L397. Could be attributed to, but not definitely.**

*We have changed this sentence to stay more hypothetical.*

The bias between OIB and Envisat estimation can also be attributed to the OIB snow depth estimation that remains slightly different from one algorithm to another (Kwok et al., 2017).

   **replaced by :**
The bias between OIB and Envisat estimation could also be partly attributed to the OIB snow depth which estimation seems sensitive to the algorithm used (Kwok et al., 2017).
* * *
 **Figure 9 and elsewhere. Define the acronyms of statistical tests in the caption.**

*Captions of Figure 7,8,9,10,11,12 have been modified to add the statistic acronyms definitions.*
* * *
**L406-407. Can this say anything about the calibration? Are the BGEP ice conditions more representative of average sea ice conditions in the Arctic and the other ULS datasets more of thin ice conditions? Was the calibration not slightly**  **overestimating thin ice thickness for Envisat?**

*This could tell something about the calibration, but it's hard to conclude knowing that BGEP comparisons and other moorings comparisons draw different conclusions. Considering BGEP, calibration seems to overestimate thin ice which is not the case comparing to moorings within the Laptev sea.*
* * *
 **L415-416. How do these numbers compare to your estimated uncertainties for the same regions?**

*We don't really know how to compare properly the Standard deviation and biases with our uncertainties as the question of the uncertainties on the SIT is another issue and for this paper the uncertainties are limited to the radar freeboard. We propose the Figures 10,11,12 and 13 that present the 95% confidence interval of the SIT but without taking into account uncertainties on snow depth, densities etc for the FBr to SIT conversion step.*
 *The plots have been updated with the variable snow density as asked by Robbie Mallet so they are not similar to those in the previous version of the manuscript. For esthetical reason, bounds are not represented for comparisons with other satellite-based SIT estimation.*
* * *
**L420-422. What are the statistics like for CS-2 data processed with this method? You don't necessarily need to show a**  **plot, but some idea of biases would be useful. Do you also see generally negative biases for CS2? Especially over FYI?**

*Just for information, Figure 9 show comparisons for CS-2 thickness with OIB, BGEP and Transdrift Laptev Sea. For BGEP bias is higher, CS-2 FYI thickness seems to be more overestimated than Envisat one but with find this same negative bias for Transdrift Laptev Sea moorings especially for thick ice.*

[Figure]

**Figure 5.** Comparative scatter-plots between Envisat sea ice thickness or radar freeboard estimations and other data sets. The x-axis indicates the sea ice thickness from (a) OIB total ice freeboard, (b) Air EM snow plus ice thickness, (c) Can Coast ice thickness, (d) UK/US submarines draft and (e) ICESat-1 total freeboard. (f) compares our Envisat radar freeboard with SI-CCI Envisat solution. Colorbars represent the normalized density. A $log_{10}$ has been applied before the normalization for (e) and (f) due to the large number of data. N is the number of the couple of values that are compared, Med refers to the Median, SD the Standard deviation, RMSE the Root Mean Square Error and $r$ the correlation coefficient.

*In order to help the reader, with have added the following sentences:*

400   As a comparison, the bias between CryoSat-2 and OIB between 2010 and 2019 is about 16 cm and the RMSE 77 cm. Concerning BGEP (2010-2021) comparisons, the bias is 21 cm with the same overestimation of FYI thickness for CS-2 and with Transdrift Laptev Sea (2010-2016) comparisons show a negative bias of -38 cm.

[Figure]

**Figure 6.** Comparative scatter-plots between Envisat sea ice thickness estimations and anchored moorings data sets. Each dot corresponds to a monthly averaged value. The x-axis indicates the sea ice thickness from (a) BGEP, (b) BGEP vs Env CCI, (c) Davis Strait, (d) IOS CHK/EBS and (e) Transdrift Laptev Sea ice draft. The colorbar shows the MYI fraction. N is the number of the couple of values that are compared, Med refers to the Median, SD the Standard deviation, RMSE the Root Mean Square Error and $r$ the correlation coefficient.
* * *
**L425-426. Could you try them also with the adapted warren climatology and see if biases get any smaller? Would help to clarify the impact of snow loading.**

The same plots with Warren 99 modified climatology are presented in Fig.10, Fig. 11, Fig. 12 and Fig. 13. Note that the scatters for OIB, CanCoast and EnvisatCCI are unchanged because no additional snow products are used. Biases with Air EM is reduced as well as for the submarines. Nevertheless, as explained in the manuscript, a bias is expected when comparing to Submarines of about 30 cm. The dispersion is augmented for the comparisons with moorings data. Thus, the snow load has an important impact, we suggest showing this figure in appendice, as it could impact the clarity of validation if it is shown in the validation section.
* * *
**L427. Is Section 2.3 correct?**

*Sorry for this typo mistake, the reference was indeed "section 4.2". This has been corrected.*

[Figure]

**Figure 7.** Comparative scatter-plots between ERS-2 sea ice thickness estimations and 3 in-situ data sets. The x-axis indicates the sea ice thickness from (a) AirEM total thickness, (b) UK/US Submarines draft and (c) Can Coast sea ice thickness. Colorbar indicates the normalized density. N is the number of the couple of values that are compared, Med refers to the Median, SD the Standard deviation, RMSE the Root Mean Square Error and $r$ the correlation coefficient.

415

[Figure]

**Figure 8.** Comparative scatter-plots between ERS-2 sea ice thickness estimations and 2 anchored moorings data sets. The x-axis shows sea ice thickness estimations from (a) IOS Beaufort Sea and (b) AWI moorings sea ice draft. The color bar indicates the respective MYI fraction. N is the number of the couple of values that are compared, Med refers to the Median, SD the Standard deviation, RMSE the Root Mean Square Error and $r$ the correlation coefficient.

**L441. It is worth making it a bit clearer on Fig 13 and throughout this section that these volumes miss out everything >81.5N.**

*The caption has been clarified as following :*

Fig 13 caption : Time series representing radar freeboard volume up to 81.5°N for each winter month for ERS-2 in orange, Envisat in teal and CS-2 in dark red. Blue triangles are winter mean volumes. Red lines are linear regressions of winter mean

[Figure]

**Figure 9.** Comparative scatter-plots between CryoSat-2 sea ice thickness estimations and OIB/BGEP. The x-axis indicates the sea ice thickness from (a) OIB total thickness, (b) BGEP draft and (c) Transdrift Laptev Sea draft. Colorbar indicates the normalized density and MYI fraction. N is the number of the couple of values that are compared, Med refers to the Median, SD the Standard deviation, RMSE the Root Mean Square Error and $r$ the correlation coefficient.

volume until 2002/2003 for dashed line and 1995/1996 for solid line.

425  **replaced by:**

Fig 13 caption : Time series representing radar freeboard volume up to 81.5°N for each winter month (no data between 81.5 °N and 90 °N, even for CS-2 for consistency). ERS-2 in orange, Envisat in teal and CS-2 in dark red. Blue triangles are winter mean volumes. Red lines are linear regressions of winter mean volume until 2002/2003 for dashed line and 1995/1996 for solid 430  line.

**addition of :** It's important to note that the volumes presented in Fig. 13 are only considering values up to 81.5 °N.
* * *
**Figure 13. I think readers would find it interesting to see more of your rFB dataset. I'd suggest an additional figure** 435  **showing trends in rFB as a map for the overlap region, highlighting where the trends are significant or not.**

*We apologize to show so limited data set, we feel that the paper is already rather long as it is and this will be the subject of another study.*
* * *
440  **Table A1. SAR you mean? or is this actually the CS2 LRM mode? I think SAR was used here right so state SAR parameters?**

*It's actually the pLRM mode, this has been explained in a previous answer, we hope that the use of this table is now clearer.*

[revised manuscript text omitted]

---

## Author Comment (AC2)

**Title: Arctic sea ice radar freeboard retrieval from ERS-2 using altimetry : Toward sea ice thickness observation from 1995 to 2021**

*Marion Bocquet, Sara Fleury, Fanny Piras, Eero Rinne, Heidi Sallila, Florent Garnier, and Frédérique Rémy*

5 **Anonymous referee n°3 - global comments**

First, I would like to express my apologies to the authors for taking this long to provide my review due to personal reasons. Nonetheless, I was asked to still provide it also in the light of the two already published referee comments. This in mind, I will focus on aspects I do not see covered yet or extend on raised issues as I see fit with a focus on the "calibration" using a neural network. I provide general comments first with some additional specific comments at the end.

10    The authors present in their study their way of generating a new dataset of altimetry-based freeboard data with ERS-2 data incorporated for the first time. This is a great achievement in itself and definitely justifies publication. Furthermore, the authors put substantial effort in validating their results against several different types of validation data. ERS data in general is a great challenge to work with and there is a reason why not many people are actually working on the task to make use of them over sea ice.

15    However, as also pointed out in the very detailed review by Robbie Mallet, who went to great lengths to analyze the results and underlying data, it appears the chosen methodology does not really work the way the authors or at least any potential reader would expect it. There appears to be strong evidence that the large mix of input data to the neural network along side the ERS freeboard estimates dominate the outcome. Hence, the NN did not learn what was expected but something else. While this is not necessarily bad, it is a fundamental problem of the presented study, as in my opinion, this is can be seen as
20    grist to the mills of all machine learning or artificial intelligence sceptics. It should clearly be stated what the impact of each dataset is on the resulting product or rather that its apparently not the input raw freeboard. Potentially, the product could even e generated without the raw freeboard? This really should be clarified upfront and likely further investigated by the authors before publication.

**Answer to Anonymous referee n°3 - global comments**

25 We would like to thank the reviewer for his careful reading of the manuscript and for the relevant remarks that have helped to improve the quality of the manuscript. In order to fit with your comments, we have made a revision of the manuscript that should have corrected the textual issues and well improved the readability of the document. We hope that these modifications will meet your requirements. Please find below the details on how your specific comments have been taken into account.
    As the referee n°3 seems to have the same concerns as referee n°2 with some conclusions taken from the other referee's
30 review, we propose the same global answer as for referee n°2.

    In our understanding, the main concern of the reviewer is : "To what extent can we claim that the resulting product is a corrected or calibrated retrieval when it doesn't reflect the variability in the raw, retracked values ? " Expressed in other word, the referee states that "nothing of the original radar freeboard measurement remains in the corrected value" and that is an issue.

    First, we would like to indicate that referee n°2's detailed analysis on the correlations between raw freeboard and calibrated
35 radar freeboard have pointed out difficulties that have motivated the calibrations (past studies) and thereafter the use of a neural network.

Using the exact same processing chain as for CryoSat-2 (with a TFMRA-50 retracker), the Envisat, and ERS radar freeboard estimates are very different to what we are supposed to observe in terms of magnitude and spatial patterns. Indeed, LRM waveforms are strongly impacted by the size of the footprint which is much larger in LRM than in SAR Mode ($\sim 180$ km$^2$ to $\sim 5$ km$^2$ [Stammer et al, 2018]). This is the main cause for the misfit between Envisat and CS-2 radar freeboards. In order to deal with this issue, differences between CS-2 and Envisat have been analyzed, please see Guerreiro et al. 2017, Paul et al. 2018 and Tilling et al. 2019 for a complete overview. As mentioned in the manuscript, the first two studies point out that the radar freeboard differences between the two altimeters are correlated (not especially linearly correlated) to the sea ice roughness, characterized by the waveform backscatter, the leading edge width or the pulse peakiness. The third study identified a link between the misfit and the distance between floes and leads.

The optimal solution would be to find a theoretical model, such as the Brown's model over open ocean, to represent the radar response over sea ice in order to correctly retrack the waveform. Despite significant progress in SAR mode (SAMOSA+, LARM, etc.), these models are not yet able to represent all the complexity of this response even in SARM. For instance, they are not able to represent snow penetration effects (i.e. volume backscatter effects). Moreover, in LRM, no study reports relevant retracked height over sea ice floes with a physical retracker and the complexity of the response is still poorly understood. The objective of this paper is neither to model any effect of the ice surface condition, nor to understand its influence on the FBr but rather to reconstruct the best possible ERS-2 radar freeboard with our actual knowledge consistently with Envisat and CS-2 ones. To do so, roughness or more globally sea ice surface state proxies are used to post-correct the estimated radar freeboard using as a reference Envisat previously calibrated on CS-2. Our study is based on the principle that the radar freeboard computed with a TFMRA50 from LRM waveforms is strongly polluted by the surface roughness. Then, we propose to calibrate LRM radar freeboard on CS-2 using some parameters characterizing the sea ice surface roughness. The same methodology is applied to calibrate ERS-2 radar freeboard on a CryoSat-2 like radar freeboard from Envisat. Thereafter, some other parameters such as the ice concentration or the sea ice age were added to improve and consolidate the learning of the NN so to reach a better match with CS-2 (in the case of Envisat and Envisat calibrated for ERS-2).

Unlike the review suggests, we would like to specify that the sea ice age is not directly used in the regression, we use a MYI fraction. The way this fraction is calculated has been developed in the manuscript, but it is not discrete values, as it is considered by the reviewer. Also, we would like to specify that correlations calculated with a variable that takes only two values can not be relevant.

To illustrate that the calibration is based on the PP and the LES, Figure 1 shows radar freeboard for April 2011 for CS-2, for Envisat with the calibration presented in the manuscript and one with another model trained only with the raw freeboard, the Pulse Peakiness and the Leading edge slope. It shows that these three parameters are sufficient to represent the magnitude and the patterns we are supposed to see in the Arctic. The other parameters help the calibration to get closer to CS-2 radar freeboard and bring more spatial variability.

The correction we have to process is strongly non-linear, so this is the reason why we have chosen a neural network approach, which has the specificity to handle well with non linearities. Then, correlation between parameters based on linear approximation are not representative of the dependencies between parameters (inputs/output) in the neural network. Indeed, it is much more complicated to estimate the relative importance of each parameter in a regression, and it is not given by the correlations between the inputs and the predicted value. As it has been already mentioned, the main reason is that the relations that have been established by the neural network are not linear, while the correlation only evaluate whether the variables are related by a linear relation. Figure 2 shows the "partial dependencies" which refers to an illustration (statistically computed) of the relations between each input parameters (x-axis) and the predicted value (y-axis). It also illustrates the relative importance of each parameter: a parameter with no influence would have a horizontal curve as a mean state but it is not a quantitative approach. Partial dependency plots should be interpreted with caution, it refers to the mean state of statistical computation, depends on a discretization choice and values of input parameters have been standardized (mean = 0 and Standard deviation=1). The Figure 2 presents 2 panels, one for Envisat calibration (left) and one for ERS-2 calibration (right). Nevertheless, we can say that curves are not linear, no input is unused or with a very low influence, we can also note that LES and PP have the largest influence on the predicted radar freeboard.

[Figure]

**Figure 1.** Radar freeboard from CS-2 (left), Envisat calibrated presented in the paper (middle) and Envisat using only the raw freeboard+LES+PP (right)

[Figure]

**Figure 2.** Partial dependencies plots for Envisat and ERS-2 calibration, from top left to top right, inputs are, the raw radar freeboard, the month, the pulse peakiness, the Leading edge slope, the concentration and the $f_{MYI}$.

The question of the non-linearity is central in our study but also in referee n°2's analysis. But, since the correction is not linear at all, the largest raw radar freeboard is not necessarily the largest corrected freeboard, just as it is not the largest raw freeboard that will benefit from the largest correction. The raw radar freeboard, even noisy, still gives information on how the altimeter perceives the surface and how much it should be corrected, which remains an important information. We expect that the raw freeboard define the space and time variability of the calibrated radar freeboard over the whole period but this is hard to show since we don't have any reference of the expected variability of the SIT/FBr/FB during 1995-2010. To enhance the fact that the raw radar freeboard impacts the corrected radar freeboard, figure 3 shows the relative difference between the predicted FBr of the NN presented in the paper and one from a NN trained without the raw FBr for April 2011. It shows that for a large part of the basin, the difference of FBr is up to 25% of the predicted radar freeboard.

Finally, it's important to keep in mind that we have trained the neural network to reach the best score i.e. the best coefficient of determination (compared to CS-2 for Envisat and to Envisat corrected for ERS-2). Choosing the best NN, means choosing the combination of hyperparameters and even the choice of input parameters that gives the best scores. This means that the

[Figure]

**Figure 3.** Relative difference in % between the corrected freeboard of our study and one with the same NN trained without the raw freeboard

95    fraction of MYI allows to better fit CS-2 radar freeboard, that's why we keep it. However, it's even expected to find a good correlation between the sea ice age and the sea ice freeboard because, in average, older ice will be thicker.

To sum up, the purpose of this paper is to retrieve a consistent radar freeboard estimation for ERS-2 using the current knowledge on LRM waveforms over sea ice. Because LRM waveforms are highly impacted by the surface state and poorly understood over sea ice, raw freeboard have to be calibrated. Two calibrations need to be implemented to get consistent ERS-2

100   radar freeboard, first Envisat against CS-2 and then ERS-2 against Envisat calibrated radar freeboard. The calibration is first based on surface roughness proxy because evident link have been emphasized with the size of the correction (previous studies) and secondly on auxiliary data that were used to reach better fit with CS-2. The calibrated radar freeboard is partly driven by the raw radar freeboard, both parameters are not linear correlated as it would say that the calibration did not perform well. The "age" or in our case the MYI fraction is not the key input for the NN training. Furthermore, 'ice with a higher-than-average

105   raw FBr in a given month" can not necessarily "end up with a higher than average corrected FBr value" as the calibration is not linear.

Unfortunately, the impact of each parameter could be a dedicated study and it would not be the purpose of this study.
* * *
*In this document, the referee's comments are in bold type, the answers are in italic type, and the corrections to the revised*
110   *manuscript are in normal type.*

**L257: One could doubt the idea to use this kind of freeboards as an input in the first place. Wouldn't it make a difference to choose a more appropriate retracker threshold for leads in LRM waveforms like 90/95%? This might not solve the problem with regional patterns but would likely eliminate the negative freeboards and deliver a better initial state.**

115   *Empirical retracker with a threshold of 90/95% can be a risky strategy. The error on the range would vary a lot according to the sampling, randomly up to a gate ($\sim$ 47cm). Using a 50% ensure the stability of the range (Poisson et al 2018 fig.9) even if we know we have a bias. Laforge et al 2020 shows that over leads comparing to physical retracker, the SLA bias is constant for altimeters in SARM, nevertheless this conclusion is also relevant for LRM as peaky waveforms are similar over leads. We*

*prefer to correct a bias than a random error, as we don't usually have a lot of measurement over leads. As suggesting for Jack*
120 *Landy's review, we propose the following modification:*

In LRM, most of this error comes from a constant bias on the Sea Level Anomaly

**replaced by**

Negative radar freeboards are mainly due to the retracker choice. Indeed, a TFMRA50 is used to retrack height on both leads
125 and floes, this introduces a bias on the height over the leads. The TFMRA threshold to retrack heights over leads should be
closer to 80% and because we use a threshold of 50% that corresponds to the position of the retrack point for ocean surfaces,
not specular ones (Poisson et al 2018), the leads are measured higher than they are and even higher than the floes. The SLA
bias (in leads) is evaluated constant for SARM altimeter in the study of Laforge et al 2020, this conclusion is also relevant for
LRM altimeters as waveforms over the leads are peaky and similar from a lead to an other.
130 This positive constant bias over the leads results to a negative bias on the radar freeboard. To avoid this bias, the retracker
threshold could be adapted for leads or the SLA could be calibrated on CryoSat-2 one. Nevertheless, a threshold of 50% ensure
the stability of the range (Poisson et al 2018, Fig.9)) contrary to higher threshold (80%-95%) that could lead up to 47 cm of
random error on the SLA. A TFMRA at 50 % for both leads and floes is preferred in this study as a constant bias is easier to
correct than an undetermined random error.

135
* * *
**On a very general note: What are the improvements over Guerreiro et al (2017)? What justifies the use of a neural network instead of simply extending this methodology? As it had a more direct link to the actual measurements of the instrument? (as suggested also by the authors in L258-260)**

*Figure 4 shows comparisons between Envisat-G radar freeboard from Guerreiro et al 2017, Envisat-B from this study com-*
140 *pared to CryoSat-2 TFMRA50 radar freeboard. Maps present the mean radar freeboard difference between Envisat and CS-2*
*for the 12 months of the mission overlap period. These maps reveal that the first radar freeboard estimates were underestimated*
*over the entire basin and especially in thick ice area. The correlation is much higher for Envisat-B radar freeboard. Compared*
*to CryoSat-2, the improvement is evident with the method developed in this study. This study also reveals two modes of radar*
*freeboard that could correspond to MYI/FYI.*
145 *As mentioned in the manuscript, the use of a neural network is justified by the non-linearities that exist between inputs and*
*output, especially since it is not necessary to make any assumptions about the nature of the relations. The calibration can be*
*seen as a usual regression in the continuity Guerreiro et al, 2017 with additional inputs to increase the match with SARM radar*
*freeboard.*
* * *
150 **L277: Out of curiosity, did the authors test various setups and this architecture of the NN showed the best results? How was it evaluated and what different setups were used? Things like the number of layers, number of neurons per layer, activation functions etc. come to mind and all the mentioned specifics come without references or justification! For example, there are pretty much no modern studies on ML/AI that do not use some sort of ReLU activation functions, why do the author use a Sigmoid? Some elaboration on this might be informative to the readers as well and also provide**
155 **a broader background also to non-ML enthusiasts in the sea-ice community.**

*Yes, a lot of setups has been tested. As already mentioned in the manuscript, the hyperparameters which also included the*
*architecture (number of neurons and layers) have been choosing by gridding, e.g. by testing a large amount of combination.*
*It was evaluated by cross validation, training 5 times the same setup on different training sets (randomly chosen) and test on*
*the remaining 10% testing dataset (each time different). The best statistics on the ensemble of the 5 five scores were used to*
160 *evaluate a combination. However, models with 102 neurons per layer instead of 100 were not significantly different.*
*The tuning of hyperparameters has been made by experimenting, that the reason why there are no references, if you mean*
*referencing other geophysical studies, it will have no sens as the tuning is really specific to the study. Activation function relies*

[Figure]

**Figure 4.** Comparison of Envisat-G (Guerreiro et al, 2017) and Envisat-B (Bocquet et al, 2022) with CryoSat-2 for the missions overlap period

*on the type of value of the output data and ReLu function does not allow negative values, which could occur here. We propose the following modifications:*

165    The neural network is a multilayer perceptron regressor (MLP) composed of 5 hidden layers, each composed of 100 neurons. The activation function used is a sigmoid. Hyper-parameters have been tuned by dichotomy by choosing at each step the hyper-parameter combination with the highest mean score (average score made on 5 models) on the test sample. The score used for this regression is the Pearson correlation coefficient. To determine the most suitable hyper-parameter combination, the dataset is randomly split into a training and a testing dataset, corresponding respectively to 90% and 10% of the initial dataset. To avoid
170    overfitting, we use early stopping to interrupt the training when the score is not improving anymore. Once the hyper-parameter combination is set, the MLP is trained with the whole dataset. The NN trained is then applied to the LRM monthly grids to obtain a monthly LRM-corrected radar freeboard.

**replaced by:**

The neural network used is a multilayer perceptron (MLP). Both calibrations have been processed with Scikit learn [Pe-
175    dregosa et al, 2011]. The MLP is composed of 5 hidden layers, each composed of 100 neurons. The choice of hyperparameters : number of neurons, the learning rate, the regularization term, batch size, activation functions, solver for the weights opti- mization, have been done using gridding methodology, e.g. testing combinations and take the one that give best score. The evaluation criterion, called the score, is chosen as the determination coefficient. Models are trained on 90% of the dataset and tested on the remaining 10%, the splitting in random. During the tuning step, models are cross validated, it means that they are
180    each trained 5 times with the same combination of hyperparameters but without the same train/test dataset, the 5 scores are then analyzed to determine the best combination. Cross validation give a better idea of the model performance as the dependence to the training dataset is limited. The activation function for the hidden layers neurons is a sigmoid, motivated by possible negative radar freeboard values and the optimizer is and ADAM [Kigma et Ba, 2014]. Moreover, in order to avoid over-fitting, an early stopping criterion is used to stop the model training as soon as the score is not improved during 10 consecutive iterations, with
185    a defined tolerance.

Finally, once the hyperparameters combination is set, the MLP is trained on the whole dataset to provide the calibration function. The trained model is then applied to the LRM monthly grids to obtain a monthly LRM-corrected radar freeboard.
* * *
**L279: The authors should clarify hyper parameters to the non-AI/ML expert readers. Without any reference I fear**
190    **this is a lot to ask from potential readers of a non-AI journal. Additionally, what optimizer did the authors use as this can also have a substantial impact on the training process and the model performance and is totally unmentioned in the current version of the manuscript.**

*We hope that the previous comment will give an element of answer. The optimizer used is an adam, according to your comment, this information has been added.*

195    ---

**L280: It is not clear to me how these 5 models are differing from each other? By slightly different choices on the learning rate? Please elaborate!**

*They are differing from their training dataset as it has been randomly split each of the 5 times. 5 models are used to make the cross validation. We also hope that the modification of the paragraph L277-285 help the reader and has clarified this part*
200    *of the manuscript.*
* * *
**L282: Common practice would be a split around 80/20% or 75/25%, how do the authors justify such a small test-set size? This could result in a quite non-representative test dataset in the end.**

[Figure]

**Figure 5.** Probability density function of target radar freeboard for Envisat calibration for Test (blue) and Train (orange) dataset

*This split can be justified by the fact that the dataset is quite important, with about 600000 values for Envisat calibration and*
205 *about 300000 values for ERS-2 calibration. Figure 5 shows the probability density of the target radar freeboard for Envisat calibration for both test and train sample. Densities are identical for both datasets, this supports the fact that this splitting, in our case, leads to a representative test dataset.*
* * *
**Answers to referee 3 : specific comments**
210

**L118 & 122: the (Lindsay and Schweiger, 2013) reference should not be in parenthesis.**

*This point has been corrected.*
* * *
**L126: I think these PP thresholds should be mentioned here in a Table or within the text.**

215 *The following table has been added in appendix (Table A1).*
* * *
**L284: This should be 'the trained NN' not the 'the NN trained'.**

*This comment has been taken into account.*

**Table 1.** Pulse peakiness thresholds for lead/floe classification

| Mission (RA mode) | PP lead threshold | PP floe threshold |
|---|---|---|
| CryoSat-2 (SAR) | 0.3* | 0.1* |
| Envisat (LRM) | 0.3* | 0.1* |
| ERS-2 (LRM) | 0.2839 | 0.1328 |

\* [Guerreiro et al, 2017]
* * *
220 **L286: I might just have missed it (sorry then) but what is the SARM abbreviation?**

*Indeed, this acronym was missing in the list of acronyms. SARM refers to 'Synthetic Aperture Radar Mode' to be consistent with LRM 'Low resolution Mode'. This point has been taken into account.*
* * *
**Figure 6: This definitely needs a much larger figure caption!**

225 *We do agree with that comment, the following modification has been done:*

[revised manuscript text omitted]

---

## Author Comment (AC3)

https://fr.overleaf.com/project/63355dc42064afcdebb2fb5e

**Manuscript ID: egusphere-2022-214**

**Title: Arctic sea ice radar freeboard retrieval from ERS-2 using altimetry : Toward sea ice thickness observation from 1995 to 2021**

*Marion Bocquet, Sara Fleury, Fanny Piras, Eero Rinne, Heidi Sallila, Florent Garnier, and Frédérique Rémy*

**Robbie Mallet (referee n°2) - global comment**

**Global comment as a community comment :**

I was really pleased to see this paper come up in TCD. I think that generating radar freeboard data from ERS1/2 is one of the most pressing tasks for the sea ice community, and so I agree with Jack Landy's review. Overall I think the paper is well written and addresses what is a very significant gap in our knowledge of the Arctic Ocean. In particular I think the figures are well-designed. I do have a couple of concerns, questions and suggestions over wordings, citations etc. I hope the authors will take these in the spirit of discussion, rather than as negative criticism. I really do think that this research is high-quality and useful.

**Global comment as a referee :**

I left a community comment on this manuscript (https://doi.org/10.5194/egusphere-2022-214-CC1) before being nominated as a referee. I have therefore read and considered the manuscript again. As part of this, I investigated the data that was made available to me as a nominated reviewer. I wanted to see the size of the correction/calibration applied by the neural network presented in this paper. This has led me to question the nature of the 'correction' being applied, and whether it is reasonable to present this data product as a series of 'corrected' radar freeboard values at all. I would like to review this manuscript again once the queries raised here have been addressed.

**Answer to Robbie Mallet (referee n°2) - global comment**

We would like to thank the reviewer for his careful reading of the manuscript and for the relevant remarks that have helped to improve the quality of the manuscript. In order to fit with your comments, we have made a revision of the manuscript that should have corrected the textual issues and well improved the readability of the document. We hope that these modifications will meet your requirements.

In our understanding, the main concern of the reviewer is : "To what extent can we claim that the resulting product is a corrected or calibrated retrieval when it doesn't reflect the variability in the raw, retracked values ? " Expressed in other word, the referee states that "nothing of the original radar freeboard measurement remains in the corrected value" and that is an issue.

First, we would like to indicate that your detailed analysis on the correlations between raw freeboard and calibrated radar freeboard have pointed out difficulties that have motivated the calibrations (past studies) and thereafter the use of a neural network.
Using the exact same processing chain as for CryoSat-2 (with a TFMRA-50 retracker), the Envisat, and ERS radar freeboard estimates are very different to what we are supposed to observe in terms of magnitude and spatial patterns. Indeed, LRM

waveforms are strongly impacted by the size of the footprint which is much larger in LRM than in SAR Mode ($\sim$ 180 km$^2$ to
$\sim$ 5 km$^2$ [Stammer et al, 2018]). This is the main cause for the misfit between Envisat and CS-2 radar freeboards. In order to deal with this issue, differences between CS-2 and Envisat have been analyzed, please see Guerreiro et al. 2017, Paul et al. 2018 and Tilling et al. 2019 for a complete overview. As mentioned in the manuscript, the first two studies point out that the radar freeboard differences between the two altimeters are correlated (not especially linearly correlated) to the sea ice roughness, characterized by the waveform backscatter, the leading edge width or the pulse peakiness. The third study identified a link
between the misfit and the distance between floes and leads.

The optimal solution would be to find a theoretical model, such as the Brown's model over open ocean, to represent the radar response over sea ice in order to correctly retrack the waveform. Despite significant progress in SAR mode (SAMOSA+, LARM, etc.), these models are not yet able to represent all the complexity of this response even in SARM. For instance, they are not able to represent snow penetration effects (i.e. volume backscatter effects). Moreover, in LRM, no study reports relevant
retracked height over sea ice floes with a physical retracker and the complexity of the response is still poorly understood. The objective of this paper is neither to model any effect of the ice surface condition, nor to understand its influence on the FBr but rather to reconstruct the best possible ERS-2 radar freeboard with our actual knowledge consistently with Envisat and CS-2 ones. To do so, roughness or more globally sea ice surface state proxies are used to post-correct the estimated radar freeboard using as a reference Envisat previously calibrated on CS-2. Our study is based on the principle that the radar
freeboard computed with a TFMRA50 from LRM waveforms is strongly polluted by the surface roughness. Then, we propose to calibrate LRM radar freeboard on CS-2 using some parameters characterizing the sea ice surface roughness. The same methodology is applied to calibrate ERS-2 radar freeboard on a CryoSat-2 like radar freeboard from Envisat. Thereafter, some other parameters such as the ice concentration or the sea ice age were added to improve and consolidate the learning of the NN so to reach a better match with CS-2 (in the case of Envisat and Envisat calibrated for ERS-2).

Unlike the review suggests, we would like to specify that the sea ice age is not directly used in the regression, we use a MYI fraction. The way this fraction is calculated has been developed in the manuscript, but it is not discrete values, as it is considered by the reviewer. Also, we would like to specify that correlations calculated with a variable that takes only two values can not be relevant.

To illustrate that the calibration is based on the PP and the LES, Figure 1 shows radar freeboard for April 2011 for CS-2,
for Envisat with the calibration presented in the manuscript and one with another model trained only with the raw freeboard, the Pulse Peakiness and the Leading edge slope. It shows that these three parameters are sufficient to represent the magnitude and the patterns we are supposed to see in the Arctic. The other parameters help the calibration to get closer to CS-2 radar freeboard and bring more spatial variability.

[Figure]

**Figure 1.** Radar freeboard from CS-2 (left), Envisat calibrated presented in the paper (middle) and Envisat using only the raw free-board+LES+PP (right)

The correction we have to process is strongly non-linear, so this is the reason why we have chosen a neural network approach, which has the specificity to handle well with non linearities. Then, correlation between parameters based on linear approximation are not representative of the dependencies between parameters (inputs/output) in the neural network. Indeed, it is much more complicated to estimate the relative importance of each parameter in a regression, and it is not given by the correlations between the inputs and the predicted value. As it has been already mentioned, the main reason is that the relations that have been established by the neural network are not linear, while the correlation only evaluate whether the variables are related by a linear relation. Figure 2 shows the "partial dependencies" which refers to an illustration (statistically computed) of the relations between each input parameters (x-axis) and the predicted value (y-axis). It also illustrates the relative importance of each parameter: a parameter with no influence would have a horizontal curve as a mean state but it is not a quantitative approach. Partial dependency plots should be interpreted with caution, it refers to the mean state of statistical computation, depends on a discretization choice and values of input parameters have been standardized (mean = 0 and Standard deviation=1). The Figure 2 presents 2 panels, one for Envisat calibration (left) and one for ERS-2 calibration (right). Nevertheless, we can say that curves are not linear, no input is unused or with a very low influence, we can also note that LES and PP have the largest influence on the predicted radar freeboard.

[Figure]

**Figure 2.** Partial dependencies plots for Envisat and ERS-2 calibration, from top left to top right, inputs are, the raw radar freeboard, the month, the pulse peakiness, the Leading edge slope, the concentration and the $f_{MYI}$.

The question of the non-linearity is central in our study but also in your analysis. But, since the correction is not linear at all, the largest raw radar freeboard is not necessarily the largest corrected freeboard, just as it is not the largest raw freeboard that will benefit from the largest correction. The raw radar freeboard, even noisy, still gives information on how the altimeter perceives the surface and how much it should be corrected, which remains an important information. We expect that the raw freeboard define the space and time variability of the calibrated radar freeboard over the whole period but this is hard to show since we don't have any reference of the expected variability of the SIT/FBr/FB during 1995-2010. To enhance the fact that the raw radar freeboard impacts the corrected radar freeboard, figure 3 shows the relative difference between the predicted FBr of the NN presented in the paper and one from a NN trained without the raw FBr for April 2011. It shows that for a large part of the basin, the difference of FBr is up to 25% of the predicted radar freeboard.

Finally, it's important to keep in mind that we have trained the neural network to reach the best score i.e. the best coefficient of determination (compared to CS-2 for Envisat and to Envisat corrected for ERS-2). Choosing the best NN, means choosing the combination of hyperparameters and even the choice of input parameters that gives the best scores. This means that the fraction of MYI allows to better fit CS-2 radar freeboard, that's why we keep it. However, it's even expected to find a good correlation between the sea ice age and the sea ice freeboard because, in average, older ice will be thicker.

[Figure]

**Figure 3.** Relative difference in % between the corrected freeboard of our study and one with the same NN trained without the raw freeboard

To sum up, the purpose of this paper is to retrieve a consistent radar freeboard estimation for ERS-2 using the current knowledge on LRM waveforms over sea ice. Because LRM waveforms are highly impacted by the surface state and poorly understood over sea ice, raw freeboard have to be calibrated. Two calibrations need to be implemented to get consistent ERS-2 radar freeboard, first Envisat against CS-2 and then ERS-2 against Envisat calibrated radar freeboard. The calibration is first based on surface roughness proxy because evident link have been emphasized with the size of the correction (previous studies) and secondly on auxiliary data that were used to reach better fit with CS-2. The calibrated radar freeboard is partly driven by the raw radar freeboard, both parameters are not linear correlated as it would say that the calibration did not perform well. The "age" or in our case the MYI fraction is not the key input for the NN training. Furthermore, 'ice with a higher-than-average raw FBr in a given month" can not necessarily "end up with a higher than average corrected FBr value" as the calibration is not linear.
* * *
Please find below the details on how your specific comments have been taken into account. We have split in two part the specific comments, as the referee gives two detailed comments, one as a community and one as a referee. *In this document, the referee's comments are in bold type, the answers are in italic type, and the corrections to the revised manuscript are in normal type.*

**Answers to Robbie Mallet (referee n°2) : specific comments**

**Specific comments - Referee comment**

**L280: I think you should by convention use the coefficient of determination rather than Pearson-r as a test score. Otherwise you'll end up with highly correlated relationships that have the wrong slope?**

*This is an error in the manuscript, we do use the coefficient of determination as the score for our regression. The correction has been made in the manuscript.*

**You need to explain quite a lot more about what's going on in Figure 6. The manuscript should not feature undefined**
**letters and symbols, and there are many in this figure.**

*Caption has been largely developed as following :*

Summary diagram of the uncertainty budget from along track to the propagation by the neural network.
**replaced by:**
Summary diagram of uncertainty budget during along track, gridding and calibration steps. Top left panel corresponds to the
along track to grid uncertainty budget. Top right panel defines the notations, for the Monte Carlo procedure : $\Omega$ for the Neural
Network input parameters, $\Gamma$ for the Neural Network output parameter (radar freeboard) with $\sigma_\Omega$ and $\sigma_\Gamma$, the corresponding
uncertainties. The middle panel corresponds to the training of M models with noisy inputs and outputs. Bottom panel show the
predictions of the N noisy input with the M neural network trained. $\gamma$ is the predicted radar freeboard estimation for one pixel
of the MxN predictions. M=100, N=200.

**Similar to above, you should explain much more about what's going on between lines 277 285. Papers in The Cryosphere
should be accessible to scientists without extensive experience in machine learning. Don't be afraid to use the supple-
ment for this, as I appreciate it's wordy. For instance, why did you choose 5 hidden layers and 100 neurons, and what
are the implications of your choice? Why a sigmoid? There are noticeably no references to support your choices, and**
**there's no element of later discussion about the impacts.**

*We have chosen a MLP because it has a very simple architecture but can deal with non-linear problem, which is the case of
this study. The choice of the architecture (number of layer and neurons per layer) as explained in the manuscript as been fixed
by testing a large amount of setup (called gridding) and choosing ones that have give the best score on the validation sample,
with a reasonable time of learning. Concerning the activation function, the sigmoid was chosen to allow negative FBr as it is*
*for target FBr and not to drop the value and bias the statistics of the predicted values. The sigmoid activation function was
chosen so that it could allow negative FBr values in order not to artificially drop the negative predicted FBr values.*

*Machine Learning is largely used for various application even for geosciences but settings architecture and hyperparameters
resides in testing testing and testing to get the best model with the best score. Citing study that use a MLP for geosciences will
not be relevant as the hyperparameters highly depends on the issue we want to deal with so it could even be wrong. The*
*paragraph you refer in your comment have been detailed to make it clearer. The implication of all choice is that by choosing
the Neural Network type, MLP, with trained the best model possible so to have the best prediction possible comparing to a
reference (CS-2 or Envisat calibrated).*

The neural network is a multilayer perceptron regressor (MLP) composed of 5 hidden layers, each composed of 100 neurons.
The activation function used is a sigmoid. Hyper-parameters have been tuned by dichotomy by choosing at each step the hyper-
parameter combination with the highest mean score (average score made on 5 models) on the test sample. The score used for
this regression is the Pearson correlation coefficient. To determine the most suitable hyper-parameter combination, the dataset
is randomly split into a training and a testing dataset, corresponding respectively to 90% and 10% of the initial dataset. To avoid
overfitting, we use early stopping to interrupt the training when the score is not improving anymore. Once the hyper-parameter
combination is set, the MLP is trained with the whole dataset. The NN trained is then applied to the LRM monthly grids to
obtain a monthly LRM-corrected radar freeboard.
**replaced by:**
The neural network used is a multilayer perceptron (MLP). Both calibrations have been processed with Scikit learn [Pe-
dregosa et al, 2011]. The MLP is composed of 5 hidden layers, each composed of 100 neurons. The choice of hyperparameters
: number of neurons, the learning rate, the regularization term, batch size, activation functions, solver for the weights opti-
mization, have been done using gridding methodology, e.g. testing combinations and take the one that give best score. The
evaluation criterion, called the score, is chosen as the determination coefficient. Models are trained on 90% of the dataset and tested on the remaining 10%, the splitting in random. During the tuning step, models are cross validated, it means that they are each trained 5 times with the same combination of hyperparameters but without the same train/test dataset, the 5 scores are then analyzed to determine the best combination. Cross validation give a better idea of the model performance as the dependence to
the training dataset is limited. The activation function for the hidden layers neurons is a sigmoid, motivated by possible negative radar freeboard values and the optimizer is and ADAM [Kigma et Ba, 2014]. Moreover, in order to avoid over-fitting, an early stopping criterion is used to stop the model training as soon as the score is not improved during 10 consecutive iterations, with a defined tolerance.

Finally, once the hyperparameters combination is set, the MLP is trained on the whole dataset to provide the calibration
function. The trained model is then applied to the LRM monthly grids to obtain a monthly LRM-corrected radar freeboard.
* * *
**I also have the view that 'radar freeboard' is not a geophysical quantity to be measured with an uncertainty. Instead it is precisely the retracked elevation of a waveform returning from sea ice, and is specific to a given radar's geometry and the chosen retracking algorithm. See the original definition in the supplement of Armitage Ridout 2015, and Tilling**
**et al. 2019 for how different radars will generate different Rfbs even if they could 'look at' the same ice. Similarly, different retrackers will generate different Rfbs when 'looking' at the same waveform, all of them valid and precise.**

**So I think you should change the phrase 'radar freeboard correction' to 'radar freeboard calibration', as you're not correcting some uncertain value. Instead you're calibrating the Rfb from one instrument so that it's consistent with another instrumental geometry. The same with 'radar freeboard estimation' - you're not estimating it: it's a precise**
**value resulting from the radar geometry and choice of retracker. I have a lot more to add on this issue, but it's quite philosophical/ subjective and I think we need to first focus on the issue concerning the representation of the TFMRA50 Rfbs in the 'corrected' product.**

*If we understood your comment correctly, we don't have the same point of view. The geophysical quantity we want to estimate is the freeboard. For that purpose, we measure the height over the floes and the height over the leads that are extrapolated below*
*the floes. The so-called radar freeboard is the difference between both heights. These two heights are subjects to uncertainties (like for all measurements) which are propagated to the difference. Among these uncertainties, we have the speckle noise, the interpretation of the retracking to estimate the range e.g. the retracking step. I can't find any definition of radar freeboard on Armitage and Ridout 2015's paper supplement more than 'The radar freeboard is then simply the retrieved elevation of the sea ice floe relative to this interpolated sea level' but it is also the definition we consider with an uncertain SLA and uncertain sea*
*ice floe elevation anomaly or what we call ILA (Ice Level Anomaly) in reference to SLA.*

*Note that "error" is used in the manuscript while dealing with speckle noise because Wingham et al 2006 used that terminology, but it refers to uncertainties, both words were often mistaken to qualify uncertainties until a few years ago. This has been clarified in the manuscript.*

*We do agree that the word 'corrected' is a bit confusing, as we also deal with uncertainties. Even though, an uncertain value*
*can not be corrected, at least the uncertainty can be reduced contrary to an error that is known and could be corrected. These two words have a different signification.*

*In order to clarify the reading, we suggest replacing corrected by calibrated while dealing with the predicted radar freeboard from the NN or the surface state bias corrected radar freeboard.*
* * *
**Specific comments - Community comment**

**L25) I would question whether "thin ice is more sensitive to climatic hazards". Bitz and Roe (2004) argued the opposite: that thick ice is thinning faster, because areas of thin ice grow more quickly in winter. Age products also show that thicker, older ice is disappearing from the Arctic and being replaced by thinner, seasonal ice (e.g. Nghiem et al., 2007). So I'm not sure it makes sense to say that thin ice is more sensitive to climatic hazards, when thin ice is coming to**
**dominate the Arctic and is more robust to temperature perturbations.**

*We do agree that because of the global warming, multi-year ice have started to disappear and be replaced (in area) the next winter by FYI. Thin ice thickness will be recovered more easily than thick ice if it melts. Nevertheless, the thinner the ice is, the faster it undergoes melting and breakup when temperature rise in late spring. It will be more supposed to break while occuring climate hazard such as cyclones or strong winds [Rheinlænder et al 2022]. During all seasons, thin ice will ridge, raft, diverge easier than thick ice [Stroeve et al 2018] so can highly affect sea ice area (and of course volume). We suggest the following modification :*

Thin ice is indeed more sensitive to climatic hazards than thicker ice but it especially enables to compute the volume.
**replaced by:**
Thick and old ice is disappearing and being replaced by younger, thin ice that has a higher mechanical sensitivity. Thin ice is more prone to deformation [Stroeve et al 2018] that induce area changes, and is more sensitive to climate hazards such as cyclones or strong winds [Rheinlænder et al 2022]. Thickness is a key parameter for sea ice study, it varies a lot according to the regions and it modulates the sea ice volume evolution in the Arctic ocean [Landy et al 2022].
* * *
**L32) I don't agree that it's "commonly accepted" that Ku-band radar waves penetrate the snow layer when it is sufficiently cold. I think that assumption is still up for discussion, and I would argue the opposite. I'm not aware of any in-situ or airborne CryoSat evaluation ever done over sea ice that has produced evidence that Ku-band radar waves consistently return from the snow-ice interface. For instance, neither of the airborne CryoVex 2006 and 2008 campaigns (Willatt et al. 2011) indicated that this was consistently the case over FYI. Results from a different radar system in Antarctica (Willatt et al. 2010) also showed that radar waves do not always return from the ice surface. Results from a third radar system deployed on MOSAiC (on SYI) indicate that more Ku- band power comes back from the snow surface than from the ice surface (Stroeve et al., 2020 Fig. 7; Nandan et al., 2022 Fig. 8). Garnier et al. (2022; Figure 9) shows results from CryoVex 2017 where the difference between Ka and Ku band ranging is at times negative, further casting doubt on the assumption. Moving to satellite-based evidence, Armitage and Ridout (2015) calculated CryoSat-2's penetration factor as 82%. Ricker et al. (2015) used buoys to show that snow accumulation caused increases in Rfb, not decreases (implying that the radar waves are not penetrating fully). This agrees with the work of Gregory et al., (2022; Figure 9) that shows that snowfall is correlated (not anti-correlated) with Rfb over both ice types. I would also argue that the often-cited work of Beaven et al. (1995) was not realistic – it featured snow that was shovelled, sifted through a screen, and then artificially smoothed at the surface by the weight of a metal plate before measurement. It is also striking that what the authors identify as the snow-ice interface appears at 20 cm range when it was 21 cm away in free space. Since it was 21 cm away in free space it should have appeared further away, at something like 25 cm in range due to the wave-propagation delay. There's no need to mention all this in your paper, but I wanted to briefly state my evidence before making the point that full Ku-band penetration is not a settled consensus, even for cold, dry snow. I think it would be fair to say that full penetration is "commonly assumed in satellite-based sea ice thickness products". But just because we're forced to assume it in our products doesn't mean the we should actually believe or accept the assumption.**

*We do agree that the knowledge of how far into the snow layer Ku-band radar waves can penetrate is still under deep discussion in the community. There is no possible consensus on the fact that signal penetration will depend on salinity, temperature, humidity, snow age and other parameters... However, it is important not to confuse the results of studies done in Antarctica with those done in the Arctic, just as it is important not to confuse the SAR results with the results of field studies, since the SAR treatment impacts the waveforms and does not only reflect the behavior of the ku-wave in snow. We suggest the following modification :*

It is commonly accepted that the Ku frequency penetrates the snow layer when it is sufficiently cold, in other situations this assumption can be questioned (Ricker et al., 2014; Nandan et al., 2017).
**replaced by:**

Implementing this method requires the assumption that the Ku-band radar wave completely penetrates the snow layer, which is still widely discussed and is not the subject of a definitive consensus (Ricker et al., 2014; Nandan et al., 2017).
* * *
**L381: Year of this citation is 1986.**

*We have taken into account this semantic shade in the manuscript.*

* * *
**L65: I think we're not really measuring sea ice thickness, but instead estimating it based on freeboard measurements (or radar-altimetry measurements). This might seem like a semantic point, but I think users of sea ice thickness products do benefit from this distinction. "estimates" rather than "measurements" is more commonly used by convention (e.g. Tilling et al., 2018, Kurtz et al., 2014; Landy et al., 2017).**

Increasing the along-track resolution of the aperture radar has led to considerable advances in the measurement of sea ice thickness.
     **replaced by:**
     Increasing the along-track resolution of the aperture radar has led to considerable advances in sea ice thickness estimation.
* * *
**L75: I think readers like me who aren't expert in roughness would benefit from a citation here. Is LRM definitely more impacted by a given roughness than SARM? I can believe it, but would like to read some evidence.**

*Kurtz et al 2014 pointed out that ice roughness or more generally, sea ice surface properties impact the waveform of return echoes. Such as a lot of remote sensing instruments, the illuminated area will impact your measurements. Concerning the difference of roughness impact between SARM and LRM range measurement, it's due to the acquisition processing itself.*
*Knowing how both work (see https://www.aviso.altimetry.fr/en/techniques/altimetry.html), the theoretical return power will be the same for both nadir and off-nadir in LRM whereas in SAR most of the return echo power will be concentrated to nadir, which reduces the impact of off-nadir and give the peaky shape to SAR waveforms and a lower impact of surface roughness [Raney et al 1998]. It was a bit more explicated in section 3.4. We propose to add this citation in the sentence you are mentioning and add a reference to the section where it is more explained.*

Contrary to SARM, LRM altimetry measurements are strongly impacted by the surface roughness of the surface illuminated by the radar, also affecting the freeboard measurement.
     **replaced by:**
     Because LRM altimetry has a larger footprint than SARM altimetry (by a factor 30), LRM range retrieval are significantly more impacted by surfaces roughness of the [Raney et al 1998] than the more nadir-focused measurement (SAR technologies).

* * *
**L103: I think you mean NSIDC 0611? This product gives the maximum of the ice age distribution in a grid cell at each timestep (see quote below). So I'm not sure how you've used these max values to generate an MYI fraction product? I think it could be done if you had access to the Lagrangian data, which is out there. But if you've used this I think you should state that. (Tschudi et al. (2020) states "This approach does not consider new ice that may form within a grid**
**cell because it retains only the oldest ice in its accounting. Thus, the product is effectively an estimate of the oldest ice in a given grid cell.")**

*The type is attributed to the 20hz along track measurements from the NSIDC age nested in two categories (whether the age is greater than 1 year). During data gridding, the type is also gridded and gives us an idea of the fraction of MYI by averaging the ice type into cells.*

L103-104 : This information comes from the NSIDC 0061 sea ice age product (Tschudi et al., 2019) that is aggregated into two classes (MYI and FYI).

**replaced by :**

The study also requires a sea ice type product, this information is derived from the NSIDC 0061 sea ice age product (Tschudi et al., 2019) that is aggregated into two classes (MYI and FYI) according to the age of the ice (FYI : ice age between 0 and 1
year, MYI : ice age of at least one-year) at a weekly frequency. Data are respectively available as daily and weekly map with a 12,5 km grid resolution. The fraction of MYI is derived from the ice type information during the gridding processing step.
* * *
**L115: I think at some point you should direct the reader to Kwok and Haas (2015), which discusses some key issues in the product that you've chosen.**

*This section aims to present the dataset not to discuss it, however, we added the reference to section results as following:*

The bias between OIB and Envisat estimation could also be attributed to the OIB snow depth which estimation seems sensitive to the algorithm used (Kwok and Haas, 2015; Kwok et al., 2017).
* * *
**L310: "Surface roughness is identified as the largest source of uncertainty" - I didn't really understand how you made**
**it to this conclusion. I think this is specifically a reference to Fig. 8 of Landy et al. (2020). The error in the sea ice roughness over FYI is 4cm, and the error from the snow basal salinity (just part of the "penetration bias") is 7 cm, and the uncertainty due to snow depth is 6 cm. So over FYI the roughness uncertainty is smaller than either the snow depth or the snow salinity. As such I don't think roughness can be reasonably characterised as "the largest source of uncertainty" over FYI based on Landy et al. 2020 Fig. 8. Over MYI the sea ice roughness uncertainty is equal**
**to the snow depth uncertainty, and admittedly larger than "partial snow penetration" uncertainty. So the statement is narrowly true if you only consider MYI and don't factor in the (highly related) uncertainty in snow depth in the comparison. But I think that only considering the largest source of uncertainty and ignoring the other uncertainties is a pretty risky strategy, given the other sources are comparable and perhaps actually larger in magnitude? If you are wedded to this approach, I think you should state that this will induce a pretty serious underestimate in your uncertainty**
**values (which is important info for product end-users).**

*This sentence is inexact, it has to be shaded to "surface roughness is identified as one of the largest sources of uncertainty". Nevertheless, the other sources of uncertainty while measuring the FBr, as summed up in Landy et al 2020, is the uncertainty due to SLA (off nadir and low density) and the limited Ku-band penetration in the snowpack (caused for instance by snow basal salinity for FYI or metamorphic snow for MYI) not the snow depth. The point that was not explained in the manuscript,*
*incorrectly, is that the uncertainties due to partial signal penetration in the snow are only indirectly taken into account, we don't ignore it. Indeed, it is not so trivial when comparing freeboards from different retrackers, to differentiate between roughness and penetration [Ricker et al 2014]. We believe that a "significant" part of the uncertainty on penetration is included in the uncertainty on roughness presented in [Landy et al 2020]. For this reason, we made the choice not to add values for the undefined limited penetration of the signal in the snowpack in this uncertainty budget. The problem of penetration is not*
*ignored, but the manuscript lack of information on this point. As contribution of sources are not defined, yes, it is possible that the final uncertainties are underestimated. We suggest the following modifications:*

Landy et al. (2020) decomposed it in two, the FBr systematic uncertainty budget, on the one hand, the uncertainties due to the penetration of the signal in the snow (depending on its salinity or if it is composed of metamorphic snow, according to the type of ice) and 310 in the other hand, the surface roughness. Surface roughness is identified as the largest source of uncertainty and we, therefore, choose to consider only this source in our systematic uncertainty evaluation. Roughness is estimated to be respectively about 20 % and 30% of the sea ice thickness for FYI and MYI (Landy et al., 2020). Note that this systematic uncertainty budget only concern CS-2 mission which are afterward propagated to Envisat and ERS-2, indeed other mission will be "corrected" from surface roughness effect during the calibration procedure.

**replaced by :**

In Landy et al 2020, the FBr systematic uncertainty budget is decomposed in two parts, on the one hand, the uncertainties due to the penetration of the signal in the snow (depending on its salinity or if it is composed of metamorphic snow, according to the type of ice) and in the other hand, the surface roughness. We assume, as in Ricker et al 2014, that the comparison of the freeboard from different retrackers does not enable to separate the contribution of the roughness from the signal partial penetration. We therefore assume to consider both sources as one mixed contribution, estimated to be respectively about 20 %

and 30% of the sea ice thickness for FYI and MYI (Landy et al 2020). The systematic uncertainties can be underestimated as the penetration of the radar waves in the snow uncertainty may be poorly handled. Note that this systematic uncertainty budget only concerns CS-2 mission which is afterward propagated to Envisat and ERS-2, indeed other missions will be "calibrated" from surface roughness effect during the calibration procedure.
* * *
**Fig. 6: I see in the top panel that you've "summed the squares", which has the implicit assumption that uncertainties that you have considered are uncorrelated. It may be that you have good evidence to support this that I'm ignorant of, but it seems, for instance, that speckle noise may well be (anti?)correlated with surface roughness? Just as an example. I think that the omitted snow uncertainties involving penetration & depth are more likely than not to be correlated in some way. I think you should state that you've assumed the uncertainties are uncorrelated in your analysis, and give**

**the reader some information as to what the results of that assumption may be.**

*Yes, we assumed the uncertainty due to the speckle noise and the SLA uncertainty to be uncorrelated, as it is precised l 304 in the manuscript. The speckle noise will not be correlated to the surface roughness but it is attributed to the surface asperities which are of the order of magnitude of the wavelength of the signal, about 2 cm, that causes interference in the signal. But a surface can have several roughness scales, for instance MYI highly rough can have asperities of about 2cm as well as newly*

*formed sea ice, both will present speckle noise that induce the same uncertainty on the range.*

*By construction, systematic and random uncertainties are not correlated, so this is not really an assumption. Concerting the uncertainties due to snow penetration, we redirect you to the previous comment (and here the snow depth is not considered, so its uncertainty either).*
* * *
**Figs. 7 & 8: These are really well designed and presented**

*Thank you for this comment.*
* * *
**L381: 4) Why take snow density as constant? SnowModel-LG outputs depth and density, and includes some physics of densification/settling over time. So I think it's odd to use one of its variables and not the other, since they're so linked in**

**the model. Snow impacts thickness retrievals by weighing the floe down and slowing radar waves: both of these effects are proportional to the mass of overlying snow – not the depth (see Mallett et al., 2021). So I think it makes a lot more sense to use both the depth and density (the SWE) in your thickness retrievals rather than just the depth. Here's a plot of the seasonal densification of SnowModel-LG snow north of 88N for the period 1995-2018. You'll see that as well as being more dense than your assumption, it also evolves over the season.**

*As suggested, we have done the scatter plot using the snow density from the model, figures have been updated in the manuscript. Nevertheless, it was not possible for CanCoast due to SnowModel-LG output coverage, and as we take OIB snow depth for OIB/Envisat SIT conversion, we have chosen to keep a constant density to keep consistency. It is interesting to see that the comparisons look really similar than with a constant snow density. Comparisons with moorings even give worse results with higher biases than using constant density. You can find the statistics in figure 4, 5, 6 and 7.*

___________________________________

**Fig 13: I'm a little unclear what the radar freeboard timeseries is supposed to represent. I imagine it mostly reflects the trend and variability in sea ice extent, and I think you should point this out to the reader. A simple correlation with SIE would quantify this relationship and reveal if the quantity is useful. For the part of it that doesn't represent SIE, would decreasing volume reflect a thinning of sea ice? Thinning snow (Webster et al., 2014) will mask the effect of thinning ice**
**on the Rfb. In areas where the snow is really thinning quickly, the Rfb could potentially even increase even if the ice is thinning. I guess I would like to see a little interpretation of this quantity figure 13 rather than being left to do it as the reader.**

    *The time series present the radar freeboard volume we have computed during the ERS-2, Envisat and CryoSat-2 period. We chose to show the evolution of the volume of radar freeboard instead of mean radar freeboard as it is more difficult to interpret*
*as the mean freeboard depend on the number of pixel covered by ice and is not necessary representative of the global ice state because low concentration area have the same weight as compact ice area. The motivation of computing the volume is to represent better this evolution. according to [Landy et al 2022], figure 8, the anomaly in volume are mainly driven by thickness anomalies and not area for the Arctic, so this would not reflect the variability of sea ice area. Concerning the impact of snow on trend, you are right, it would change the trend as snow load have changed during the past 30 years, but it will give the same*
*trend as using a snow depth climatology to derive the volume as it has been done in the majority of the previous studies. We suggest adding the following precision:*

    The evolution of the snow load is not taken into account in Figure 13, which means that the evolution of the volume is not fully represented, in the same way as if the total volume were derived with a snow depth climatology. Indeed, a decrease in FBr volume may merely indicate that the snow depth is greater and the ice thickness unchanged.

___________________________________

**L450: I think you should state the limitations in your uncertainties here. In particular (and I think this is key), do the "observed" thicknesses fall within your uncertainty bounds? If not, then either your uncertainty bounds are wrong or the validation data is wrong. I think uncertainty bounds on retrievals are not useful unless you can show that observed data fall within them.**

*Thank you for this suggestion, uncertainties of satellite estimates have been added to validation plots for the review. Nevertheless, as explained in the manuscript section results, conclusion are not as simple to draw out, validation or retrieval are not necessarily wrong if validation dataset don't fall within uncertainties bounds. First, because validation data also present uncertainties but also because validation procedure of monthly satellite estimation assume that we observe in average the same sea ice surface than the validation do and that can be questioned for airborne or submarines dataset for instance.*
*Figures 4,5,6 and 7 present the 95% confidence interval of the SIT but without taking into account uncertainties on snow depth, densities etc for the FBr to SIT conversion step.*

    *The plots have been updated with the variable snow density. For esthetical reason, bounds are not represented for comparisons with other satellite-based SIT estimation.*
* * *
[Figure]

[revised manuscript text omitted]

---

## Referee Report (RR1)

This is a review of "Arctic sea ice radar freeboard retrieval from ERS-2 using altimetry: Toward sea ice thickness observation from 1995 to 2021"

I should start with a quick self-correction: I previously misidentified Armitage and Ridout (2015) as having provided an "original definition" of the term *radar freeboard*. This was a mistake, as it was used prior to that for CryoSat-2 by Ricker et al. (2014), before that in an airborne context (Hendricks et al. 2010[1]), and even before that in non-peer-reviewed work.

**Synopsis**

While the authors have made a number of useful clarifying additions to their manuscript, I'm afraid to say my fundamental concern about the relative size and nature of their adjustment[2] remains unresolved.

I still don't agree that the final data truly represent a "radar freeboard retrieval" (per the title), when they are so loosely related to the process of waveform retracking, and so dominated by other signals from ancillary data used in the adjustment process. So I still believe that the final product would be more accurately described as a *modelled* or *proxy* radar freeboard data set, given the second order influence that retracked elevations seem to have in the final result.

In their manuscript, the authors have not addressed or even mentioned my point that most of the work done by the adjustment is to oppose the TFMRA radar freeboard value. This is not how most calibration and adjustment workflows behave in my experience. I am confident that potential TC readers will not appreciate or understand this important feature of the method if the manuscript is published in its current form.

[Figure]

In my view, a calibration-based adjustment should be fairly *small* relative to the original and final values, and not be comparable in magnitude and opposite in sign to the value undergoing adjustment.

At the end of this review I have indicated some changes that I think should be implemented before this manuscript can be published.

All quoted line numbers that follow refer to the revised manuscript, not the "track-changes" manuscript, which I thank the authors for providing.
* * *
1    Hendricks, Stefan, et al. "Effects of surface roughness on sea ice freeboard retrieval with an Airborne Ku-Band SAR radar altimeter." 2010 IEEE International Geoscience and Remote Sensing Symposium. IEEE, 2010.

2    I'm using "adjustment" rather than "calibration" here, see my point below in **Minor Comments**

**Technical Comments**

Font sizes of axis labels on figure 3 should be equalised.

Double parentheses on L216

Need to define "ASA" and include it in your list of acronyms.

**Minor comments**

Given my concerns about whether the presented neural network can be reasonably described as a "calibration", I should point out that *calibration* actually refers to the comparison of one data set to a standard, and does not refer to subsequent adjustment of the initial data set to match the standard. Really what is being done by the NN in prediction mode (after training) is *adjustment.*

*https://calibrationselect.co.uk/general/calibration-or-adjustment/*
*https://soluzionesolare.com/news/difference-between-calibration-and-adjustment/*
*https://allometrics.com/what-is-the-difference-between-calibration-adjustment/*

I appreciate that *calibration* is sometimes uses wrongly to refer to adjustment. I've done this, and there are several instances of this in the remote sensing literature. But the BIPM is clear in its most recent glossary[3] that calibration should not be confused or used interchangeably with adjustment.

The last paragraph of Section 1 actually uses "adjust" correctly three times, but elsewhere the process of adjustment is referred to as calibration.

I think by using the proper terms both here and in the manuscript we would be able to better discuss the performance of the blur-corrected TFMRA50 Rfbs in the calibration procedure, and the relatively large magnitude of the subsequent adjustment required.

L377-378, When quoting the standard deviation and bias between data like this it would be best to see them as scatter plots, with the y=x line superimposed. Especially as these numbers appear in your abstract.

L107: The authors have not meaningfully responded to my previous comment about the MYI fraction product they have constructed. Trivially, they have again misidentified the NSIDC ice age dataset as "0061", when it is "0611". More importantly, they have not addressed my other point that 0611 is not a simple map of ice age, but a map of *the oldest ice contained in a given grid cell* (see Tschudi et al. 2020 and my previous review). At the moment they have treated grid cells containing any MYI at all as being made up of 100% MYI (see figure 10), which will undoubtedly introduce a high-bias into the MYI fraction product that they construct. I would urge the authors to go back and engage with my previous comment on this matter.

L80: I don't think the change that you've put in your review responses has been included in the resubmitted manuscript? The text reads the same as before. I'm also still not convinced that SARM is less sensitive to *a given roughness* than LRM, as implied. I understand that the footprint is smaller, so potentially the footprint contains less variability in surface height (by perhaps including fewer floes of distinct freeboards). But isn't the point of altimetry to characterise some kind of average height? I can't see how a ridge in the footprint is less of a problem for a SAR waveform
* * *
3   International Vocabulary of Metrology (2021) International Bureau of Weights and Measures; see p39 for definition of *calibration*. Indeed both the entries for *adjustment* and *calibration* include notes stressing that the two concepts should not be confused or used interchangeably. I note that the definition in this draft publication is essentially unchanged from the previous 2008 publication.

than it is for an LRM waveform. I'm not saying that the authors' claim isn't true – but a clear underlying mechanism and relevant reference should be provided.

L306: "Moreover, in order to avoid over-fitting, an early stopping criterion is used to stop the model training as soon as the score is not improved during 10 consecutive iterations, with a defined tolerance." I don't see how this would stop overfitting? You could easily have overfitted your model by the time that you fail to get an improvement in score, because overfitting can result from spurious improvements to the score that don't reflect improvements in the underlying model. I think overfitting is normally identified through comparison of performance on test sets with training sets.

In fact, I would suggest this model may well suffer from overfitting, due to the *very* low ratio of bias to variance (3mm to 9 cm in the Envisat-CS2 evaluation, 2mm and 3.8 cm for ERS/Envisat). This ratio is a well known indicator of overfitting, and suggests your model has captured the noise in its training data as well as the underlying signal.[4] You should address this in your discussion.

L300: If I understand right, you've specified a range of discrete hyperparameters and then investigated which combination optimises the score. But this description should be contextualised with the ranges and intervals over which you searched, the dimensionality of the hyperparameter space that was searched, and most importantly the results of the search. You've made choices about the learning rate, regularization term and weights-solver based on your grid search, but you haven't reported what those choices were. Given you performed what must have been a highly multidimensional and computationally intensive score-sensitivity analysis to get at these specific hyperparameters, it would be great to know what they are. I'm also not totally clear how the activation function was both found through this grid-search, and also motivated by the domain of the TFMRA Rfbs. I guess the domain of possible activation functions was motivated by the domain of Rfbs, and then you ended up selecting the sigmoid specifically through the grid search method?

L429: Replace "seems" with "is". It definitely is sensitive to the algorithm used.

L34 of the manuscript now reads "Sea ice thickness estimation by spatial altimetry was first introduced by Laxon (1994) and Peacock and Laxon (2004) based on the freeboard methodology".
   I think the first real description of sea ice thickness estimation from radar altimetry derived freeboards was by Stanley et al. (1979)[5]. Their paper compared elevations from repeated tracks of the GEOS-3 altimeter in both the presence and absence of sea ice. They explained that the differences between the ocean elevations and the sea ice elevations (which were too large in their investigation due to poor gain control) contained information on the freeboard, which in turn could be used to estimate thickness if better constrained.
   The breakthrough of Laxon's work with ERS1 was the individual classification of waveforms into lead/floe, allowing effective interpolation of the local sea surface height – this unlocked the method. So I think to make Laxon (2004) the valid citation, you should just add something like "in its modern form" to your sentence.

I can't see why a non-geophysical parameter such as "date" is being used to adjust the TFMRA waveforms (Figure 5). What physical mechanism could justify its inclusion in the neural network? It seems to me that the inclusion of a date parameter will only make the neural network behave like a seasonal climatology (further reducing the ratio of bias against variability); this would evaluate very well against in-situ sources in the way that you've constructed your tests (given the dominance of seasonal variability over interannual variability in sea ice thickness), but would potentially be an example of spurious overfitting. At minimum, a bit of justification needs to be given for its inclusion, preferably with some information about the final effect of its inclusion.
* * *
4 Some nice discussion of the relationship between bias-variance tradeoff and overfitting here
  https://towardsdatascience.com/understanding-the-bias-variance-tradeoff-165e6942b229

5 Stanely, R.H., Brooks, R.L and Brown, G.S: Ice Freeboard Determination by Satellite Altimetry (1979) Proceedings of the International Workshop on Remote Estimation of Sea Ice Thickness, St. John's, Newfoundland, Canada.

I think the language surrounding the pulse-blurring correction needs a bit of clarifying. The reader should be in no doubt that a height correction has been applied to the retracked elevations to remove the effects of pulse blurring. The waveforms have not been "deblurred" and then retracked. I think the authors use of "corrected" is good, and I would suggest removing references to "deblurred" and "deblurring".

On the pulse-blurring correction – have the authors checked to what degree this actually improves their radar freeboard estimates after the NN-based adjustment? Seems like this would be a fairly significant finding, and would justify both their efforts and their phrasing of it as a "correction".

L41: I think a better reference for snow penetration is potentially Ricker et al. (2015)[6], which is entirely focussed on the topic.

**Major Comments**

I think the authors have misinterpreted my point regarding the correlations between sea ice age and their adjusted product. My point was that their adjusted product is more strongly determined by sea ice age than it is by the value actually being adjusted. The TFMRA50 radar freeboards are not the first order component of the adjusted values, and in fact contribute relatively little compared to a well-known proxy variable. To what extent, therefore, can this meaningfully be considered a calibration and adjustment exercise? Instead, it looks like more predictive variables such as PP, MYI fraction and LES are themselves being slightly adjusted using TFMRA50 data. I would have preferred to do my original analysis based on PP or LES, but I didn't have that data to hand so just used sea ice age which was easily downloadable.

Put another way – if you're constructing a radar freeboard data set, I think readers and users will expect the TFMRA50 radar freeboard to tell you more about your final result than the MYI fraction does, and indeed to be the first-order determinant. That's not to say that MYI fraction shouldn't tell you about the radar freeboard – there are obvious links. It's just to say that you'd hope the TFMRA radar freeboard would tell you more, *especially* as you've spent so long and worked so hard correcting the data for blurring.

Turning to the rebuttal, I still think that the relationships between parameters such as pulse peakiness and the adjustment value can be properly characterised even though they are non-linear. I think the authors should scatter-plot or point-density-plot the adjustment size and final data against (a) input radar freeboard (b) input pulse peakiness (c) MYI fraction (d) LES (e) date, and see which input has the clearest and strongest relationship to the output (regardless of the linearity of that relationship).

If I am interpreting Figure 3 correctly, then I conclude that the radar freeboard goes from having a limited role in the training of the neural network, to having a 25% influence on the training near the ice edge. However my concern is really that the input Rfb seems to have relatively little influence on the output Rfb in the *prediction* phase.

While the authors have given a lengthy rebuttal to my concern about the size and nature of the adjustment, they have made relatively few changes to the manuscript in this regard. I was expecting at least one or two plots to be added to the manuscript (or a supplement) quantifying and explaining how the different input parameters affect the adjustment values and the final results.
* * *
6 Ricker, Robert, et al. "Impact of snow accumulation on CryoSat-2 range retrievals over Arctic sea ice: An observational approach with buoy data." *Geophysical Research Letters* 42.11 (2015): 4447-4455.

**Summary**

My opinion is that the adjustments presented here is so large relative to the value being adjusted, that the final value cannot reasonably be presented as a calibrated and adjusted "retrieval". This is particularly the case given the adjustment so often acts to counter the value being adjusted. I see this substantial body of work as a useful and interesting exercise in modelling a CryoSat-2-like radar freeboard value based on date, pulse peakiness, MYI fraction and leading-edge slope data, with a relatively small input from retracking waveforms. In particular, the work to resolve pulse-blurring issues is valuable.

Most importantly, I think *users* of this product (perhaps from the field of modelling or seasonal forecasting) would very likely mistake these data as being primarily derived from retracking radar waveforms to retrieve elevations. When actually they are primarily produced by a neural network which assimilates several other, quite diverse parameters.

If this manuscript is to be published in TC, I believe it must undergo some major revisions and additions as follows:

1) A proper exploration of the impact that each parameter in Figure 5 has on the output. I appreciate that an effort has already been made to do this via the "partial dependence" figure in the reviewer replies, but it's unclear to me how this was constructed and what it represents. I think it would be much clearer to scatter-plot or point-density plot (a) the size of the calibration against the different input variables, and (b) the final result against the input variables. Without this, the machine learning approach is simply the deployment of a "black box" machine.

[Figure]

At absolute minimum, the authors should explicitly quantify and report the apparently loose correspondence between the TFMRA radar freeboards and the final radar freeboard product.

2) Show that their blur-corrected TFMRA50 Rfb's from ERS2 are related in at least some way to TFMRA50 Rfbs from Envisat in the overlap period.

3) Show that the influence of TFMRA50 Rfbs in the NN's adjustment improves the match between their ERS2 results and the EnviSat radar freeboards. At the moment, it seems that most of the work done by the adjustment is in *counteracting* the influence of the Rfbs. This makes me think you might be better off just straight-up modelling the radar freeboard based on your ancillary data sets alone.

I see that you've already thought about this with the map in Figure 3 of the reviewer replies – but this only shows the difference that's caused by omitting the TFMRA50 radar freeboards in the training. It doesn't show that the incorporation of TFMRA50 radar freeboards in the adjustment actually improves the situation at all (just that it makes a difference). To summarise points 2 & 3, I still have a suspicion that TFMRA50 retracking and blur-correction of ERS2 waveforms doesn't produce meaningful information about the sea ice, and thus doesn't contribute to your neural network's success in matching Envisat data sin the ERS/Envisat overlap period.

---

## Referee Report (RR2)

**Second review for The Cryosphere https://doi.org/10.5194/egusphere-2022-214**

**"Arctic sea ice radar freeboard retrieval from ERS-2 using altimetry: Toward sea ice thickness observation from 1995 to 2021." Bocquet et al., 2022**

The authors have taken my comments into consideration and made changes to their manuscript which improve the clarity and presentation.

Another reviewer, Robbie Mallett, spent some time investigating the dataset provided by the authors (which I did not) and identified concerns with the presentation of the results as a "reconciled time series of sea ice radar freeboard". His analysis appeared to show that the corrected ERS2 radar freeboards reflected little of the variability of the underlying raw retracked ERS2 freeboards and the correction could be drawing more information from other inputs to the neural network, such as the ice age or ERS waveform parameters. These concerns were echoed by the third reviewer, who submitted their review after seeing Dr. Mallett's analysis.

In response to Dr. Mallett's concerns, the authors argue that the lack of covariance between sea ice radar freeboards obtained from the same processing chain (TFMRA50) applied to different radar altimeter missions was a known motivation of their study. The different sensing geometries of the altimeters encourage varied impacts of surface roughness, snow properties, etc, on the backscattering of the surface and consequently the retracked heights of floes and leads. Therefore, the correction obtained from the NN is highly nonlinear, so there should be no simple linear correlation expected between raw and corrected freeboards. The partial dependency plot (Fig 2 of the review) they provide illustrates that the corrected freeboards – at least their mean patterns – are sensitive to all the input parameters of the NN, including the raw radar freeboards. However, there are several other parameters, such as the leading-edge slope of the waveform, that show stronger dependency than the raw freeboards.

In general, my view is that the corrected freeboard time series is only as good as its evaluation against independent sea ice freeboard/draft/thickness observations. This is the case for all satellite sea ice freeboard and thickness products, including single sensor products, calibrated multi-sensor records, and proxies for thickness derived from other datasets related indirectly to the ice thickness. In my original reading of the manuscript, I paid careful attention to the Validation section 4.2 and felt that the assessment of Envisat and ERS2 freeboards against various independent datasets was convincing enough to back up the main conclusion of the paper. I.e., that the time series of pan-Arctic radar freeboard volume in Fig 13 has had a significant negative trend since the 1990s. The biases wrt reference observations in Fig 12 are affected by snow loading assumptions but, if anything, imply the negative rfbv trend could be even steeper. However, results from the analysis by Dr. Mallett question whether the full spatial and temporal patterns of the radar freeboard anomalies in corrected grids from ERS2 and Envisat should be trusted.

I would recommend the authors carefully reframe their manuscript to highlight the issues with their time series raised by Dr. Mallett's analysis while emphasizing what they think can be gained from it and why. What should the gridded record of radar freeboard be used for and what do they think it shouldn't be used for by the community, in future studies? Along these lines, I would suggest the authors consider renaming their product a radar freeboard "proxy" from ERS-2 and Envisat (ideally a CryoSat-2 radar freeboard proxy – which relates to another comment by Dr. Mallett on the definition of radar freeboard – but this is another matter..). This change would alleviate the problem with a correction/calibration that is larger in magnitude than the raw freeboard and provide a strong caveat to prevent 'misuse' by the research community when the product is made available. There is absolutely nothing wrong with a freeboard proxy and, in my view, will not diminish the impact of the

obtained climate-relevant trend. (n.b. I also like the addition of the green 'climatology freeboard' line on Fig 13, which reinforces the importance of freeboard/thickness compared to ice area)

I also have a few more comments on points that came up during the review and response:

- Radar freeboard as a geophysical quantity with an uncertainty or not.
  I think this depends entirely on the definition of the radar freeboard, and it has not been consistently defined across different studies, even if some have tried. As soon as the radar freeboard is assumed to represent the mean height of the snow-ice interface within the radar footprint, with some displacement caused by unknown wave propagation delay in snow, then it is a measurable geophysical quantity with a characterizable uncertainty. However, if you do not make geophysical assumptions on the backscattering height represented by the radar freeboard, then it does not have a geophysical uncertainty. It is not necessarily a precise (i.e., repeatable) value though; this depends on the precision of the retracking algorithm which, for waveform fitting methods, may not be insignificant.

- Ideally, we will develop more sophisticated retrackers that can accurately account for sensor-related and mode differences between missions, without needing to statistically calibrate satellite records to one another. This *may* be sufficient to raise the signal above the noise and produce a robust 25-year time series of gridded radar freeboard observations with trustworthy spatial and temporal variations. (It may not, of course, if the 'target'-related noise is too high between sensors, not just the sensor-related noise). However, I consider this calibrated proxy record of CS2 radar freeboard to still be valid within the degree of uncertainty constrained by the validation exercise.

- On the analysis of 20-30% uncertainties in freeboard caused by roughness and whether they are separable from partial penetration.
  These values come from the application of different retrackers to the same CS2 SAR/SARIn mode data, so in reality it is quite an arbitrary representation of the range/freeboard uncertainty, affected by the assumptions and subjective decisions of those applying each retracking algorithm. This uncertainty spread is attributed to roughness because the roughness has a first-order impact on the variability of the retracked range: a larger roughness produces a larger difference between retrackers. The authors are correct that part of this variability in range between retrackers comes from the fact we are uncertain about the radar penetration in snow – is our basic assumption of full snowpack penetration correct? However, this is challenging to evaluate on a pan-Arctic seasonal basis without independent reference observations (although note: Nab et al, GRL, 2023). The most conservative approach would be to independently consider roughness, partial penetration, and SLA-related errors on the freeboard (assuming the final radar freeboard = the ice freeboard), but this will likely overestimate the total uncertainty if numbers from Landy et al 2020 Fig 8 are used.

Unfortunately, I do not have time to provide another thorough review of the revised manuscript with minor/grammatical suggestions. So, I ask the authors to consider my general comments here when reframing the paper. Congratulations again on a really valuable study – it is critical we work towards a long-term record of Arctic sea ice thickness from multi-sensor altimetry that will constrain the thinning(?) trend on a climate-relevant timescale.

Feel free to get in touch if you have any questions, Jack Landy

---

## Referee Report (RR3)

Firstly, I thank the authors for their considerable efforts to improve the manuscript. I appreciate this takes a great deal of time and can feel frustrating. However the manuscript has clearly been improved in my view.

The authors are correct to identify the following:

"The reviewer's main concern is whether our results are mainly driven by TFMRA50 radar freeboard or derived altimetry parameters or by auxiliary measurements"

Indeed this is my only remaining concern; I am very satisfied with the authors responses to my other comments.

However I maintain that this concern must be addressed before publication, since the authors describe their product as a timeseries of radar freeboard. In its current form, this product will be interpreted by those both within and outside the radar altimetry community as a product that primarily reflects the difference in retracked heights between floes and leads (what people know as radar freeboard).

For instance the authors have now added the line: "This radar freeboard time series product based on CryoSat-2 estimations intends to provide a record of monthly sea ice changes over the last three decades and for climate studies". This description clearly implies that "radar freeboard" is the main ingredient or at least the largest ingredient in the product.

If the authors describe their product as such, they simply cannot avoid examining the relationship between their corrected product and the actual radar freeboard (the difference in retracked height between floes and leads). If the corrected values are not clearly related, then their description is misleading in my opinion.

The authors have now nicely addressed this question for Envisat (Figure A2) and I thank them for this. I believe figure A2 offers great insight into the relative impacts of the different data sources on their final product during the Envisat period. The lower panel (included below) indicates to me that there is a positive but weak relationship (r = 0.253) between retracked height above interpolated sea level (the radar freeboard) and their corrected product. While this is low, I think it is definitely acceptable.

However I can't see why they have not presented this relationship for ERS-2, which is the main subject of both their product and the concerns in my review. In my opinion the authors must submit this key piece of information for review. The analogous version of Figure A2 would illustrate it perfectly.

If there is no clear relationship in the lower panel for the ERS version of Figure A2 (shown below for Envisat), it would not be reasonable to present this as a "radar freeboard product".

If however there is a clear relationship, then my concerns will have been unfounded. In this case, it would indeed be entirely reasonable to describe the product presented here as a "corrected radar freeboard product".

If the editor can determine that the corresponding figure (inset right) for ERS2 has a similarly strong relationship (or stronger!), then I am satisfied for a manuscript **with that figure included** to be published without further review by me. In my view, no change in title would be necessary in this case.

[Figure]

If the relationship is not clear or significant, then the authors should reconsider framing this as a "proxy" product or similar as previously suggested, and I would like to review again. In the absence of a clear relationship between radar freeboard and "corrected radar freeboard" in the ERS-2 period, framing this product as a radar freeboard timeseries would be misleading.

I thank the authors for their time on this,
Robbie

---

## Author Response (AR2)

Manuscript ID: egusphere-2022-214

**Title: Arctic sea ice radar freeboard retrieval from ERS-2 using altimetry : Toward sea ice thickness observation from 1995 to 2021**

*Marion Bocquet, Sara Fleury, Fanny Piras, Eero Rinne, Heidi Sallila, Florent Garnier, and Frédérique Rémy*

We would like to thank again all reviewers as well as the editor for their relevant comments and for improving the quality of the manuscript. To answer to the major concern, we started our answer with a note as well as a new title suggestion.

**General note concerning the manuscript objectives and new title suggestion :**

The numerous questions raised by reviewer 1, and repeated in this 2nd review, lead us to believe that there has been a misinterpretation of our objectives.

Indeed, in this study, we intend to reconstruct a first homogeneous series of sea ice radar freeboard over the period covered by ERS-2, Envisat and CryoSat-2, by relying on the CS-2 SAR measurements. Indeed, CS-2 radar freeboard is the only one

15 consistent for now and with the present knowledge of LRM over sea ice.

This work is in direct continuity of Guerreiro et al. 2017, Paul et al. 2018 and Tilling et al. 2019, all three of whom have worked in the past to extend the CS2 measurements by calibrating the Envisat LRM measurements on CS2 using one or two auxiliary data (PP for Guerreiro, LEW and Sig0 for Paul, distance between leads and floes for Tilling). They have obtained first series that are references, and Paul's version has even been adopted for SI-CCI.

20 Our objective was to extend this homogeneous series with CS2 back to ERS-2. The remaining discrepancies between CS2 and Envisat in the previous products, prevented us from considering Envisat as a new CS2 reference to extend the series to its predecessors ERS-21. We have therefore developed a method based on NNs that allows us i) to generalize the number of input parameters and ii) to take into account non-linear correlations between them. This allows us to effectively combine altimetry-bases parameters (FB TFMRA50, LES, PP), but also auxiliary dataset (ice type ratio, ice concentration, month of the year).

25 Our solution is not only uniform with respect to CS-2, but it also improves the agreement with available in-situ measurements. The same approach was then extended to ERS-2, whose results were also validated with in-situ measurements.

Main concerns from the review are not related to the homogeneity of the series obtained, nor their physical representativeness, which are the intended objectives of this work. The reviewer's main concern is whether our results are mainly driven by TFMRA50 radar freeboard or derived altimetry parameters or by auxiliary measurements. The question is indeed interesting,

30 and we attempted to address this concern, but this would be the subject of a proper study and unfortunately not the purpose of this paper.

Our concern come from the fact that same questions were repeated in the 2nd review with new analysis requests. We have once again responded to these questions in this new report, but we claim that the work requested deviate from the main purpose of the paper.

35 Indeed, we have never claimed that this series is based exclusively on altimetry data, especially since it is obviously impossible to reconstitute SAR measurements using LRM altimeter only (either SAR progresses would have been useless). Nevertheless, it can be noted that: 1. Three of the six used parameters are based on waveforms characteristics, and we had made many preliminary tests to check that each of these 6 parameters improves the solution. Even if these additional parameters would have been sufficient on their own (which would have been surprising), this would not have invalidated our objectives.

40 2. More importantly, CS-2 radar freeboard product is a pure based altimetry product, and our objective is really to reconstitute this radar-freeboard using Envisat and ERS-2.

Thus the radar freeboard estimation we propose, even for ERS-2 is mainly based on CS-2 one and less on TFMRA50 FBr from ERS2.

Nevertheless, we agree that the title "Arctic sea ice radar freeboard retrieval from ERS-2 using altimetry : Toward sea ice thickness observation from 1995 to 2021" is ambiguous from this point of view. We therefore propose to change it to **«A first homogeneous Arctic sea ice radar freeboard from ERS-2, Envisat and CryoSat-2 : Toward sea ice thickness observation from 1995 to 2021"**. This title better explains our objectives and avoids associating directly and exclusively the notion of radar freeboard to ERS-2. (We have not yet changed the title within the manuscript and in TC to avoid changing it several times if reviewers and editor do not find it relevant).

*In this document, the referee's comments are in bold type, the answers are in italic type, and the corrections to the revised manuscript are in normal type.*

**Answers to report #1**

**Minor comments**

**Given my concerns about whether the presented neural network can be reasonably described as a "calibration", I should point out that calibration actually refers to the comparison of one data set to a standard, and does not refer to subsequent adjustment of the initial data set to match the standard. Really what is being done by the NN in prediction mode (after training) is adjustment. I appreciate that calibration is sometimes uses wrongly to refer to adjustment. I've done this, and there are several instances of this in the remote sensing literature. But the BIPM is clear in its most recent glossary 3 that calibration should not be confused or used interchangeably with adjustment. The last paragraph of Section 1 actually uses "adjust" correctly three times, but elsewhere the process of adjustment is referred to as calibration. I think by using the proper terms both here and in the manuscript, we would be able to better discuss the performance of the blur-corrected TFMRA50 Rfbs in the calibration procedure, and the relatively large magnitude of the subsequent adjustment required.**

*In the first version of the manuscript, the term 'correction' was used, which we replaced with "calibration" according to your suggestion. Your new review suggests to replace 'calibration' by 'adjustment' following the definition of 'International Vocabulary of Metrology'. As you noticed, 'calibration' is not adapted according to this reference, as it doesn't properly characterize the procedure we developed. However, this change has been operated according to you advise. We suggest keeping the initial term 'correction' that seems to be right and keep consistency with former studies (Guerreiro et al 2017, Tilling et al 2019) that deal with Envisat LRM TFMRA issue, even if it is a more general term. As it is a way to correct a systematic error on the radar freeboard evaluated thanks to CS-2 and due to surface roughness and signal penetration. To be clearer in the manuscript, we suggest writing "NN FBr" while dealing with radar freeboards that have been corrected based on the Neural Network.*

**L377-378 : When quoting the standard deviation bias between data like this, it would be best to see them as scatter plots, with the line superimposed. Especially as these numbers appear in your abstract.**

*It is not always easy to choose the most adequate illustration, however in this case what we want to show is that the radar freeboard distribution has changed. It also allows to show the distribution of the residual difference between the two missions. As indicated, we still provide the main statistics, thus the scatter would not provide mush more information. We prefer to keep the histograms and not to overload the manuscript with other figure and scatter plots.*
* * *
**L107: The authors have not meaningfully responded to my previous comment about the MYI fraction product they have constructed. Trivially, they have again misidentified the NSIDC ice age dataset as "0061", when it is "0611". More importantly, they have not addressed my other point that 0611 is not a simple map of ice age, but a map of the oldest**

**ice contained in a given grid cell (see Tschudi et al. 2020 and my previous review). At the moment, they have treated grid cells containing any MYI at all as being made up of 100% MYI (see figure 10), which will undoubtedly introduce a high-bias into the MYI fraction product that they construct. I would urge the authors to go back and engage with my previous comment on this matter.**

*We thank the referee to have noticed this mistake regarding the NSIDC reference, it has been corrected in the manuscript. Concerning the "myi fraction", we are aware that it is a map of ice age corresponding to the oldest ice contained in a given grid cell. It is still possible to derive an information concerning the type of ice that occurs within this grid cell. Indeed, Yufang Ye et al. 2023 shows that in most of the case "NSIDC-SIA can generally capture the SITY distribution pattern but exhibits a slight over- or underestimation of MYI, which can be explained by the ice age assignment of the oldest ice and different temporal resolution of NSIDC-SIA compared to SAR. These results agree with previous studies (Korosov et al., 2018; Ye et al., 2019) and once again confirm the use of the NSIDC-SIA product as a cross-validation dataset".*

*To our knowledge, there are no available datasets that provide 30-years of MYI concentration. However, it is an important information to better estimate thickness from sea ice that can have similar backscatter but from different ice type (e.g rough ice and ice close to the edge). This MYI proportion will be overestimated but as there is nothing better, it gives the information we need, to recover what we are looking for: the radar freeboard.*

*The major concern here comes from the fact that it's not a proper multi-year ice fraction, but a mean ice type seen by the satellite. We propose to change this paragraph to inform readers on that point by adding these sentences:*

The proportion of multi-year ice of a given grid cell refers, in this study, to the mean ice type observed by all the tracks (for each month of each mission) that pass within a 25 km radius of this grid cell. This value is computed during the gridding step. The proportion would consequently be overestimated compared to what can be estimated with ice age tracking algorithms.

Ye, Y., Luo, Y., Sun, Y., Shokr, M., Aaboe, S., Girard-Ardhuin, F., Hui, F., Cheng, X., and Chen, Z.: Inter-comparison and evaluation of Arctic sea ice type products, The Cryosphere, 17, 279–308, https://doi.org/10.5194/tc-17-279-2023, 2023.

Korosov, A. A., Rampal, P., Pedersen, L. T., Saldo, R., Ye, Y., Heygster, G., Lavergne, T., Aaboe, S., and Girard-Ardhuin, F.: A new tracking algorithm for sea ice age distribution estimation, The Cryosphere, 12, 2073–2085, https://doi.org/10.5194/tc-12-2073-2018, 2018.
* * *
**L80: I don't think the change that you've put in your review responses has been included in the resubmitted manuscript? The text reads the same as before. I'm also still not convinced that SARM is less sensitive to a given roughness than LRM, as implied. I understand that the footprint is smaller, so potentially the footprint contains less variability in surface height (by perhaps including fewer floes of distinct freeboards). But isn't the point of altimetry to characterize some kind of average height? I can't see how a ridge in the footprint is less of a problem for a SAR waveform**

*The change is missing in the resubmitted manuscript, we apologize for this omission. It has been actualized according to this review.*

*Coming back to the relative impact of surface roughness between SAR and LRM, the SAR treatment haw two main effects regarding LRM : the first is to reduce the footprint significantly (with a ratio 20-40). and the second is to make this footprint constant. Indeed, the footprint of CS-2 SARM is about $5 km^2$. The reduced footprint allows to focus the measurements on or closer to the nadir. Moreover, as explained in Raney et al 1995., the footprint size is constant by construction in SARM, contrary to LRM. Indeed, the area illuminated by the altimeter depends on the surface roughness (Chelton et al 1989), Envisat and ERS footprint vary from 100 to 200 $km^2$ (studies mainly deal with oceanic surface, and the surface roughness over the ocean is characterized by the wave height). Thus, roughness impacts the waveforms, in particular the leading edge, especially in LRM. For oceanic surfaces, Brown's model deal with the roughness and can recover the average surface height and the roughness. During the retracking step, the modeled roughness is taken into account. Such a model, unfortunately, doesn't exist on sea ice. Heuristic approach gives an idea of the surface height impacted by a roughness. SARM sightly less impacted by this phenomenon thanks to its footprint and characteristics, (it is in some way an intermediary between LRM and laser technology). SARM heights are the only radar measurement that is directly usable on sea ice, whereas LRM as to be corrected from surface roughness using SARM. From this point of view, CS-2 was a real revolution.*

130     *Chelton, D. B., Walsh, E. J., and MacArthur, J. L.: Pulse Compression and Sea Level Tracking in Satellite Altimetry, Journal of Atmospheric and Oceanic Technology, 6, 407–438, https://doi.org/10.1175/1520-0426(1989)006<0407:PCASLT>2.0.CO;2, 1989.*

    *Raney, R.: A delay/Doppler radar altimeter for ice sheet monitoring, in: 1995 International Geoscience and Remote Sensing Sym-posium, IGARSS '95. Quantitative Remote Sensing for Science and Applications, vol. 2, pp. 862–864, IEEE, Firenze,*
135     *Italy,665 https://doi.org/10.1109/IGARSS.1995.521080, 1995*

    *We propose to replace :*

Contrary to SARM, LRM altimetry measurements are strongly impacted by the surface roughness of the surface illuminated by the radar, also affecting the freeboard measurement (Raney, 1995) (see Sect 3.4 for more information). Our approach is to make use of these processing mode differences to derive an LRM corrected freeboard. To that end, we compare Envisat and
140 CryoSat-2 datasets during missions-overlap period which runs from November 2010 to March 2012 (see Sect. 3.4).

    **by:**

The surface illuminated by the satellite is significantly larger in LRM than in SARM (by a factor of about 30). Moreover, in LRM the surface roughness conditioned the size of the illuminated footprint, the larger the roughness, the larger the footprint is, whereas it is constant in SARM (Chelton et al., 1989; Raney, 1995). Thus, surface roughness will impact more LRM range
145 retrieval than more nadir-focused measurement, SAR technologies (see Sect. 3.4 for more information). There is no waveform model for sea ice to account for the effect of roughness, and conventional retracking methods don't allow relevant radar freeboard estimation using LRM information alone. Our approach is to exploit these processing mode differences to derive an LRM-corrected freeboard. To this end, we compare Envisat and CryoSat-2 datasets during the mission overlap period from November 2010 to March 2012 (see Sect. 3.4).

150                     ————————————————————————

**L306: "Moreover, in order to avoid over-fitting, an early stopping criterion is used to stop the model training as soon as the score is not improved during 10 consecutive iterations, with a defined tolerance." I don't see how this would stop overfitting? You could easily have overfitted your model by the time that you fail to get an improvement in score, because overfitting can result from spurious improvements to the score that don't reflect improvements in the underlying model.**
155 **I think overfitting is normally identified through comparison of performance on test sets with training sets. In fact, I would suggest this model may well suffer from overfitting, due to the very low ratio of bias to variance (3mm to 9 cm in the Envisat-CS2 evaluation, 2mm and 3.8 cm for ERS/Envisat). This ratio is a well-known indicator of overfitting, and suggests your model has captured the noise in its training data as well as the underlying signal. 4 You should address this in your discussion.**

160 *Early stopping works with a validation dataset that is not used to fit the model during the training step and which is different from the testing set. Early stopping doesn't work with the training set, of course it would not be able to prevent any overfitting. We suggest to develop this part a way more in the manuscript to answer your concerns.*

    *We are not sure to understand where these numbers come from, nevertheless, all the statistics presented in section 4.2.1 and 4.2.2 are corrected radar freeboard, predicted from the NN that have been trained and the whole dataset (2010-2011/*
165 *2011/2012 or 2002/2003), minus the validation set that is randomly chosen. So we think that these numbers won't reveal any overfitting. However, unexpected a light overfitting can obviously occur. As mentioned by Jack Landy, comparisons with validation datasets show good agreement even for ERS-2, so even if we have a light overfitting, ERS-2 thickness remains consistent with validation dataset.*

                    ————————————————————————

170     **L300: If I understand right, you've specified a range of discrete hyperparameters and then investigated which combination optimizes the score. But this description should be contextualized with the ranges and intervals over which you searched, the dimensionality of the hyperparameter space that was searched, and most importantly the results of the search. You've made choices about the learning rate, regularization term and weights-solver based on your grid search,**

but you haven't reported what those choices were. Given you performed what must have been a highly multidimensional and computationally intensive score-sensitivity analysis to get at these specific hyperparameters, it would be great to know what they are. I'm also not totally clear how the activation function was both found through this grid-search, and also motivated by the domain of the TFMRA Rfbs. I guess the domain of possible activation functions was motivated by the domain of Rfbs, and then you ended up selecting the sigmoid specifically through the grid search method ?

*In the previous review it was suggested to justify the sigmoid choice compared to RELU, that the reason why we added a possible explanation for why sigmoid works better. However, all available activation functions and solvers were tested (with grid search) before choosing the best combination.*

*The hyperparameters selection is presented in appendices in the new version of the manuscript, as suggested.*
* * *
**L429: Replace "seems" with "is". It definitely is sensitive to the algorithm used.**

*Modified*
* * *
L34 of the manuscript now reads "Sea ice thickness estimation by spatial altimetry was first introduced by Laxon (1994) and Peacock and Laxon (2004) based on the freeboard methodology". I think the first real description of sea ice thickness estimation from radar altimetry derived freeboards was by Stanley et al. (1979). Their paper compared elevations from repeated tracks of the GEOS-3 altimeter in both the presence and absence of sea ice. They explained that the differences between the ocean elevations and the sea ice elevations (which were too large in their investigation due to poor gain control) contained information on the freeboard, which in turn could be used to estimate thickness if better constrained. The breakthrough of Laxon's work with ERS1 was the individual classification of waveforms into lead/floe, allowing effective interpolation of the local sea surface height – this unlocked the method. So I think to make Laxon (2004) the valid citation, you should just add something like "in its modern form" to your sentence.

*Taken into account.*
* * *
I can't see why a non-geophysical parameter such as "date" is being used to adjust the TFMRA waveforms (Figure 5). What physical mechanism could justify its inclusion in the neural network?

It seems to me that the inclusion of a date parameter will only make the neural network behave like a seasonal climatology (further reducing the ratio of bias against variability); this would evaluate very well against in-situ sources in the way that you've constructed your tests (given the dominance of seasonal variability over interannual variability in sea ice thickness), but would potentially be an example of spurious overfitting. At minimum, a bit of justification needs to be given for its inclusion, preferably with some information about the final effect of its inclusion.

*This choice comes from the seasonal variability of sea ice properties. Indeed, The physical properties of sea ice and snow on sea ice can be slightly different from one season to another, this will necessarily impact waveforms so the derived TFMRA50 FBr, LES, PP. Given the difference between SARM and LRM antenna gains, this impact will be different depending on the mode, and the correction will be more accurate if it takes into account this seasonality. Figure 1 shows BGEP A and C sit time series with NN FBr for Envisat and the NN FBr with a NN trained without month as a parameter. Envisat draft that has been trained with month information capture better the seasonal variability. Since the time series run before October 2010, it's shows that it doesn't lead to an overfitting as it works similarly for 2010-2012 than for some other winters, except for 2006-2008. For this two winters the draft varies a lot and its evolution differs from other winters, we can imagine that moorings captured floes from*

[Figure]

**Figure 1.** BGEP A (top) and D (bottom) daily and 31-days smoothed draft compared to Envisat NN draft between 2003 and 2012. In blue Envisat NN draft and in orange Envisat NN but trained without the month information.

*different types and that MYI edge was at moorings locations.*

215     *The following sentence has been added to justify the non-geophysical inputs, the period of the year :*

Note that the period of the year is taken to capture better the seasonal variability, as snow on sea ice as well as sea ice physical properties change along the seasons.

[Figure]

**Figure 2.** ERS-2 TFMRA50 radar freeboard before and after blurring correction compared to Envisat TFMRA50 radar freeboard for the missions overlap period (winter 2002-2003) with monthly maps.
* * *
**I think the language surrounding the pulse-blurring correction needs a bit of clarifying. The reader should be in no**
220 **doubt that a height correction has been applied to the retracked elevations to remove the effects of pulse blurring. The waveforms have not been "deblurred" and then retracked. I think the authors use of "corrected" is good, and I would suggest removing references to "deblurred" and "deblurring".**

*This remark has been taken into account.*

**On the pulse-blurring correction – have the authors checked to what degree this actually improves their radar free-**
225 **board estimates after the NN-based adjustment? Seems like this would be a fairly significant finding, and would justify both their efforts and their phrasing of it as a "correction".**

*We don't have trained the neural network on ERS2 blurred data, but we have evaluated the signal-to-noise ratio between ERS and Envisat before and after blurring correction. We showed in the past review as well as in the manuscript that the blurring essentially noise the heights and that this noise is highly asymmetric, which introduces a non-negligible bias. As presented in*
230 *the manuscript, correcting the blurring largely reduced the radar freeboard variability and after the blurring correction, its variability is similar to Envisat radar freeboard one (A1) and the mean bias between radar freeboards is divided by 5. 2*
*Nevertheless, since blurring is mainly noise, it is hard to quantify precisely the impact of the correction on monthly means and on another side as RA are sufficiently different to prevent from relevant along track deep comparisons.*

**L41: I think a better reference for snow penetration is potentially Ricker et al. (2015) 6, which is entirely focussed on**
235 **the topic.**

*This reference has been added.*

**Major comments**

*The purpose of this study was to present a consistent and homogeneous FBr for the ERS-2, Envisat and CS-2 missions. The challenges for LRM waveform retracking have been explained both in the previous review and in this one. We hope that the*
240 *additional explanation of LRM/SARM differences will help to address some concerns. To sum up, LRM waveform retracking*

*process need to take into account the surface properties more than SARM to have meaningfull FBr, so fixed threshold retracking in LRM can not capture the surface height for sea ice. To recover a consistent FBr we need at least information on observe roughness as demonstrated in previous studies (Guerreiro et al 2017 and Paul et al 2018). Readers should not expect TFMRA50 FBr to tell something on the SIT because LRM waveform fixed-threshold retracking is not enough to recover a valuable height in LRM on sea ice. Thus, TFMRA50 FBr and some waveform characteristics e.g. TFMRA50 heights, LES, PP, Sigma0 is expected to be enough to capture enought information to estimate the SIT. Moreover, month, concentration, MYI proportion are also used to give more information to the regression in order to capture better the seasonal variability, the spatial variability and improve the consistency with CS-2. So indisociably, TFMRA50 FBr and roughness will tell you a coherent FBr. Because MYI proportion is in a way linked to roughness, it will be correlated to the NN FBr, but we can still not still that because the correlation of the NN FBr is stronger with age/type than with TFMRA50 FBr, the TFMRA50 contribute relatively little compared to type/age in the regression. As shown in the previous review, regression without MYI proportion, concentration of month information give a meaningful FBr.*

*As requested, Figure 4 shows inputs in function of NN FBr. Month and MYI proportion are not represented as it is discrete values, scatters don't represent anything. As the purpose of the study is not to mathematically describe the relationship between parameters and NN FBr, we won't go further into details and propose to put this figure in Appendix. We will unfortunately not go further to understand the size and the nature of the correction so we will not explain more the impact of the different input parameters. We do think that it could be a full study to do it and that the understanding around sea ice roughness with altimetry and especially LRM altimetry does not enable to go further. Concerning the point that for some regions, the size of the correction is equal to TFMRA50, this is a pretty interesting results. This scatter shows that there is very few TFMRA50 FBr that have not been corrected, and that noisy TFMRA50 (close to marginal ice zone for instance) is set to 0, so the correction has to be equal to the TFMRA50. It thus gives interesting information but do not draw conclusion on the correction size for all the point, as it is only a low proportion of the dataset. These same points also correspond to Envisat NN FBr that are equal to zero see 4 .*

*Concerning Figure 3 of the previous review, we apologize for not responding to your concern. Indeed, Fig. 3 doesn't show whether NN trained with TFMRA50 FBr, show better results than without. Note that without TFMRA50, PP and LES the NN doesn't converge to consistent values.*

*We assume that theese conclusions (about the NN) are also true for ERS-2 as the RA as well as orbit are even more similar.*

**Answer to report #2**

*We would thank the reviewer for his comment, for helping and bringing clarification and supplementary information of the previous and present comments. As suggested, we have added some information within the introduction as well as in the conclusion to reframe the purpose of the study and give information to the readers on the usage of the time series.*

**Answers to report #3**

*Regarding your general reply: I appreciate the exhaustive analysis and the work the authors put into this review. There are two points that however still concern me:*

**1) All results and additional comparisons you show are with ENVISAT instead of ERS2. Wouldn't the authors agree that with all additional dead weight (pulse blurring, degradation of satellite systems, etc.) ERS2 is supposedly the sensor of interest here as it is clearly the NEW thing? Envisat was shown before in several studies that it can be made 'reliable'. I would love to see some comparison with ERS2 and in addition.**

**2) The authors went to such great lengths to put this analysis together but nothing of it appears to be in the resubmitted manuscript. I think the authors should give themselves credit for their additional work and put this analysis**

[Figure]

**Figure 3.** Envisat NN FBr compared to some of the regression input parameter, the sea ice concentration, the Leading Edge Slope, the Pulse Peakiness and TFRM50 FBr.

[Figure]

**Figure 4.** Envisat NN FBr compared to CS-2 with a NN trained with and without TFMRA50 FBr.

**into their discussion. Otherwise the raised points of concerns by the reviewers potentially remain also for any future readers.**

*1) If you refer to figures within the previous review, they were made to explain a bit more the objectif and to answer to some concerns about the methodology, but are not at all part of the purpose of the paper. We added some new comparisons in*
285 *appendices to show the impact of Blurring correction as requested by Robbie Mallet, but for other it will take a lot of time to do to order and to integrate knowing that it is not the purpose of the study. What is interesting is that the most difficult part of the "LRM correction" using this methodology, is between Envisat and CryoSat-2 as they don't have the same orbit neither the same altimeter, they do not observe the same sea ice floe at the same moment contrary to Envisat and ERS-2 that are few hours delayed with the same orbit. In other words, sea ice properties are much more comparable between Envisat and ERS-2 than*
290 *CS-2 and Envisat. If you refer to ERS-2 validation, there are several comparisons with independent dataset in the paper.*

*2) We hope the previous paragraphe will answer to your concern. Moroever, we added some new comparisons between inputs and outputs of the Neural network for Envisat. Similary it is just given as information but it is not a part of the study.*

**Regarding my general comments: Similarly, to my suggestions above, the comparison to Guerreiro et al, could be added to your manuscript or at least to the appendix.**

295 *Thanks for the suggestion of adding Envisat FBr from guerreiro et al 2017, it has been taken into account.*

Regarding my specific comments: L183 of the rebuttal: Minus the 'and' in front of ADAM.

*Taken into account*

**Regarding the Test/Train split: I found this quite surprising that a random split lead to perfectly equally distributed data sets? And this is valid for each of the 5(?) randomly chosen test runs per model setup?**

300 *Yes it is, the number of data that represent 10% of the missions overlap period is large enough to capture the distribution according to the law of large numbers.*

---

## Author Response (AR3)

Manuscript ID: egusphere-2022-214

**Title: Arctic sea ice radar freeboard retrieval from ERS-2 using altimetry : Toward sea ice thickness observation from 1995 to 2021**

*Marion Bocquet, Sara Fleury, Fanny Piras, Eero Rinne, Heidi Sallila, Florent Garnier, and Frédérique Rémy*

5

We appreciate the reviewer's recognition of our efforts in addressing his questions regarding the roles of altimetry and other parameters in the calibration procedure. We fully understand the nature of referee's last concern, we hope we have address it adequately. Our answer is divided in two parts :

10     1. In response to the referee's request, we have included the A2 figure for ERS-2 (see Fig.1. Statistics between missions are close except the correlation that falls to 0.07 for ERS-2. Regarding the scatter plot as well as the correlation coefficient, the both radar freeboards (TFMRA50 and NN) can not be reasonably identified as correlated. Furthermore, we would argue that a positive significant correlation would imply (among other things) that the lowest freeboard corresponds to the lowest corrected freeboard, which contradicts the LRM correction.

15     To our mind, this point is not linked to the usage of 'radar freeboard' in the paper. For instance, correlation between TFMRA50 Envisat radar freeboard and TFMRA50 CryoSat-2 radar freeboard (monthly gridded comparisons) is 0.18, however they are both considered as radar freeboard. (See Fig. 2).

    2. We agree that the term 'radar freeboard' used as it can be misleading (less now since we added 'NN', for Neural Network,
20 thanks to your suggestion).To avoid any confusion, every reference to a 'radar freeboard' should explicitly mention the type of radar, its frequency, the algorithm used, etc., for instance : LRM-KU-TFMRA50 radar freeboard in the case of Envisat, but it would not be digest...

    We can add 'proxy' before 'radar freeboard' but we are not convinced that it would help reader to understand better the nature of the time series. We suggest adding precision within the paper in order to explain and motivate the choice of the
25 wording 'radar freeboard' for each step/mission.

To justify a bit more:
    The definition of radar freeboard is the difference between the measured height over floes and the measured height over leads. However, regardless of the altimeter and mode, the height measurements over leads and floes are subject to various
30 sources of uncertainty. To date, only SAR mode estimates provide consistent heights over floes and leads, that is to say, a consistent radar freeboard with ground-based measurements. As discussed in this paper and in previous reviews, this is not the case for LRM mode, and our goal is to achieve a radar freeboard close to that of SAR for LRM. If we assume (as discussed in the paper) that the height bias over leads between the two modes is constant, we could have focused the correction on the height over floes with SAR TFMRA height over floes as a reference. However, we did not choose this approach as it
35 would require examining only co-located points between missions, significantly reducing the number of available points and introducing substantial data noise, thereby complicating the calibration. Nonetheless, it is an interesting but more cumbersome approach with some other limitations. Since our correction is applied to the radar freeboard, once the correction is determined, we directly correct it. Stephan Paul et al. (2018) employs a similar correction approach on the radar freeboard to determine the threshold of the TFMRA retracker over floes that should have been used if the radar freeboard matched to CryoSat-2 one.
40 However, our approach differs as we aim to work with monthly maps and avoid re-tracking waveforms, particularly for missions such as ERS where the gridding of data helps reduce the residual noise from pulse blurring. Therefore, we correct the radar freeboard, primarily focusing on correcting the height over floes since the bias on the SLA (Sea Level Anomaly) remains constant. The calibration approach can be seen complementary to the re-tracking step; we simply do not subtracted the SLA

[Figure]

**Figure 1.** An outline of the link established by the NN between some of the inputs (Standardized LES, PP, sea ice concentration and TFMRA50 FBr) and ERS-2 NN FBr. WPU means Waveform power unit, for ERS-2 correction

[Figure]

**Figure 2.** Envisat TFMRA50 radar freeboard against CryoSat-2 radar freeboard

and added it back afterward. Lastly, this radar freeboard is consistent with a SAR KU TFMRA50 radar freeboard, which is
45   coherent with supplementary validation data (in-situ, airborne, etc.) after conversion. This justifies our choice of using the term

"radar freeboard" accompanied by "NN" to indicate that it has undergone a different step from simple re-tracking and height difference. The radar freeboard thus becomes a kind of SAR TFMRA50 radar freeboard, a SAR-like FBr.

We propose adding further details in the introduction and a sentence in the conclusion. We hope that our response will enable us to converge faster.:

In the introduction :
*Since this LRM-TFMRA50 radar freeboard will be corrected to be consistent with CryoSat-2 and not conventionally obtained by making the difference of the height over floes and height over leads, it will be specified as NN FBr, which stands for radar freeboard adjusted using the neural network.*
In the conclusion :
*The final NN FBr does not conventionally result from a difference of two retracked heights, but corresponds to a TFMRA50 SAR-like radar freeboard corrected by a neural network.*